# Mitotic bookmarking redundancy by nuclear receptors in pluripotent cells

Almira Chervova[1,2,6], Amandine Molliex[1,2,6], H. Irem Baymaz[3], Rémi-Xavier Coux [1,2], Thaleia Papadopoulou [1,2], Florian Mueller[4], Eslande Hercul[1,2], David Fournier[1,2], Agnès Dubois[1,2], Nicolas Gaiani [1,2], Petra Beli[3,5], Nicola Festuccia [1,2]✉ & Pablo Navarro [1,2]✉

Mitotic bookmarking transcription factors (TFs) are thought to mediate rapid and accurate reactivation after mitotic gene silencing. However, the loss of individual bookmarking TFs often leads to the deregulation of only a small proportion of their mitotic targets, raising doubts on the biological significance and importance of their bookmarking function. Here we used targeted proteomics of the mitotic bookmarking TF ESRRB, an orphan nuclear receptor, to discover a large redundancy in mitotic binding among members of the protein super-family of nuclear receptors. Focusing on the nuclear receptor NR5A2, which together with ESRRB is essential in maintaining pluripotency in mouse embryonic stem cells, we demonstrate conjoint bookmarking activity of both factors on promoters and enhancers of a large fraction of active genes, particularly those most efficiently reactivated in G1. Upon fast and simultaneous degradation of both factors during mitotic exit, hundreds of mitotic targets of ESRRB/NR5A2, including key players of the pluripotency network, display attenuated transcriptional reactivation. We propose that redundancy in mitotic bookmarking TFs, especially nuclear receptors, confers robustness to the reestablishment of gene regulatory networks after mitosis.

During mitosis many transcription factors (TFs) are inactivated or evicted from the chromatin and transcription is halted; however, some TFs remain active and bind to a subset of their targets, typically containing numerous and high-quality binding motifs[1]. Traditionally, it has been considered that this phenomenon, known as mitotic bookmarking, enables daughter cells to promptly resume transcription of key genes upon mitotic exit[2]. However, the exact role and importance of these bookmarking factors is unclear. Indeed, only a small subset of mitotically bookmarked genes displays a clear, albeit partial, dependence on the presence of their respective mitotic bookmarking factor for proper reactivation[3–13]. This has led to the hypothesis that the effects of mitotic bookmarking TFs might be relatively minor and represent a mere consequence of other properties, such as chromatin modifications[13]. However, an alternative hypothesis is that of mitotic bookmarking redundancy: distinct mitotic bookmarking TFs may simultaneously and independently bookmark key genes important for cell identity such that the loss of one would be largely inconsequential.

The nuclear receptor ESRRB, a mitotic bookmarking TF that maintains nucleosome organization in mouse embryonic stem (ES) cells[5,10], represents an ideal candidate to assess the notion of mitotic bookmarking redundancy. Indeed, while mitotic ESRRB binds around a third of its interphase targets (~10,000 gene regulatory regions)[5,10],

[1]Department of Developmental and Stem Cell Biology, Institut Pasteur, Université Paris Cité, CNRS UMR3738, Epigenomics, Proliferation, and the Identity of Cells Unit, Paris, France. [2]Equipe Labéllisée Ligue Contre le cancer, Paris, France. [3]Institute of Molecular Biology, Mainz, Germany. [4]Department of Computational Biology, Institut Pasteur, Université Paris Cité, CNRS UMR3691, Imaging and Modeling Unit, Paris, France. [5]Institute of Developmental Biology and Neurobiology, Johannes Gutenberg-Universität, Mainz, Germany. [6]These authors contributed equally: Almira Chervova, Amandine Molliex. ✉e-mail: nicola.festuccia@pasteur.fr; pnavarro@pasteur.fr

only 150 genes require ESRRB to be properly reactivated immediately after mitosis[5]. Moreover, ES cells in which ESRRB or another nuclear receptor, NR5A2, is individually knocked out remain viable; in contrast, the simultaneous loss of both nuclear receptors is incompatible with self-renewal and the maintenance of pluripotency[14]. While the inactivation of either ESRRB or NR5A2 has minor molecular consequences, the loss of the two TFs triggers strongly reduced binding of major ES cell regulators, such as POU5F1 (hereafter OCT4), SOX2 and NANOG, at thousands of enhancers[14]. This strong complementarity between ESRRB and NR5A2 showcases the importance of the functional redundancy between nuclear receptors. In fact, a high level of redundancy among members of this super-family of TFs might be expected since they are evolutionarily and structurally related and share highly similar DNA-binding motifs[15–18]. Moreover, several nuclear receptors have been shown to coat mitotic chromosomes[19–25], even though the engagement in site-specific interactions genome wide has only been analyzed for ESRRB[5,10]. Together, these observations suggest that cohorts of nuclear receptors could be involved in mitotic bookmarking processes in conjunction with ESRRB to ensure the proper post-mitotic reactivation of the pluripotency network.

In this Article, to investigate mitotic bookmarking redundancy from the perspective of ESRRB, we first established that the most recurrent and prevalent proteins with which it associates on mitotic chromatin are nuclear receptors. Second, focusing on NR5A2 we showed that its retention on mitotic chromatin is long-lived and characterized by site-specific interactions at gene regulatory elements harboring a specific variant of the DNA-binding motif recognized by nuclear receptors. Third, we assessed the functional consequence of dual bookmarking by ESRRB and NR5A2 using Auxin-inducible degrons: we found both factors to be conjunctly required for efficiently reactivating a group of around 1,000 genes after mitosis. These ESRRB/NR5A2-responsive genes during the M–G1 transition are collectively downregulated during ES cell differentiation, transiently induced in pluripotent compartments of the early mouse embryo and enriched for pluripotency regulators. We conclude that nuclear receptors execute the key task of rapidly reinstating the gene regulatory networks supporting ES cell identity after mitosis.

## Results

### Mitotic association of ESRRB with nuclear receptors

Previous work showed that ESRRB interacts with a large number of chromatin-associated proteins, including chromatin remodelers, members of the basal transcriptional apparatus, pluripotency TFs and other nuclear receptors[26,27]. We therefore aimed at identifying which proteins are associated with ESRRB in interphase and in mitosis. To do so, we applied chromatin immunoprecipitation combined with mass spectrometry (ChIP–MS), a technique similar to chromatin immunoprecipitation followed by sequencing (ChIP–seq) but that uses mass spectrometry to identify the factors crosslinked to the immunoprecipitated protein[28], in our case ESRRB (Extended Data Fig. 1a). Since we have previously shown that fixation with disuccinimidyl glutarate (DSG) and formaldehyde (FA) greatly improves the detection of TF localization to mitotic chromosomes[10], we performed three replicate assays in such conditions, together with negative immunoprecipitation (IP) controls. We have also established that FA alone is enough to detect substantial numbers of ESRRB mitotic binding sites[10]. Therefore, we also performed ChIP–MS after crosslinking with either DSG and FA or FA only and compared the results to the respective inputs. In agreement with previous reports[26,27], we found known interactors of ESRRB, such as other pluripotency TFs and members of the Mediator, NuRD and Swi/Snf complexes, to be associated with ESRRB in asynchronous cells (Fig. 1a and Extended Data Fig. 1b). All these proteins were, however, largely undetectable in mitosis. Indeed, a more comprehensive analysis (Extended Data Fig. 1b) identified 105 proteins in asynchronous cells, belonging to different functional groups (Fig. 1b). Of those proteins,

we found only 16 that were identified in mitosis, regardless of the crosslinking strategy, and strongly enriched as compared to both the control IP and corresponding input (in red in Fig. 1b and Extended Data Fig. 1b). We also identified 21 additional proteins that were enriched as compared to the control IP but not to the input, raising the possibility that these are highly expressed proteins in mitotic cells leading to some level of unspecific detection (in pink in Fig. 1b and Extended Data Fig. 1b). Notably, none of the pluripotency TFs associated with ESRRB in interphase were identified in mitosis; in fact, the vast majority of proteins was undetectable in mitosis for every functional group except for one, nuclear receptors and related factors, which were almost all found associated with ESRRB in both asynchronous and mitotic cells (Fig. 1a and red- and pink-colored proteins in Fig. 1b). While the mitotic detection of selected proteins in most functional groups is interesting, as it suggests a priming mechanism mediated by ESRRB to recruit other partners after mitosis, the high occurrence of nuclear receptors associated with ESRRB in mitotic cells is compelling, representing half of the most confident mitotic hits (Extended Data Fig. 1b). The identified nuclear receptors, with established roles in ES cells[29], are the following: ESRRB itself but also the highly related ESRRA; NR0B1 (also known as DAX1), a nuclear receptor lacking a DBD; NR5A2 (also known as LRH1), whose functional role together with ESRRB has been already demonstrated in ES cells; and RXRB, a retinoic acid receptor with established roles in pluripotency and differentiation. In addition, other factors that were reported as interacting with nuclear receptors are also associated with ESRRB in asynchronous and mitotic chromatin, such as NSD1 (ref. 30), TRIM24 (ref. 31) and SNW1 (ref. 32). Among this small set of proteins, and given its major role in ES cells[14,27], we focused on NR5A2 for further validation and characterization.

### Long-lived chromatin retention of NR5A2 in mitosis

TFs acting as mitotic bookmarking factors often coat mitotic chromosomes. This is the case for ESRRB[5] as reproduced here using endogenously expressed ESRRB–TdTomato fusion proteins (Fig. 2a). As expected from our ChIP–MS results (Fig. 1), endogenously expressed NR5A2–green fluorescent protein (GFP) fusion proteins were detected with ESRRB on mitotic chromosomes (Fig. 2a). Moreover, the retention of these two factors on mitotic chromosomes is mutually independent, as established using previously described knockout (KO) cell lines[14]: in ESRRB KO mitotic cells, NR5A2 remains associated with the chromatin; conversely, in NR5A2 KO cells ESRRB still coats the mitotic chromosomes (Fig. 2b). These results are in accord with the large binding independence of the two factors to their common sites on DNA[14]. Despite this seemingly similar behavior, ESRRB and NR5A2 coating of the mitotic chromosomes also displays notable differences: while chromosomal coating by ESRRB is disrupted by FA crosslinking[10], as shown for many other TFs[33], that of NR5A2 is retained (Extended Data Fig. 2a). A potential explanation for this behavior, already reported for CTCF[11], could be that the residence time of NR5A2 on the chromatin is long, allowing FA crosslinking[10,33]. Accordingly, fluorescence recovery after photobleaching (FRAP) analyses performed in parallel for ESRRB and NR5A2 clearly show that, while fluorescence is rapidly recovered on bleached chromatin for ESRRB, in the case of NR5A2 the recovery is incomplete over 10 s (Fig. 2c). Even after long times following photobleaching (up to 100 s), NR5A2 signal does not recover entirely, indicating that the association of this TF with the chromatin is long lived. This is particularly evident in interphase, but holds true also in mitosis, where a seemingly immobile fraction of around 25% characterizes NR5A2 coating of the mitotic chromosomes (Fig. 2d). These results, in sharp contrast to most, if not all, other bookmarking TFs analyzed and more specifically to ESRRB (Extended Data Fig. 2b), set apart NR5A2 as a TF that is stably associated with the chromatin in interphase and in mitosis. Intrigued by this finding, we asked whether the DNA-binding domain (DBD) of NR5A2 was sufficient for such long-lived interactions by ectopically expressing DBD–GFP

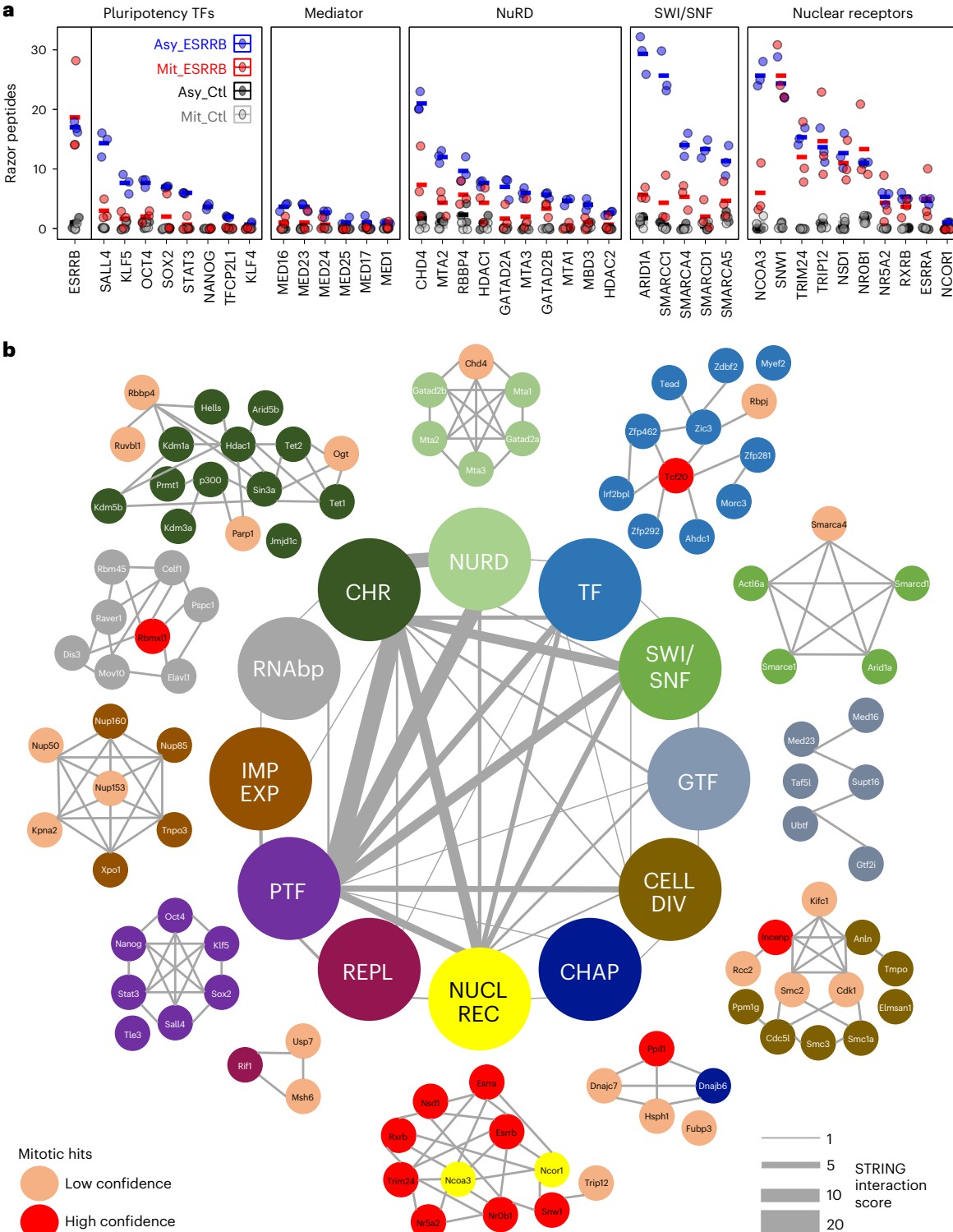

**Fig. 1 | Nuclear receptors are maintained on mitotic chromatin with ESRRB.**
**a**, Quantification of peptide abundance for specific examples belonging to different functional groups known to interact with ESRRB in interphase. The numbers of identified razor peptides are shown; similar observations were made with iBAQ and LFQ quantifications (see respective tabs in Source data). **b**, The ESRRB-centered, chromatin-associated proteome in interphase and mitotic cells, organized by functional groups. NURD, nucleosome remodeling and deacetylase complex; SWI/SNF, switch/sucrose nonfermentable nucleosome

remodeling complex; GTF, general transcription factors; CELL DIV, proteins with known roles during mitosis; CHAP, proteins with roles in chaperoning activity; NUCL REC, nuclear receptors and known associated factors; REPL, replication machinery; PTF, pluripotency transcription factors; IMP EXP, proteins with roles in nuclear import and export; RNAbp, RNA-binding proteins; CHR, proteins with known roles in chromatin regulation. Mitotic hits were grouped in two classes of low (pink) and high (red) confidence, depending on the fold change of the enrichment between the IP and the input observed in mitosis.

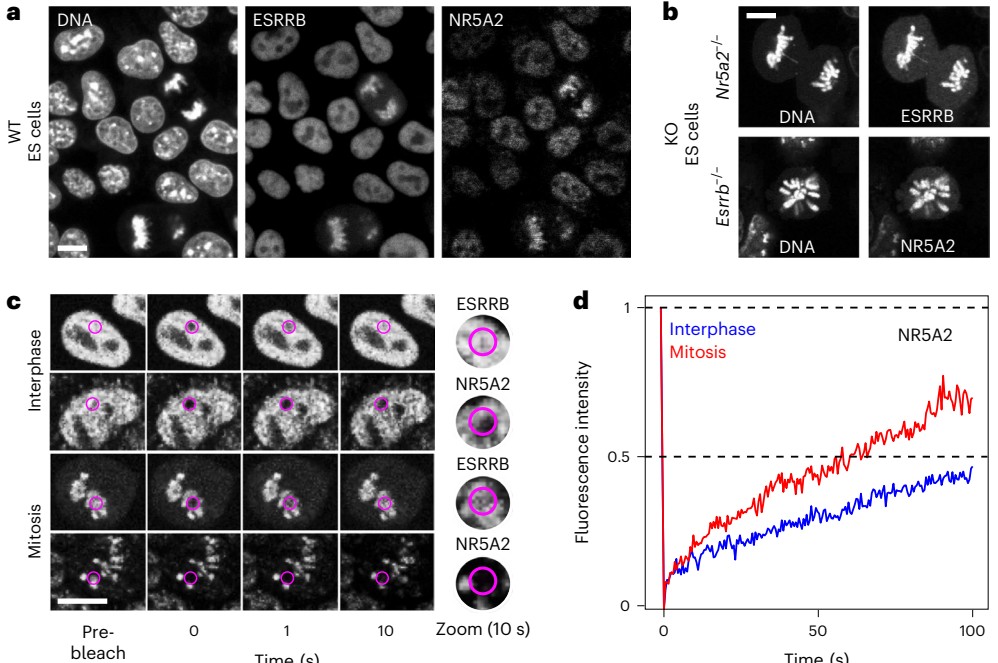

**Fig. 2 | NR5A2 is robustly bound to the chromatin in interphase and during mitosis. a**, Live imaging of ESRRB–TdTomato and NR5A2–GFP fusion proteins expressed from the endogenous loci in ES cells. Note the mitotic retention of both TFs. WT, wild type. **b**, Live imaging of ESRRB–GFP in NR5A2 KO cells and of NR5A2–GFP in ESRRB KO cells illustrating the mutually independent association with mitotic chromosomes. **c**, Representative example of FRAP assays in interphase and in mitosis for ESRRB–GFP and NR5A2–GFP fusions expressed from the endogenous loci. The pink area shows the photobleached region. **d**, Quantification of the recovery of fluorescence in cells endogenously expressing NR5A2–GFP, during 100 s following photobleaching (average of 29 and 22 cells analyzed in interphase and in mitosis, respectively; representative result of four independent assays). The white line represents 10 μm in all panels. All representative images were obtained from a minimum of two independent cell cultures.

fusion proteins. The recovery of fluorescence for the NR5A2 DBD alone was seemingly rapid in interphase (Extended Data Fig. 2c). Addition of all the remaining C-terminal domain that contains the ligand-binding domain, responsible for the interaction of nuclear receptors with coactivators, considerably increased the time of fluorescence recovery (Extended Data Fig. 2c). In mitosis, we found that both the DBD alone and the DBD–Ct fusion proteins efficiently coated the chromosomes (Extended Data Fig. 2d). Moreover, and in contrast to interphase, the dynamics of fluorescence recovery were found to be similar between the two constructs (Extended Data Fig. 2c). From these overexpression assays (Extended Data Fig. 2e), we conclude that the DBD alone is sufficient to trigger relatively long-lived chromatin interactions in mitosis, whereas in interphase the ligand-binding domain, possibly through the mediation of protein–protein interactions, confers further stability to NR5A2 binding to the chromatin.

**Mitotic bookmarking by NR5A2**
We next aimed at assessing whether NR5A2 engages in site-specific interactions with its DNA targets, a defining property of mitotic bookmarking factors[1]. To do this, we performed ChIP–seq assays in asynchronous and highly pure populations of mitotic ES cells. Exploration of the binding profiles throughout the genome confirmed the capacity of NR5A2 to bind mitotic chromatin at specific sites (Fig. 3a), which can be described with four main binding trends (Fig. 3b): regions that are bound by both factors in interphase and in mitosis, hereafter dB (for double bookmarked), regions that are bookmarked by either ESRRB (eB) or NR5A2 (nB) and regions that are bound by the two factors exclusively in asynchronous cells (hereafter lost regions, L). Globally, ESRRB and NR5A2 binding levels were higher at bookmarked regions, including in asynchronous cells (Extended Data Fig. 3a,b), in keeping with the idea that mitotic bookmarking often takes place at regions

of robust binding. The behavior of ESRRB and NR5A2 is not, however, fully symmetric. Indeed, at nB regions ESRRB is bound in interphase but mostly lost in mitosis whereas at eB regions NR5A2 is not efficiently recruited neither in interphase nor in mitosis (Fig. 3b and Extended Data Fig. 3a,b). This observation, together with our previous finding that NR5A2 and ESRRB show a preference for slightly different DNA motifs[14], prompted us to determine whether distinct sequences are enriched over these four groups of regions. For all groups, we found an overrepresentation of the TCAAGGTCA sequence characteristic of estrogen-related receptors (Fig. 3c), which contains the classical AGGTCA box of nuclear receptors[34,35], extended by a half-site. However, at regions bound by NR5A2 in mitosis (dB and nB), thymidine (T) was less prominent at the seventh base, and as frequent there as cytosine (C). The occurrence of T or C at the seventh position in the motif was already identified as favoring ESRRB versus NR5A2 binding in asynchronous ES cells[14]. Analysis of the presence of the motif and more particularly of T/C variants across all regions showed that the efficiency of mitotic bookmarking correlates with the presence of this consensus, with loci containing both T and C variants, or exclusively C, being strongly enriched at dB and nB regions (Fig. 3d); in contrast, eB regions were exclusively associated with motifs containing a T. As expected, L regions showed the lowest occurrence of motifs, which almost always contain a T. Moreover, we observed that the type of motifs present in the regions (Fig. 3e), as well as their number per region (Fig. 3f) and quality (Fig. 3g), is quantitatively associated with the binding levels of ESRRB and NR5A2, particularly in mitosis. Hence, the average motif score of the regions clearly differentiates bookmarked versus lost status (Fig. 3h). Finally, we noted that the motif identified at dB and nB regions was longer, containing an extra AGT at the 5′ end (Fig. 3c). While we had already identified the presence of this longer motif in ESRRB bookmarked regions[5], this new analysis clearly

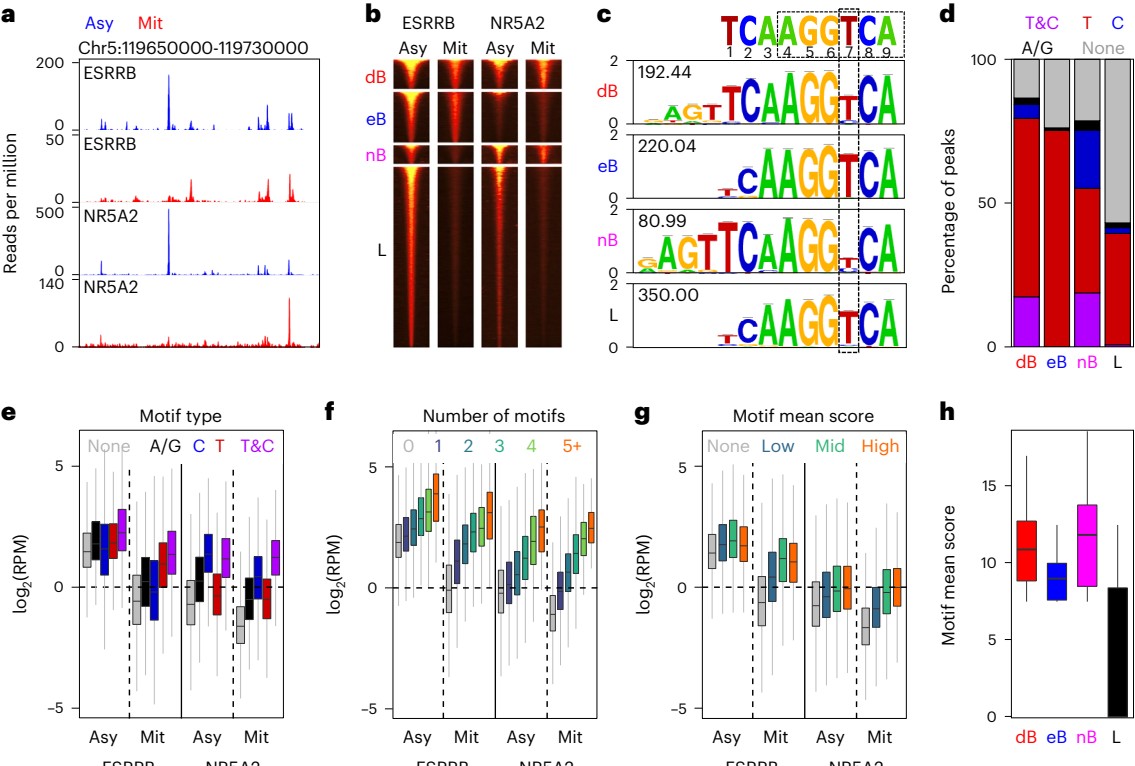

**Fig. 3 | Mitotic bookmarking by NR5A2 of a subset of ESRRB/NR5A2 target loci. a**, Representative binding profiles of ESRRB and NR5A2 in asynchronous (Asy) and mitotic (Mit) cells. **b**, Clustering of ESRRB/NR5A2 binding regions as double bookmarked (dB), bookmarked by ESRRB (eB) or by NR5A2 (nB) only, or exclusive to interphase (L, for lost in mitosis). **c**, Consensus motifs found de novo at the four clusters described in **b**, compared to the orphan nuclear receptor cognate sequence. The number within each motif denotes the corresponding –log$_{10}$(FDR). The seventh base partially discriminating ESRRB (T) versus NR5A2 (C) preferential binding is highlighted. **d**, Percentage of peaks harboring both T and C motifs, or exclusively T, C, A/G or no motifs. **e**–**g**, Enrichment levels (reads per million, RPM) of ESRRB and NR5A2 in asynchronous (Asy) and mitotic (Mit) cells as a function of the presence of different motif types (**e**), the total number of motifs per region (**f**) or the mean score of all the motifs per region (**g**). **h**, The motif mean score per region shows drastic changes between mitotically bookmarked (dB, eB and nB) and lost (L) regions. All boxplots in **e**–**h** represent the median as the horizontal bar, 25–75% percentiles as the box and 1.5-folds the interquartile range as whiskers. All ESRRB and NR5A2 quantifications in this figure were obtained by averaging two (ESRRB) and three (NR5A2) independent replicates in interphase and in mitosis.

associates it with bookmarking by NR5A2, with or without ESRRB. Overall, we conclude that the presence of different versions of the ESRRB/NR5A2 motif, together with their degree of similarity to the consensus and their number of occurrences per region, is directly related to the behavior of ESRRB and NR5A2 in mitosis; in interphase, though, ESRRB and NR5A2 can also be recruited by other TFs, often excluded from mitotic chromatin (Fig. 1).

**Chromatin states of ESRRB/NR5A2 binding regions**

We next separated ESRRB/NR5A2 binding regions using epigenomic signatures characteristic of active promoters, active enhancers or enhancers lacking marks of activity, and quantified pluripotency TF binding (OCT4, SOX2 and NANOG; Fig. 4a). At enhancers, we observed that pluripotency TFs were almost exclusively constrained to L regions displaying poor ESRRB/NR5A2 motifs (Fig. 4a), indicating that ESRRB and NR5A2 are probably indirectly recruited at these regions. This subset was also characterized by slightly more accessible chromatin and p300 recruitment (Fig. 4a and Extended Data Fig. 4a). At active enhancers and promoters this trend was not apparent (Fig. 4a); instead, we found a small positive correlation between marks of activity and mitotic bookmarking status, with dB globally displaying higher levels of active marks than eB/nB, which in turn showed more enrichment than L regions (Extended Data Fig. 4a). Moreover, although the effect was rather modest, dB and eB regions also displayed a less pronounced reduction in accessibility in mitosis compared to asynchronous cells

(Extended Data Fig. 4a). We also observed that, proportionally, active promoters display the highest frequency of bookmarked regions (50% against 20–30% for other elements; Fig. 4a and Extended Data Fig. 4b). This is reflected by a two- to threefold enrichment of promoters within dB and eB regions, which nevertheless remain in absolute terms less frequently bound by these factors than enhancers (Extended Data Fig. 4b). Overall, this analysis indicates that at enhancers losing ESRRB/NR5A2 in mitosis, NANOG, OCT4 and SOX2 are probably key regulators triggering ESRRB/NR5A2 recruitment in interphase and a modest enrichment for epigenomic features associated with activity. In contrast, at active regulatory elements, ESRRB and NR5A2 have a positive impact in pluripotency TF binding and enrichment for active marks, particularly at regions where ESRRB/NR5A2 engage in mitotic bookmarking. Next, we turned to the analysis of nucleosome organization, since we had previously shown that ESRRB and CTCF bookmarked regions maintain nucleosome order in mitosis[10,11]. In this regard, nB regions are particularly interesting as they were previously considered as lost regions. Using previously published analyses of nucleosome mapping using micrococcal nuclease sequencing[10], we observed that at all bookmarked regions (dB, eB and nB), the nucleosomes were ordered as nucleosomal arrays in both interphase and mitosis (Fig. 4b). While the best phasing of the nucleosomes and the more pronounced nucleosome depleted region over the motif were observed at dB regions, both eB and more significantly nB regions displayed substantial maintenance of nucleosome organization. In contrast, L regions showed poor organization and

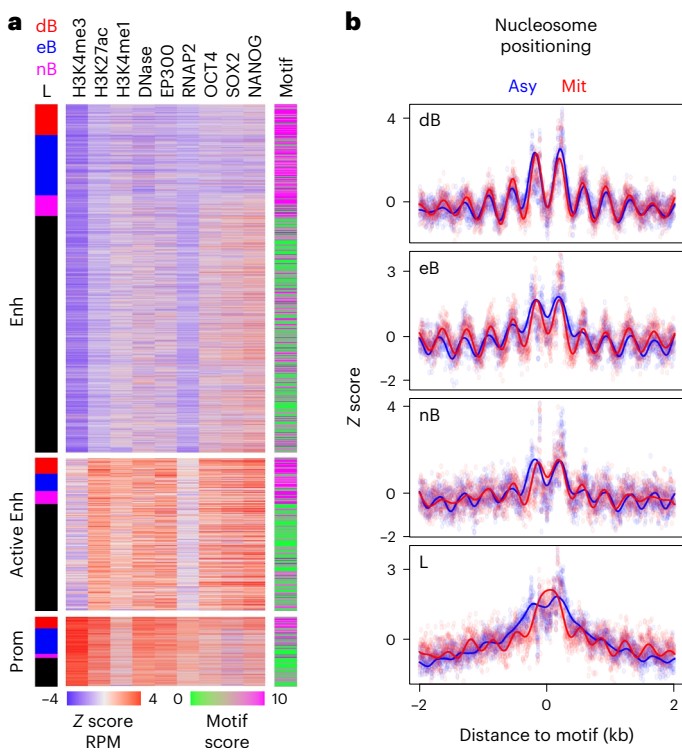

**Fig. 4 | The chromatin status of ESRRB/NR5A2 binding regions. a**, Enrichment levels of marks of activity (H3K4me3, H3K27ac, H3K4me1, DNase accessibility, EP300 and RNAP2) as established by Encode and characteristic of enhancers (Enh), active enhancers (Active Enh) or promoters (Prom), as well as of TF binding (OCT4, SOX2 and NANOG) across all ESRRB/NR5A2 binding regions. Left: the attribution as dB, eB, nB and L. Right: the mean motif score per region. **b**, Average nucleosome positioning profiles across dB, eB, nB and L regions in asynchronous (Asy) and mitotic (mit) cells. The dots represent raw data and the lines a Gaussian process regression.

a clear central accumulation of nucleosomes, specifically in mitosis, as previously observed for regions losing TF binding during division[10,11]. Therefore, the nucleosome organization capacity of ESRRB is also shared by NR5A2.

## ESRRB/NR5A2 mark rapidly reactivated genes

We next aimed at establishing whether ESRRB/NR5A2 binding, particularly during mitosis, is associated with post-mitotic gene transcription dynamics. To simplify these analyses, we focused on three binding groups: regions bookmarked by two factors (dB), regions bookmarked by a single factor (B) and regions losing both factors in mitosis (L). We considered a gene as a target when either a 2-kb-long window centered on its transcription start site, or an enhancer to which it had been previously linked using 3D conformational data and epigenomic analyses[36], overlapped with ESRRB/NR5A2 binding regions. This association was hierarchical: when a gene promoter or enhancer was overlapped by a dB region, it was labeled as dB and the remaining genes were subsequently labeled, in order, as B, L or unbound. This led to gene groups of similar size: dB, 3,529; B, 2,594; L, 2,671; unbound, 5,163. Next, we plotted the post-mitotic transcription dynamics of these 14,000 genes (Fig. 5a) using highly temporally resolved data[9], and computed the proportion of genes in each dB/B/L/unbound category for sliding windows of 1,000 genes (step = 10) displaying continuously increasing reactivation intensities. We observed that ESRRB/NR5A2 binding regions were progressively enriched as genes reactivate faster and more strongly (Fig. 5b, left, and Extended Data Fig. 5a). Moreover, this progressive increase of the enrichment was solely due to regions bookmarked by both ESRRB and NR5A2 (dB, Fig. 5b, right). This indicates that the combined mitotic

bookmarking by ESRRB and NR5A2 may drive efficient post-mitotic gene transcription. To assess this more quantitatively, we calculated the mean transcription profile of dB, B and L genes and compared them to unbound genes: all three groups of ESRRB/NR5A2-bound genes reactivated faster and more drastically than genes not bound by the two nuclear receptors (Fig. 5c, left). Moreover, dB genes were by far the most efficiently reactivated genes, followed by B and L genes, which displayed relatively similar kinetics. Splitting the associations by the type of regulatory element further showed that genes bookmarked by ESRRB and NR5A2 reactivate more robustly than unbound genes, whether they bind at promoters or at enhancers (Extended Data Fig. 5b). However, only promoters showed some level of increased reactivation when bookmarked by a single nuclear receptor as compared to unbound promoters (Extended Data Fig. 5b). Moreover, ignoring promoters and enhancers, and associating genes to dB, B and L groups by proximity (<50 kb), further confirmed that both the timing and the strength of gene reactivation were favored by ESRRB/NR5A2 binding in mitosis and to a lower extent by binding of a single factor (Extended Data Fig. 5b). Altogether, these observations support the notion that bookmarking factors promote gene reactivation in daughter cells, especially when two TFs are involved. In addition, the effect of double mitotic bookmarking by ESRRB and NR5A2 was observed both for genes that are, and that are not, targeted by other TFs such as NANOG, OCT4 and SOX2 (Fig. 5c, middle and right). Having established that mitotic bookmarking by ESRRB and NR5A2 is associated with strong post-mitotic gene reactivation, we aimed at analyzing more closely their link to the speed of the reactivation. For this, we ordered genes by the percentage of transcription with respect to the last time point analyzed (Extended Data Fig. 6a,b). We observed that the most rapidly reactivated genes were also enriched for ESRRB/NR5A2 mitotic bookmarking (Extended Data Fig. 6c). Thus, genes bookmarked by the two TFs are both more strongly and more rapidly reactivated. Finally, to provide functional evidence of such correlations, we tagged each of the alleles of ESRRB and NR5A2 with the Auxin-induced degradation domain, which we previously used to efficiently degrade CTCF in mitosis and upon release into interphase[9,11]. Accordingly, treatment with the auxin analog 5-Ph-IAA (hereafter IAA) led to a drastic reduction of both ESRRB and NR5A2 protein levels in interphase and during mitosis (Extended Data Fig. 7a), associated with a nearly complete loss of mitotic chromosome coating as judged by immunofluorescence (Extended Data Fig. 7b,c). Western-blot analysis of ESRRB/NR5A2 upon biochemical fractionation of mitotic cells further confirmed the acute depletion of these two TFs from the chromatin (Extended Data Fig. 7d). Thus, we proceeded to synchronize the cells in mitosis with a two-step approach, first inhibiting CDK1 to enrich ES cells in G2, and then inhibiting microtubule dynamics to arrest cells in pro-metaphase. During the second step we added IAA to initiate ESRRB/NR5A2 degradation as the cells enter mitosis and kept it throughout the release into the next interphase (Extended Data Fig. 7e), as described[9]. We collected multiple time points after mitosis for RNA extraction, and, after controlling the efficient depletion of ESRRB/NR5A2 in three independent replicates (Extended Data Fig. 7f), we prepared total ribodepleted RNA sequencing (RNA-seq) libraries to quantify pre-mRNAs as a proxy of transcriptional activity[9]. Contrary to our expectations, we did not observe major differences in the global reactivation dynamics of ESRRB/NR5A2-depleted cells compared to their respective control (Extended Data Fig. 8a–c).

## Gene activation by ESRRB/NR5A2 during the M–G1 transition

While the global effects of the double depletion of ESRRB/NR5A2 are minor, further exploration of the dataset enabled us to identify two principal component analysis (PCA) dimensions (principal components PC5 and PC6) describing clear differences between IAA-treated and control cells, even if capturing a small proportion of the total variance (Fig. 6a). This was expected given that most of the variance in this dataset is captured by the very vast transcriptional changes taking

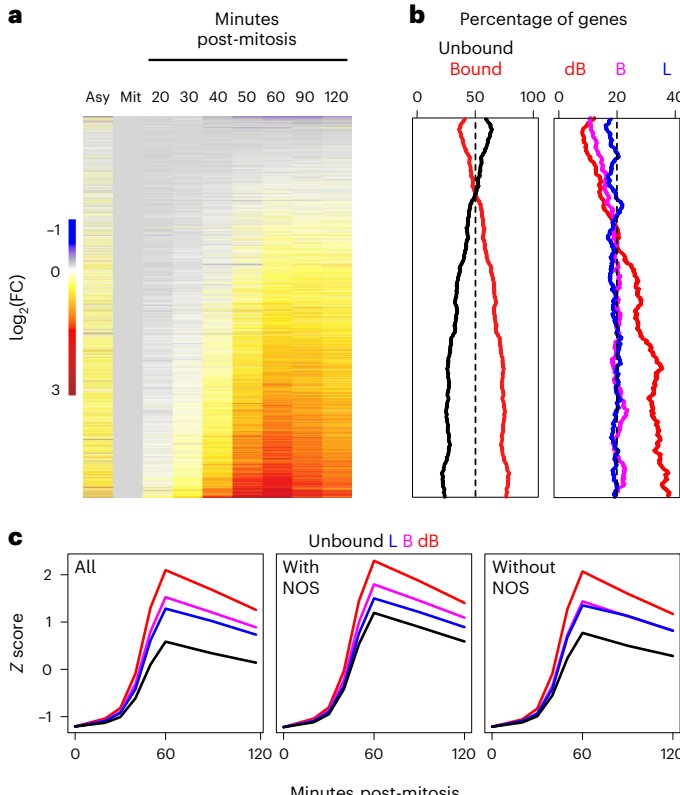

**Fig. 5 | Mitotic bookmarking by ESRRB and NR5A2 is associated with fast post-mitotic gene reactivation. a**, Heatmap of post-mitotic transcription levels (pre-mRNA), represented as the fold change (FC) to mitosis (Mit), in asynchronous (Asy) as well as after different minutes following mitosis, ordered by increasing mean reactivation. **b**, Percentage of genes either bound or not by ESRRB/NR5A2 at known promoters or enhancers (left), or assigned to dB, B and L regions (right), for groups of 1,000 genes sliding from top to bottom of the heatmap shown in **a**, with a step of ten genes. **c**, Average post-mitotic reactivation dynamics of genes bound by ESRRB/NR5A2 at known promoters or enhancers and characterized as dB, B or L. On the left, all genes are shown, and in the middle and on the right those are further split by the presence/absence of either NANOG or OCT4 or SOX2 (NOS).

place after mitosis[9]. Extraction of the genes contributing mostly to PC5 and PC6 identified two groups that were either down- or upregulated upon IAA treatment (Fig. 6b) and displayed concordant regulation in independent datasets generated in inducible ESRRB/NR5A2 double-KO ES cells[14] (Extended Data Fig. 8d), indicating that they represent bona fide ESRRB/NR5A2-responsive genes. Notably, while downregulated genes responded throughout the whole time-course analysis, the upregulation observed was largely not statistically significant (Fig. 6b). Gene Ontology and gene set enrichment analyses showed that downregulated genes were strongly enriched in members of the pluripotency network, and to a lesser extent in metabolic pathways (Fig. 6c). Indeed, several pluripotency TFs were found downregulated after mitosis, albeit at variable levels, with some being more affected than others by ESRRB/NR5A2 depletion (Fig. 6d). Prompted by these results, we aimed at comprehensively identifying ESRRB/NR5A2 responsive genes after mitosis, by using direct statistical comparisons between control and IAA treated cells (Extended Data Fig. 9a), as well as by taking advantage of previously identified targets[14]. Combined, we found 1,013 genes that were downregulated after mitosis in the absence of ESRRB/NR5A2 and 941 that were upregulated, which displayed a similar behavior (Extended Data Fig. 8e) and functional associations to those observed in the more restricted gene sets extracted from PCA (Fig. 6b,c). Of note, only 42 downregulated and 38 upregulated genes

were found to be differentially transcribed during early G1 upon the single loss of ESRRB[5], supporting the notion of ESRRB/NR5A2 redundancy. Importantly, we also found a strong association between the ESRRB/NR5A2 binding status and the gene's responsiveness to ESRRB/NR5A2 depletion (chi-squared $P$ value $5.84 \times 10^{-77}$), with around 85% of downregulated genes being bound at known regulatory elements (Fig. 6e) and around 50% being bookmarked by both ESRRB and NR5A2 (Fig. 6e), which represents a strong and significant association (Fig. 6f). In contrast, upregulated or nonresponsive genes were partially depleted of double bookmarked sites (Fig. 6e,f). Moreover, when we analyzed previous genes responsive to the long-term loss of ESRRB and NR5A2, we observed that only those displaying attenuated reactivation upon IAA treatment (Extended Data Fig. 9b) were associated with regions bookmarked by both ESRRB and NR5A2 (Fisher exact test $P = 8.736 \times 10^{-8}$, odds ratio 2.54). We conclude that the combined bookmarking activity of ESRRB and NR5A2 at promoters and enhancers primarily fosters the activation of around 1,000 genes enriched in regulators or markers of pluripotency, including important regulators of self-renewal such as *Tfcp2l1*, *Tbx3*, *Nanog* or *Klf5*. Accordingly, we observed that the depletion of both TFs during a single M–G1 transition led to reduced self-renewal efficiency, although with high intrinsic variability between independent assays, due to the extensive manipulation of the cells (Extended Data Fig. 9c). Hence, while the quantitative consequences of ablating ESRRB/NR5A2 during the M–G1 transition are relatively modest, they are associated with measurable phenotypical consequences. Furthermore, it remains possible that additional mitotic bookmarking by other nuclear receptors (Fig. 1) partially compensates for the loss of ESRRB/NR5A2. In this regard, it is noteworthy that both the short motif identified at regions exclusively bookmarked by ESRRB as well as the long motif identified at NR5A2-bookmarked regions compose the core nuclear receptor binding consensus and, in particular, perfectly match the ESRRA and RXRB motifs, respectively (Extended Data Fig. 10a). Moreover, both ESRRA and RXRB are present in the mitotic ESRRB proteome we report (Fig. 1), suggesting their direct contribution to the post-mitotic reactivation of ESRRB/NR5A2 target genes. In agreement, preliminary observations indicate that ESRRA binds at regions targeted by ESRRB and NR5A2 in interphase and displays weak but consistent mitotic bookmarking activity at regions bookmarked by ESRRB (Extended Data Fig. 10b,c), as predicted by the analysis of the motifs. Therefore, the activity of a complex network of nuclear receptors acting redundantly might ensure the robustness of mitotic bookmarking. Finally, we analyzed the expression dynamics of ESRRB/NR5A2 responsive genes after mitosis during embryoid body differentiation[37] and early mouse embryogenesis[38,39]. We found that genes activated by ESRRB/NR5A2 after mitosis are rapidly downregulated upon in vitro differentiation (Fig. 6g, left, and Extended Data Fig. 8f) and, conversely, transiently upregulated in the naive pluripotent compartments of the early blastocyst (Fig. 6g, right, and Extended Data Fig. 8g). Conversely, genes repressed by ESRRB/NR5A2 after mitosis are globally upregulated during differentiation (Fig. 6g, left, and Extended Data Fig. 8f) and, in vivo, maternally inherited, cleared and then re-expressed at E5.5 (Fig. 6g, right, and Extended Data Fig. 8g), when the dismantlement of naive pluripotency starts. These observations provide further support to the biological significance of the group of genes activated by ESRRB/NR5A2 during the M–G1 transition and strongly associated to their mitotic bookmarking activity.

## Discussion

The pluripotency network is composed of a plethora of TFs that promote self-renewal and preserve pluripotency. This network is believed to be robust against the loss of single factors, since only the individual depletion of OCT4 or SOX2 leads to drastic differentiation[40,41]. This is in part due to TF redundancy, as computationally predicted[42] and as shown for KLF2/KLF4/KLF5 (ref. 43) or for ESRRB/NR5A2 (ref. 14). Nevertheless, self-renewal necessarily involves undergoing mitosis,

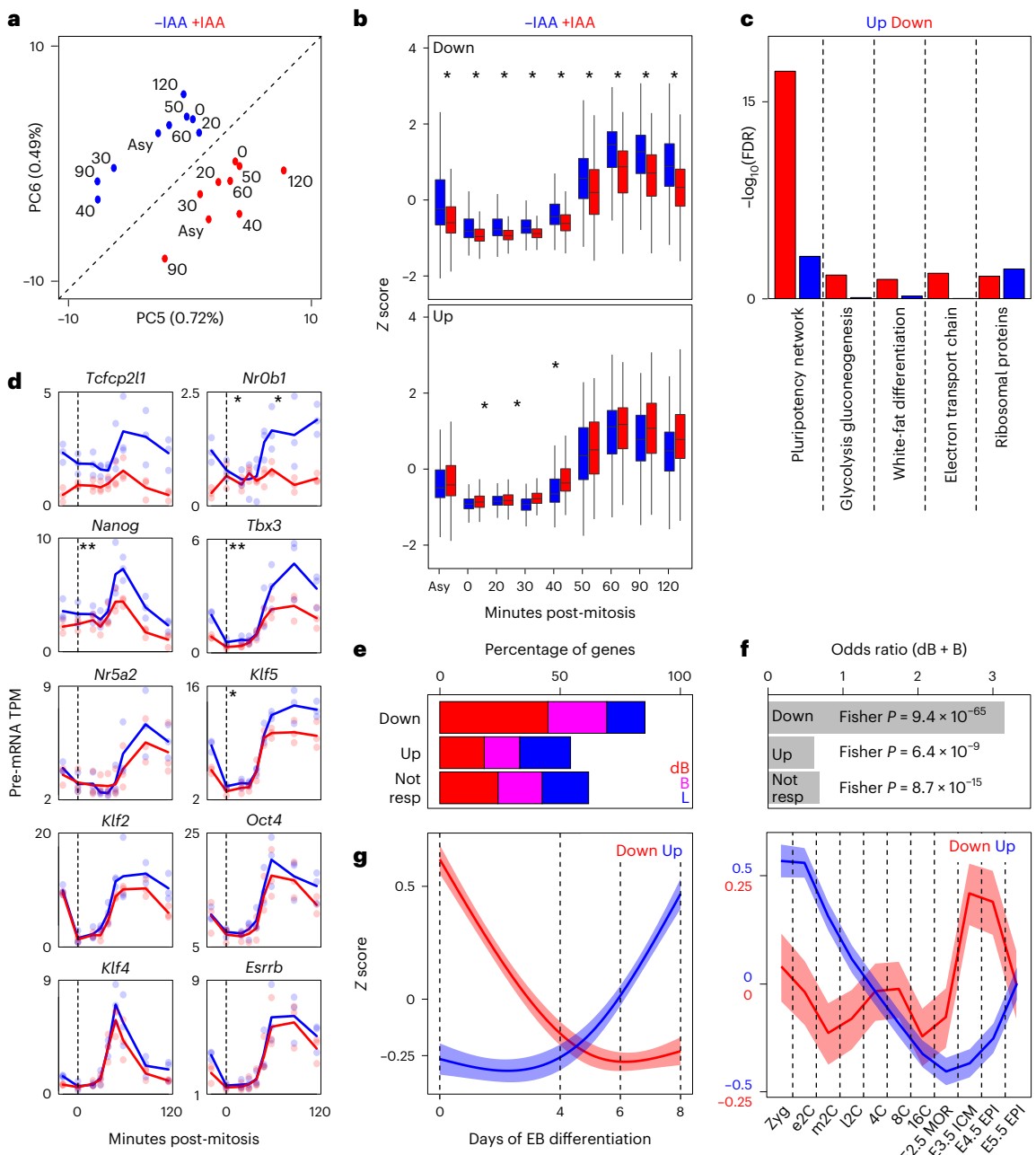

**Fig. 6 | Mitotic bookmarking by ESRRB/NR5A2 promotes the reactivation of genes associated with pluripotency. a**, PCA of post-mitotic gene transcription dynamics of cells exiting mitosis in either the presence (−IAA) or absence (+IAA) of ESRRB/NR5A2. PC5 and PC6 capture the differences dependent on ESRRB/NR5A2 depletion. **b**, Post-mitotic gene transcription dynamics of genes derived from the loadings of PC5/PC6 and displaying either a downregulation (top) or an upregulation (bottom) upon ESRRB/NR5A2 depletion. The asterisks denote $P < 0.05$ (one-sided Student's $t$-test). Error bars represent standard deviation of three independent experiments for each time point and condition. **c**, Gene Ontology analyses of the gene groups shown in **b**. **d**, Examples of post-mitotic gene reactivation dynamics for genes belonging to the pluripotency network, in the presence (blue) or the absence (red) of ESRRB/NR5A2. Two asterisks denote genes with a global difference during the whole kinetic with $P < 0.05$ (one-sided Student's $t$-test); one asterisk only during the last three time points. **e**, Percentage of genes associated with ESRRB/NR5A2 dB, B (eB + nB) or L regions, for extended sets identified as downregulated, upregulated or not responsive. **f**, Statistical analysis (two-sided Fisher exact test) of the association of the sets shown in **e** with bookmarked regions (dB + eB + nB). **g**, Mean expression profile during embryoid body (EB) differentiation (left) or early mouse embryogenesis (right) of genes downregulated or upregulated upon ESRRB/NR5A2 depletion during the M–G1 transition. Zyg, zygote; e/m/l2C, early/mid/late two-cell stage; 4/8/16C, 4/8/16-cell stage; MOR, morula; ICM, inner cell mass; EPI, epiblast. The colored band represents 99% confidence interval for the smoothing.

which represents an obstacle to the regulation mediated by most TFs[1]. Therefore, specific mechanisms may have evolved to ensure the regulatory continuity throughout cell generations and to prevent inappropriate escape from pluripotency. Mitotic bookmarking offers, in this regard, a suitable mechanism to facilitate the reassembly of functional regulatory complexes after mitosis and promote target gene reactivation. In this study we have identified a family of gene regulators, nuclear receptors, as potentially common mitotic bookmarking factors in ES cells, and have focused on two, ESRRB and NR5A2, to further show that they bind together at a large subset of their mitotic targets. Using an Auxin-inducible protein degradation system we further show that the reactivation of ESRRB/NR5A2 mitotic targets is partially compromised

when they are simultaneously degraded in mitosis and during reentry in interphase. However, current experimental limitations linked to the kinetics of ESRRB/NR5A2 degradation/resynthesis upon Auxin treatment/washout do not enable specifically testing the role of mitotic binding per se versus an independent role during mitotic exit. Alternative tools enabling extremely fast modulation of TF localization and/or activity rather than expression will now need to be developed to address this key question rigorously.

While ESRRB and NR5A2 are structurally related and their binding profiles highly similar, they can display different biophysical behaviors, with NR5A2 displaying unusually long binding to mitotic chromatin. These two TFs also display subtle differences in terms of the DNA-binding motifs with which they preferentially interact, which become particularly relevant in the context of mitosis where the chromatin poses additional constraints for TF binding as compared to interphase. Notwithstanding, they functionally work together to maintain ordered nucleosomal arrays at their targets in mitosis. Given that both ESRRB and NR5A2 are simultaneously required to recruit NANOG, OCT4 and SOX2 at a large number of regulatory elements, notably enhancers[14], it is likely that these ordered arrays facilitate the reassembly of functional enhanceosomes to promote gene reactivation in daughter cells. However, the consequences of the loss of ESRRB/NR5A2 during the M–G1 transition are, for most genes, relatively modest, except for prominent examples such as *Nanog*, *Tfcp2l1* or *Tbx3*. Moreover, genes losing ESRRB/NR5A2 binding in mitosis, or genes controlled by TFs that do not act as mitotic bookmarking factors in our cells[10] (OCT4, SOX2 and NANOG), are also associated with accelerated transcription patterns after mitosis in comparison with unbound genes. These observations may be ascribed to three major phenomena. First, additional bookmarking factors may cooperate with ESRRB/NR5A2, as shown for ESRRA, or target independent regions. Second, the global coating of mitotic chromosomes by ESRRB/NR5A2 and other TFs[44], or even their mere cytoplasmic inheritance, may also be important to rapidly activate target genes in G1, even if not linked to site-specific interactions with DNA during mitosis. In particular, TFs with pioneer activities such as ESRRB[45], NR5A2 (ref. [45]), OCT4 (ref. [46]) and SOX2 (ref. [46]) may accelerate transcriptional reactivation of their interphasic targets compared to unbound genes. Third, the global, strong and transient hyper-transcriptional burst taking place after mitosis[47], which is particularly prominent in ES cells[9], may be robust enough to buffer the loss of ESRRB/NR5A2. Promoters may be subject to additional mechanisms fueling their post-mitotic activation, such as particularly well-preserved accessibility[48] or histone acetylation[7], mitotic bookmarking by TBP[8] or by CTCF[9], or APC/C-driven regulation[49]. Hence, in this permissive context the additional contribution of ESRRB/NR5A2 might have a minor impact. Considering these three nonmutually exclusive aspects, and given the role of ESRRB/NR5A2 in enabling enhancer occupancy by other pluripotency TFs in interphase[14], it is likely that their mitotic bookmarking activity is more strongly associated with enhancer function and, therefore, with the modulation of transcriptional levels rather than with off–on switching processes. Notably, the genes most influenced by ESRRB/NR5A2 mitotic bookmarking include factors of particular importance for ES cell identity, as suggested by the specificity in their developmental pattern of expression and their enrichment in regulators of the pluripotency network. This might be a general characteristic of the genes responding during M/G1 to the action of cell type-specific mitotic bookmarking TFs, as supported by the recurrent identification of cell identity genes within repertoires of mitotic bookmarking targets[3–5].

Overall, we argue that the control of post-mitotic transcription is twofold. On the one hand, yet to be fully characterized general mechanisms are responsible for the global transcriptional reactivation burst, probably operating at promoters. On the other hand, cohorts of redundant mitotic bookmarking activities, such as nuclear receptors, impart specificity to the kinetics and amplitude of gene reactivation by modulating enhancer activity. This may confer the required robustness to the reactivation of cell type-specific genes and gene regulatory networks preserving cell identity in a manner that is largely resilient to the loss of one or several regulators. Although challenging, it will be important in the future to establish new tools to invalidate the activity of all nuclear receptors identified as mitotic ESRRB binding partners, such that the importance and redundancy of this process can be fully tested. Furthermore, extending our meticulous analysis of mitotic binding events and their consequences to other cell types where nuclear receptors have been shown to at least coat mitotic chromosomes[19–25] may enable generalization of the role of nuclear receptors as mitotic bookmarking factors.

## Online content

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

## Methods

### Cell culture and mitotic preparations

ES cells—E14Tg2a, EKOiE[5], EKOie–NrKO[14], FLAG–Nr5a2 (ref. [14]), NR5A2–GFP/ESRRB–mCherry[14], ESRRB/NR5A2–GFP lines and ESRRB/NR5A2–IAA ESCs (see details below)—were cultured on serum and leukemia inhibitory factor conditions as previously described[11]. Mitotic ES cells (>95% purity as assessed by 4′,6-diamidino-2-phenylindole staining and microscopy) were obtained using a double synchronization method based on the CDK1 inhibitor RO-3306 (10 μM; Sigma, SML0569), nocodazole (50 ng ml⁻¹; Sigma, M1404) and shake-off, as previously described[9]. For post-mitosis analyses, cells were seeded in separate dishes (one per time point), purposely uncoated with gelatin and lysed in cold TRIzol (ThermoFisher, 15596026) 20, 30, 40, 50, 60, 90 and 120 min after release from the mitotic block[9]. ESRRB/NR5A2 depletion was achieved with 0.5 mM auxin (5-Ph-IAA BioAcademia, 30-003), added during the 5 h of nocodazole block and maintained during the whole post-mitotic release. Asynchronous cells were treated in parallel during 5 h.

### ES cell derivation

ESRRB/NR5A2–GFP cells were generated by stable transfection of a CAG-driven vector expressing C-terminal fusions of ESRRB or NR5A2 variants to GFP (connected by a glycine linker and linked to an IRES–puromycin resistance cassette) and selection of single clones. NR5A2 variants included the full protein from ENSEMBL transcript *Nr5a2-205* (Uniprot Q1WLP7) and two truncated version coding for the DBD alone or for a fragment spanning the DBD and all the remaining C-terminal portion of the protein (amino acids: DEDLEE … LHAKRA). For experiments shown in Fig. 2b, similar expression constructs were transiently transfected by lipofection in EKOiE, or EKOie–NrKO cells. ESRRB/NR5A2–AID cells were generated by CRISPR–Cas9, first inserting a CAG–OsTir2–T2a–NeomycinR cassette at the TIGRE locus (gRNA 3′-ACTGCCATAACACCTAACTT-5′), then a LoxP–PuromycinR–LoxP–HA–AID–Gly5 cassette at the start codon of *Nr5a2* (ENSEMBL transcript Nr5a2-205; gRNA 5′-CCACTTTGGGCAGCATGACA-3′) and finally a LoxP–PuromycinR–LoxP–3xFLAG–Gly5 cassette at the start codon of *Esrrb* (ENSEMBL transcript Esrrb-206; gRNA 5′-TGAACCGAATGTCGTCCGAC-3′). After each round, single colonies were expanded and cells homozygous for correctly targeted alleles identified by polymerase chain reaction on genomic DNA and sequencing. In addition, after each insertion of the AID degron, the selection cassette was removed by Cre-mediated recombination. All cell lines are available upon request.

### Protein analyses

To establish the ESRRB-centered proteome, we used both asynchronous and mitotic cells obtained during successive experimental rounds, fixed with DSG (2 mM; Sigma, 80424-5 mg) for 50 min at room temperature (RT) followed by 10 min with FA (1%, Thermo, 28908), sonicated as previously described[10] and stored at −80 °C until 300 × 10⁶ cells for each were accumulated. Next, ESRRB and control IPs were performed in parallel in triplicates using 50 × 10⁶ cells per IP and a standard ChIP procedure with anti-ESRRB (Perseus Proteomics, H6-705-00) and control antibodies, except that after the last wash the beads were resuspended in 2× lithium dodecyl sulfate buffer/100 mM dithiothreitol (DTT), incubated for 35 min at 95 °C while shaking and spun for 10 min at RT at maximum speed. Samples were stored at −20 °C until further processing. The eluates, after equilibrating their temperature to RT, were alkylated by incubating with 5.5 mM chloroacetamide for 30 min in the dark and then loaded onto 4–12% gradient sodium dodecyl sulfate–polyacrylamide gel electrophoresis gels. Proteins were stained using the Colloidal Blue Staining Kit (Life Technologies). Due to DSG/FA crosslinking, the proteins appeared as a smear upon sodium dodecyl sulfate–polyacrylamide gel electrophoresis; therefore, the bands corresponding to heavy and light chains of the IP antibodies were not cut out to avoid losing potentially relevant proteins. All proteins were digested in-gel using trypsin. Peptides were extracted from gel and desalted on reversed phase C18 StageTips. Peptide fractions were analyzed on a quadrupole Orbitrap mass spectrometer (Q Exactive Plus, Thermo Scientific) equipped with an ultrahigh-performance liquid chromatography system (EASY-nLC 1000, Thermo Scientific). Peptide samples were loaded onto C18 reversed phase columns (15 cm length, 75 μm inner diameter, 1.9 μm bead size) and eluted with a linear gradient from 8% to 40% acetonitrile containing 0.1% formic acid in 2 h. The mass spectrometer was operated in data-dependent mode, automatically switching between MS and MS2 acquisition. Survey full-scan MS spectra ($m/z$ 300–1,700) were acquired in the Orbitrap. The ten most intense ions were sequentially isolated and fragmented by higher-energy C-trap dissociation. An ion selection threshold of 5,000 was used. Peptides with unassigned charge states, as well as with charge states less than +2, were excluded from fragmentation. Fragment spectra were acquired in the Orbitrap mass analyzer. Raw data files were analyzed using MaxQuant (development version 1.5.2.8)[50]. Parent ion and MS2 spectra were searched against a database containing all mouse protein sequences obtained from the UniProtKB released in 2016 using Andromeda search engine[51]. Spectra were searched with a mass tolerance of 6 ppm in MS mode, 20 ppm in higher-energy C-trap dissociation MS2 mode, strict trypsin specificity and allowing up to three miscleavages. Cysteine carbamidomethylation was searched as a fixed modification, whereas protein N-terminal acetylation and methionine oxidation were searched as variable modifications. The dataset was filtered on the basis of posterior error probability to arrive at a false discovery rate (FDR) of below 1% estimated using a target-decoy approach[52]. All the proteins identified by MS in triplicate control and ESRRB IPs in asynchronous and mitotic cells, along with quantifications, normalized intensities and additional metrics, are provided as Source data. Razor and unique peptides were used to compute a fold enrichment and $P$ value between the ESRRB and control IP in either asynchronous or mitotic cells using the DEseq package[53] as described[54]. Proteins displaying a fold change above 5 and a $P$ value below 0.05 in either asynchronous or mitotic cells were selected for further analyses. This list was filtered on the basis of 'Reverse' hits, 'Contaminants' and 'identified by site' parameters. All detected immunoglobulins as well as other proteins belonging to the top 1,000 frequently identified proteins in MS datasets (https://www.thegpm.org/lists/index.html) were ignored. This led us to 105 proteins identified as associated with ESRRB in either asynchronous or mitotic cells, available in Extended Data Fig. 1 and in Source data. Proteins displaying a $P$ value below 0.05 were considered as positive mitotic hits. However, we further classified them as high or low confidence depending on their general abundance in mitotic cells and their identification after FA-only fixation. To do this, we performed an additional round of MS using the mitotic replicate 2 for ESRRB IP together with its corresponding input (DSG/FA fixed) as well as an IP/Input generated in parallel after FA fixation. Proteins displaying higher Razor and unique peptides in both the DSG/FA and FA IPs compared to their respective inputs were considered of high confidence (Source data). These 105 hits were analyzed using the STRING database[55]: first, all proteins were used as an input to identify functionally related groups (subnetworks) based on Ontology annotations; second, all possible functional and biochemical interactions between each protein and the rest were computed; third, subnetworks were connected using the sum of all the interaction scores existing between all individual proteins of each group. To monitor the efficiency of ESRRB/NR5A2 depletion from mitotic chromatin upon IAA treatment, 10⁷ mitotic cells were lysed in Buffer A (10 mM HEPES pH 7.9, 10 mM KCl, 1.5 mM MgCl₂, 0.34 M sucrose, 10% glycerol and 1 mM DTT) supplemented with complete protease inhibitors and 0.1% Triton X-100. After centrifugation, the supernatant was set apart as the cytosolic fraction and the pellets were further lysed in Buffer B (3 mM ethylenediaminetetraacetic acid, 0.2 mM egtazic acid and 1 mM DTT)

supplemented with complete protease inhibitors. After centrifugation the pellets (chromatin fraction) were resuspended in 2× reducing Laemmli buffer with 5% β-mercaptoethanol, sonicated (ten cycles; 30 s on/30 s off in a Bioruptor X machine, Diagenode), boiled and centrifugated at max speed at RT. The cytosolic fraction was precipitated with TCA stock solutions 100% (w/v), washed with cold acetone, dried on a heat block at 95 °C and resuspended in 2× reducing Laemmli buffer + 5% β-mercaptoethanol. The two fractions were analyzed by western blot.

## Imaging

For immunofluorescence, cells were plated on IBIDI hitreat plates coated overnight with poly-L-ornithine 0.01% (Sigma, cat. no. P4957) at 4 °C, washed and coated 2 h with 10 μg ml⁻¹ laminin (Millipore, cat. no. CC095). Fixation and immunofluorescence were performed with either paraformaldehyde or DSG + paraformaldehyde, as described[10], using 2 μg ml⁻¹ mouse monoclonal anti-flag for ESRRB (M2 Sigma, F3165) and 1 μg ml⁻¹ polyclonal rabbit anti-HA for NR5A2 (Abcam, ab9110) antibodies. Images were acquired with a LSM900 Zeiss microscope using a 64× oil-immersion objective. For live imaging and FRAP analyses, ES cells expressing fluorescent protein fusions were grown on IBIDI plates, incubated with 250 nM Hoechst-33342 for 30 min before imaging and imaged at 37 °C in a humidified atmosphere (7% $CO_2$). Images were acquired with a 63× oil immersion objective on a Nikon Ti2E equipped with a Yokagawa CSU W1 spinning disk module and a Photometrics sCMOS Prime 95B camera. For FRAP, fluorescence recovery was analyzed every 0.3–1 s (NR5A2) and 50–100 ms (ESRRB) after photobleaching (500 μs pulse, spot of 0.5 × 0.5 μm size) in MATLAB as described previously[10,56] and the plots corrected to min = 0 and max = 1.

## Identification of ESRRB/NR5A2 binding sites displaying distinct behaviors in mitosis

NR5A2 ChIP–seq was performed in FLAG–Nr5a2 mitotic cells as previously described and in parallel with already published datasets generated in asynchronous cells[14]. Briefly, cells were crosslinked with DSG (2 mM; Sigma, 80424-5mg) for 50 min at RT followed by 10 min with FA (1%, Thermo, 28908). Cells were then sonicated with a Bioruptor Pico (Diagenode) and immunoprecipitated with anti-flag antibodies (M2 Sigma, F3165). Precipitated DNA was used for library preparations[10] and sequenced externally by Novogene Co Ltd. Reads were aligned with Bowtie2 (ref. [57]) to the mm10 genome; only those with a single discovered alignment were kept. Peaks were called against relevant inputs for all mitotic samples using MACS2 (ref. [58]) and filtered to have (1) MACS2 FDR <0.05 in all three replicates, (2) mean enrichment over the input >2 and (3) FDR of the enrichment over the input <0.05, calculated with a previously described generalized linear model[10]. The resulting mitotic NR5A2 peaks were combined with previous collections of confident ESRRB/NR5A2 peaks[10,14] and mitotic bookmarking calls for ESRRB[10]. Finally, peaks were annotated as dB/eB/nB/L. The compendium of ESRRB/NR5A2 regions, their quantifications and mitotic bookmarking status are available as Source data.

## Identification of differentially expressed genes during the M–G1 transition

RNA was extracted with 500 μl TRIzol (ThermoFisher, 15596026), treated with DNAse I (Qiagen) and used for the generation of Ribo-depleted, stranded and paired-end RNA-seq libraries, prepared and sequenced by Novogene. Pre-mRNA levels were quantified exactly as previously described[9] and those with more than 0.1 transcripts per million in at least one sample were kept for further analyses. Differentially expressed genes were obtained from two separate analyses: first, using previous lists of ESRRB/NR5A2-responsive genes[14], where we selected genes with an FDR <0.05 upon the double knockout of the two TFs, quantified their pre-mRNA levels across the datasets generated here, and identified those displaying concordant changes in plus/minus IAA during the M–G1 transition using

*k*-means clustering, and, second, using a direct comparative strategy of each time point analyzed here and keeping those with an FDR <0.1 (DEseq2 (ref. [53])). A fully annotated table with all genes considered, quantifications and differential expression parameters is provided in Source data.

## Bioinformatic analyses

Analyses were performed in R (version 3.6.3). ChIP–seq quantifications were performed with the bamsignals package with systematic correction to the library sizes and counting the number of reads either falling into peak coordinates (for boxplots) or covering each base of a 2-kb window centered on the middle of the peak (for enrichment heatmaps and metaplots). Boxplots (median, bar; 25–75% percentiles, box; 1.5-fold the inter-quartile range, whiskers) and metaplots were visualized with ggplot2 package[59] and heatmaps with ComplexHeatmap package[60]. To characterize the regions as promoters (high H3K4me3), active enhancers (high H3K27ac) or enhancers (high H3K4me1), available ES cell data from the Encode consortium were used. All additional histone modifications and DNase accessibility datasets were downloaded from Encode. Assay for transposase-accessible chromatin with sequencing as well as ChIP–seq for NANOG, OCT4 and SOX2 was previously published[10,61]. All quantifications are available in Source data. To identify the most prevalent DNA motif associated with distinct ESRRB/NR5A2 binding regions, as well as their precise genomic coordinates, we used RSAT[62] with the command -markov auto -disco oligos -nmotifs 1 -minol 6 -maxol 8 -merge_lengths -2str -origin start -scan_markov 1. For each motif, the presence of a T, a C or an A/G at the seventh position of the estrogen-related receptor consensus was manually annotated. The information relative to the motifs can be found in Source data. Nucleosome positioning plots were generated considering midpoints of 140–200 bp fragments from published micrococcal nuclease sequencing paired-end datasets[10], after correcting the MNase-driven bias with a *k*-mer approach and smoothing the average profiles with a Gaussian process regression, as described[10]. When multiple motifs were available per region, only one was selected to center nucleosome positioning plots on its 5′ end; the following prioritized criteria were used: (1) best similarity score to the consensus, (2) closest distance to the middle of the peak and (3) highest enrichment for small MNase fragments (footprints <100 bp). Genes were associated to the different groups of ESRRB/NR5A2 binding regions when either their promoter regions, defined as 2-kb-long regions centered on the 5′ ends of known mRNA isoforms, or putative enhancers, defined by their epigenomic status and 3D interaction contacts as independently reported[36], overlapped with our set of ChIP–seq peaks. The approach was hierarchical, favoring dB associations over eB/nB and then L regions; genes lacking ESRRB/NR5A2 binding at these regions were qualified as unbound (Source data). The resulting groups of genes were used to plot average reactivation trends and the proportion of dB/eB/nB/L genes displaying increased reactivation dynamics. PCA of the RNA-seq was run with the prcomp function with centered data corresponding to log₂(transcripts per million (TPM)) to capture direct differences between IAA treatments. The genes more prominently contributing to PC5/PC6 were identified with the boxplot.stats function applied to the 'rotation' output of prcomp. Gene Ontology and gene set enrichment analyses were performed using the enrichR package[63]; the full set of statistically significant terms can be found in Source data. The relationships between ESRRB/NR5A2 binding at promoters/enhancers and gene responsiveness to ESRRB/NR5A2 depletion were assessed with a contingency table and chi-squared test followed by individual two-sided Fisher's exact tests for specific combinations of gene bookmarking and responsiveness.

## Reporting summary

Further information on research design is available in the Nature Portfolio Reporting Summary linked to this article.

## Data availability

The mass spectrometry proteomics data have been deposited to the ProteomeXchange Consortium via the PRIDE partner repository[64] (PXD038950). NR5A2 ChIP–seq and RNA-seq of ESRRB/NR5A2–AID cells exiting mitosis are available at Gene Expression Omnibus (GSE220253). Source data are provided with this paper.

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

## Acknowledgements

We acknowledge M. Cohen-Tannoudji, T. Gregor, G. Cecere, K. McElreavy and S. Tajbakhsh for critical reading of the manuscript. We also acknowledge the Biomics, Flow Cytometry, Photonic BioImaging and Image Analysis Hub platforms of Institut Pasteur. P.N. acknowledges the Institut Pasteur, the CNRS and Revive (Investissement d'Avenir; ANR-10-LABX-73) for recurrent funding and the Agence Nationale de la Recherche (ANR 20CE12002801 CHRODYNE), Ligue Contre le Cancer (LNCC EL2018 NAVARRO) and the European Research Council (ERC-CoG-2017 BIND) for financial support. The funders had no role in study design, data collection and analysis, decision to publish or preparation of the manuscript.

## Author contributions

A.C. performed all bioinformatic analyses with help from D.F. A.M. performed most of the experiments except ChIP–seq and cell line derivation (N.F.). H.I.B. and P.B. performed proteomics with samples gathered by A.M. with help from E.H. T.P. established the initial conditions for proteomics. F.M. provided help to analyze FRAP data. A.D. and N.G. provided technical help and R.-X.C. provided determinant help during the revision of this manuscript. N.F. and P.N. conceived the project and analyzed the data. P.N. wrote the manuscript.

## Competing interests

The authors declare no competing interests.

## Additional information

**Extended data** is available for this paper at https://doi.org/10.1038/s41594-023-01195-1.

**Correspondence and requests for materials** should be addressed to Nicola Festuccia or Pablo Navarro.

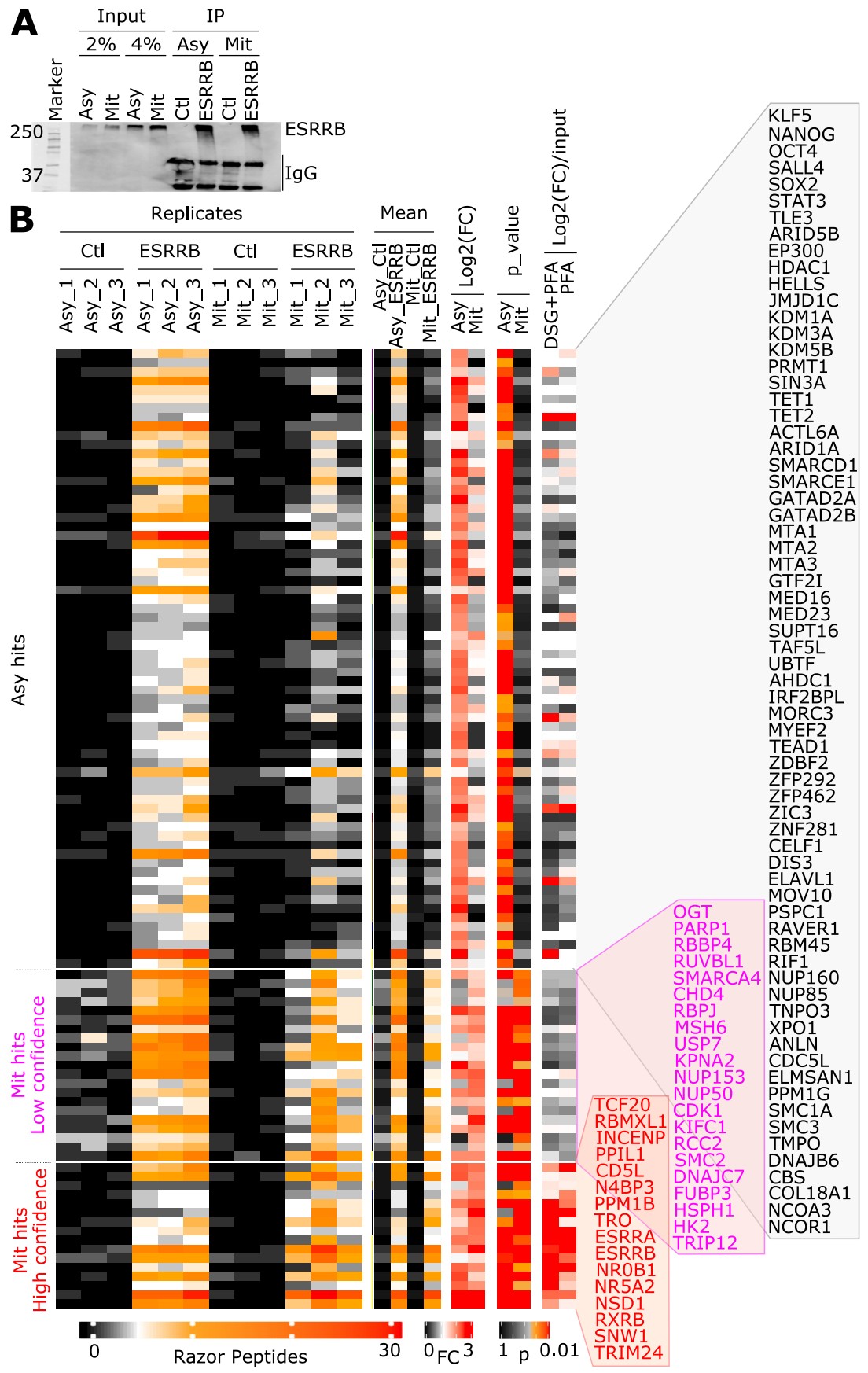

**Extended Data Fig. 1 | See next page for caption.**

**Extended Data Fig. 1 | Quantifications of all protein hits in interphase and in mitosis.** (A) Representative immunoprecipitation (IP) of ESRRB from DSG/FA-fixed chromatin. Asy: asynchronous cells; Mit: mitotic cells; Ctl: blank immunoprecipitation. Four independent experiments showed similar efficiencies. (B) Quantifications (Razor peptides), fold changes and associated p-values (calculated with a generalized linear model in DEseq[53]) used to categorize the proteomic hits as exclusive of interphase (ASY hits) or also found in mitosis (Mit hits). Two categories of mitotic hits were made, based on the fold-change of ESRRB IP versus the INPUT in mitosis (last parameters on the right of the heatmap). All the proteins belonging to each category are shown.

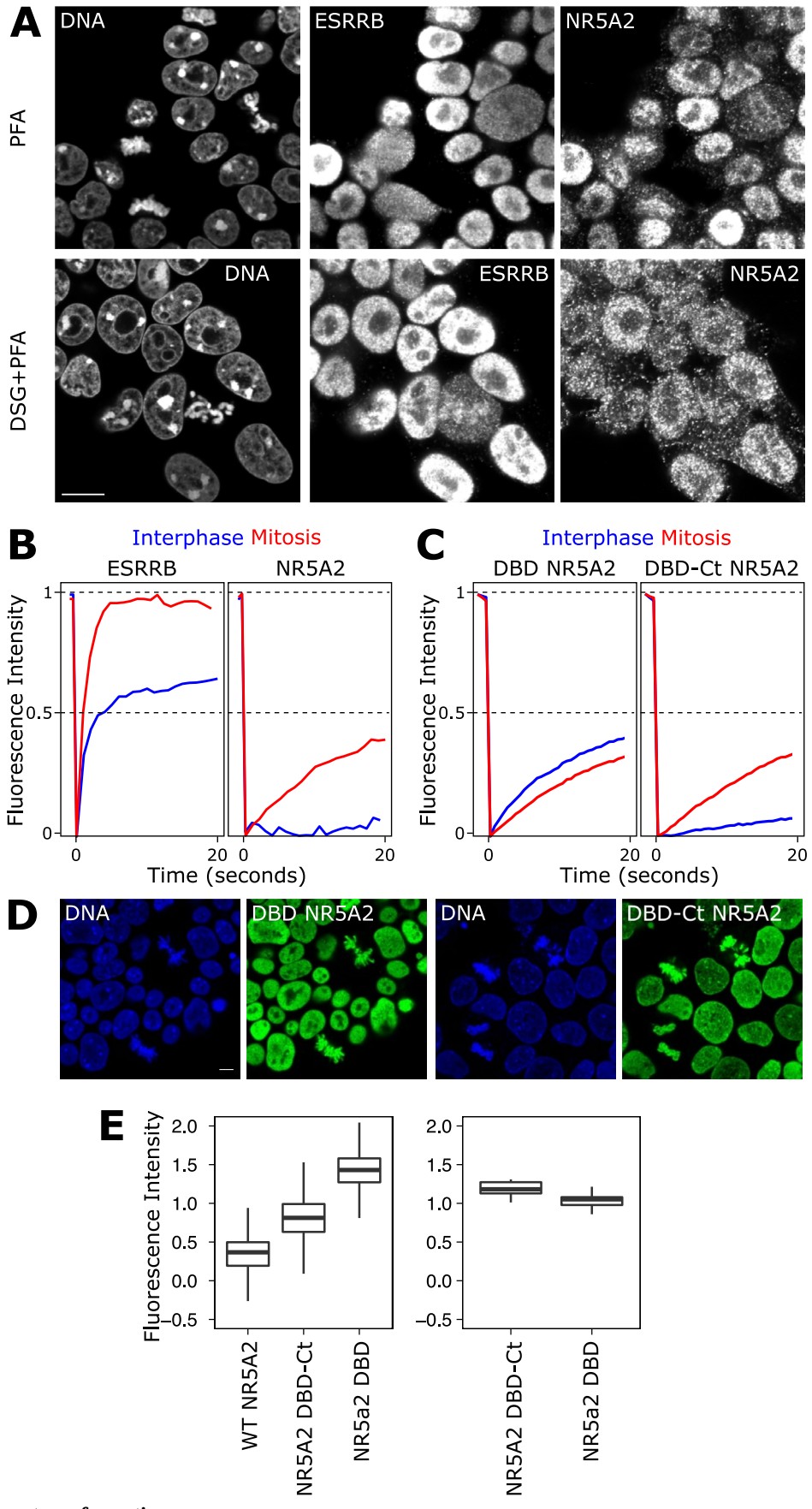

**Extended Data Fig. 2 | See next page for caption.**

**Extended Data Fig. 2 | NR5A2 long-lived chromatin interactions are driven by its DNA binding domain.** (A) Illustrative example of ESRRB/NR5A2 immunostaining upon formaldehyde or DSG and formaldehyde crosslinking, as indicated. The white line represents 10 μm in all panels. (B) Quantification of comparative FRAP assays of ESRRB-GFP and NR5A2-GFP expressed from the endogenous loci. Similar observations have been systematically observed over more than 3 independent cultures and experiments. (C) Quantification of comparative FRAP assays of GFP fusions with the DNA binding domain of NR5A2 alone (DBD NR5A2) or with its C-terminal moiety containing the ligand binding domain (DBC-Ct). Both constructs were randomly integrated in ES cells. (D) Representative live imaging of the DBD or the DBD-Ct fused to GFP.

Identical results were systematically observed over more than 3 independent cultures. The white line represents 10 μm in all panels. (E) Expression of NR5A2-GFP analyzed by FACS for the three versions analyzed (left panel; WT expressed from endogenous locus (22,456 cells); DBD (13,804 cells) and DBD-Ct (28,004 cells) ectopically overexpressed). For the FRAP assays, cells with low DBD and high DBD-Ct expression were selected, as illustrated by quantifying the expression in the individual cells subject to FRAP (right). Representative data of a minimum of 2 experiments is shown. Boxplots represent the median as the horizontal bar, 25-75% percentiles as the box and 1.5-folds the inter-quartile range as whiskers measured for.

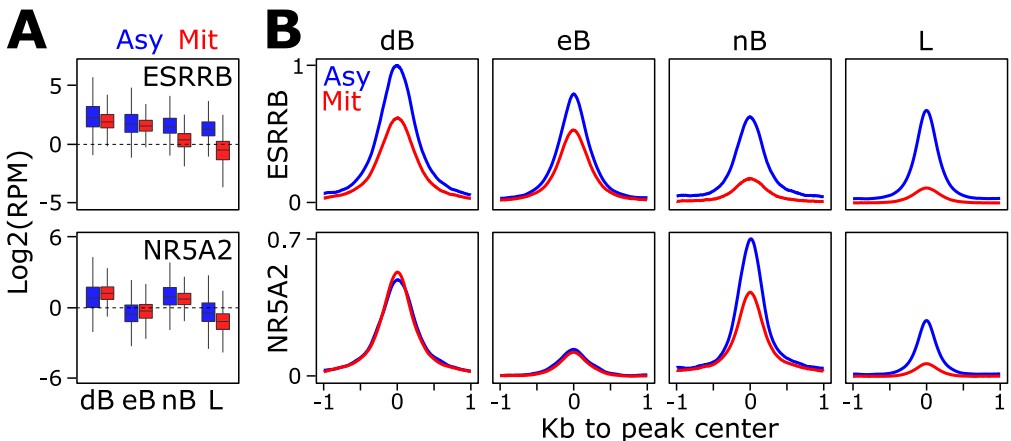

**Extended Data Fig. 3 | Comparison of ESRRB/NR5A2 binding levels and motifs matching the identified consensus.** (A-B) Enrichment levels (Reads per million) of ESRRB and NR5A2 across the four identified clusters (dB, eB, nB, L), represented as boxplots (A) or metaplots (B). All boxplots represent the median as the horizontal bar, 25-75% percentiles as the box and 1.5-folds the inter-quartile range as whiskers. All ESRRB and NR5A2 quantifications in this figure were obtained by averaging 2 (ESRRB) and 3 (NR5A2) independent replicates in interphase and in mitosis.

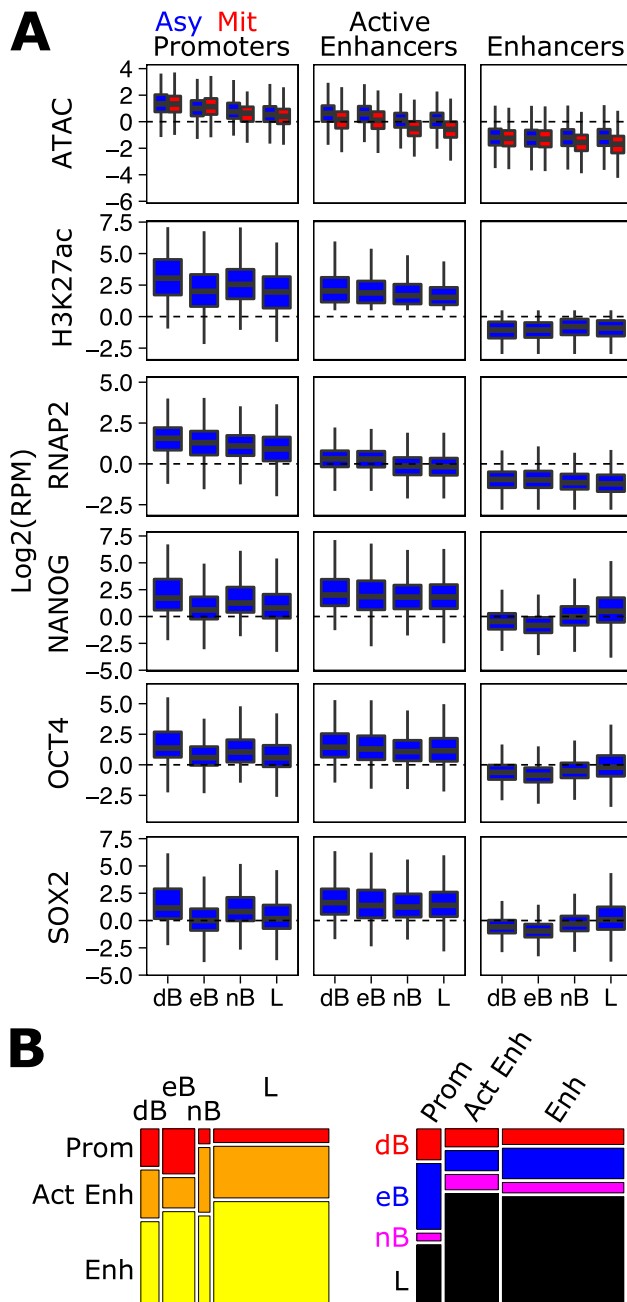

**Extended Data Fig. 4 | Relationships between ESRRB/NR5A2 binding and other chromatin properties at distinct gene regulatory elements.** (A) Enrichment levels of the indicated factors across dB, eB, nB and L regions separated as promoters, active enhancers or enhancers. All boxplots represent the median as the horizontal bar, 25-75% percentiles as the box and 1.5-folds the inter-quartile range as whiskers. A minimum of 2 independent datasets from Encode (H3K27ac and RNAP2) or from published datasets[10] were used (ATAC, OCT4, SOX2, NANOG). (B) Mosaic plots describing the relative proportions of promoters, active enhancers and enhancers across dB, eB, nB and L regions (left) and vice-versa (right).

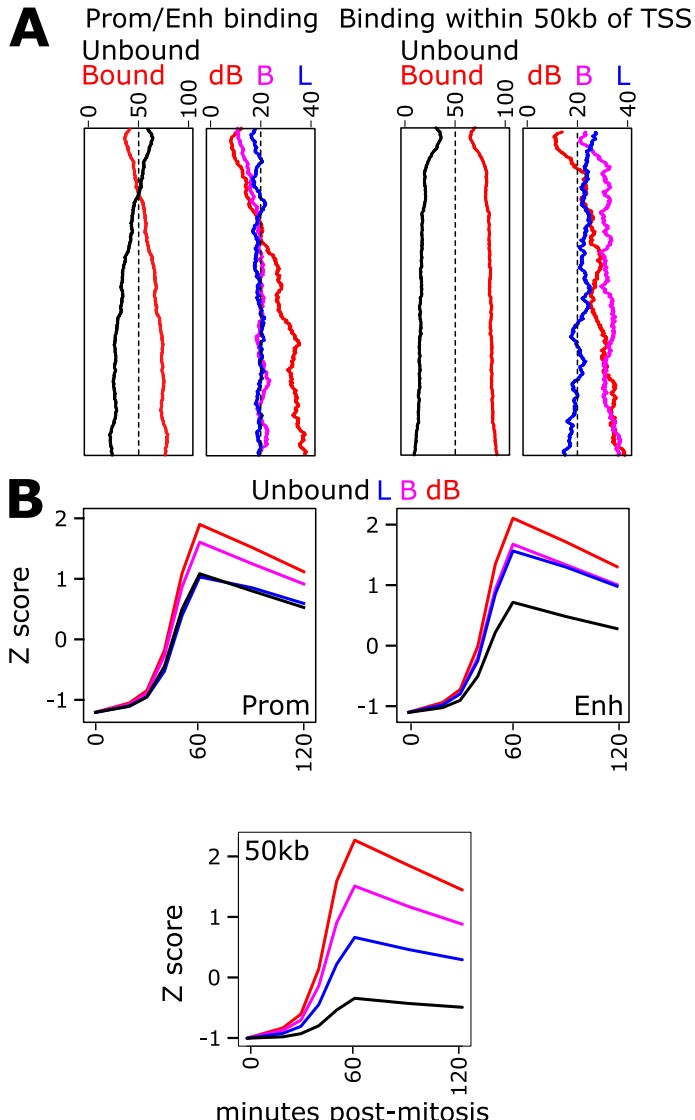

Extended Data Fig. 5 | Additional correlations between ESRRB/NR5A2 and post-mitotic gene transcription. (A) Comparison of the proportion of genes identified as dB, eB, nB and L regions for groups of 1000 genes sliding from top to bottom of the heatmap shown in (Fig. 5A), with a step of 10 genes, when the gene-region association is based on known enhancers and promoters (left, identical to Fig. 5B for comparison purposes) or when it is based on the presence of a binding regions within 50 kb of the promoter (right), as previously done[9]. (B) Post-mitotic gene transcription dynamics when the associations consider exclusively promoters (Prom, left), exclusively enhancers (Enh, right), or any peak located within 50 kb of the promoter (50 kb, bottom).

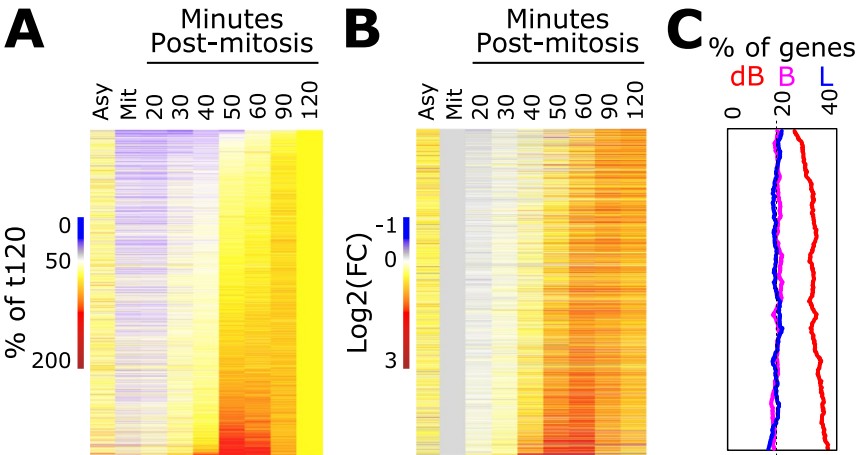

**Extended Data Fig. 6 | Analysis of transcription reactivation timing.**
(A) Heatmap displaying the percentage of expression at each time-point versus the end point, ordered by increasing reactivation (calculated by the mean of t30 to t90). (B) Heatmap showing the Log2(FC) post-mitosis, for genes ordered as in A, calculated as in Fig. 5B and ED5A. (C) Correlation between ESRRB/NR5A2 binding categories and gene reactivation as evaluated by the % of t120, computed and presented as in Fig. 5B.

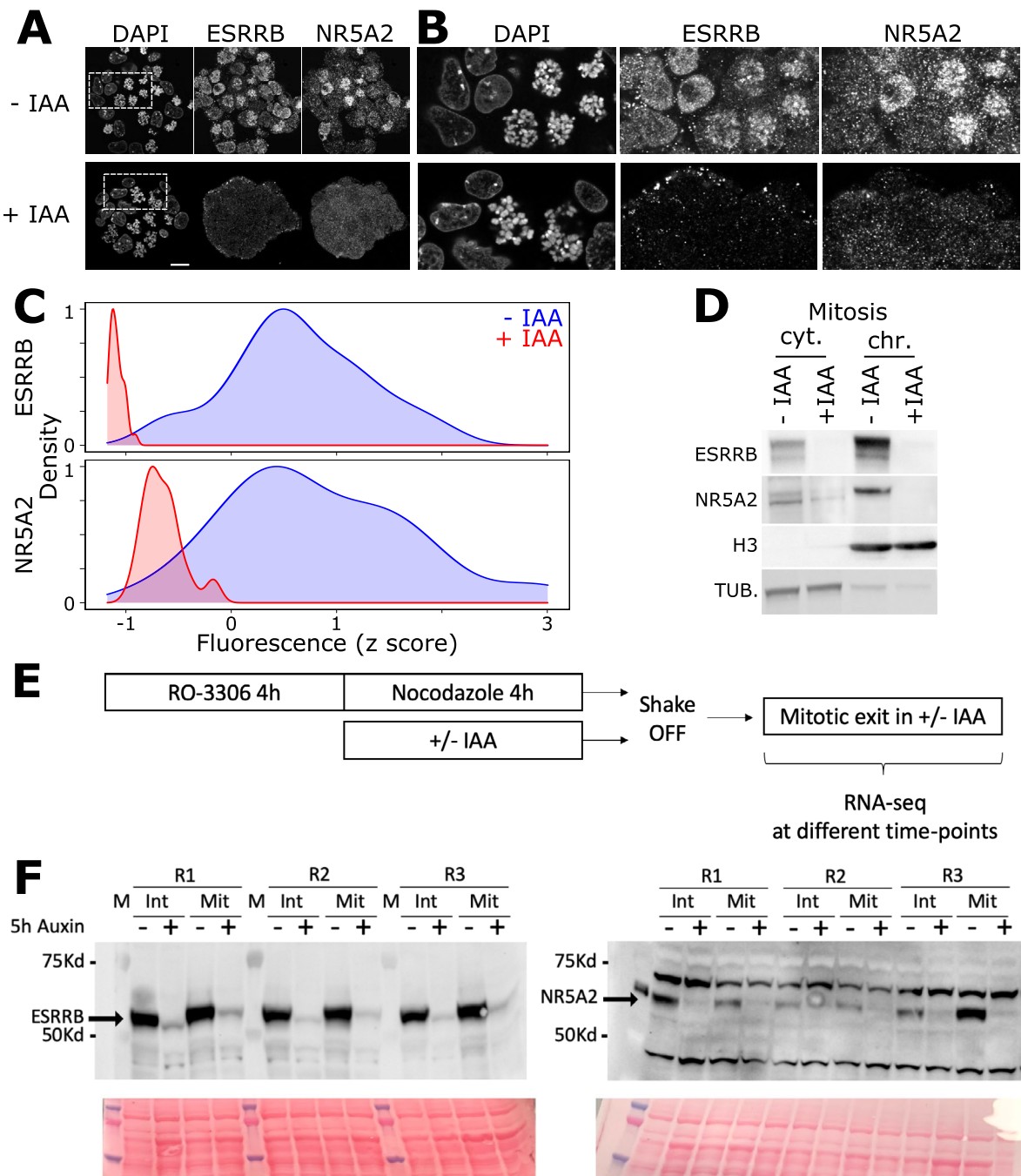

**Extended Data Fig. 7 | Double Auxin depletion system.** (A,B) Immuno-fluorescence of ESRRB and NR5A2 in control (top) and IAA-treated cells (bottom). The rectangle denotes a region selected for a zoom view in (B). The horizontal line represents 10 μm. Identical observation were for all replicates used (n = 3). (C) Quantification of ESRRB and NR5A2 immuno-fluorescence in mitotic chromatin. Note NR5A2 staining produces more background than ESRRB upon IAA treatment. (D) Western-blot analysis of ESRRB/NR5A2 depletion upon IAA treatment in mitotic cells, after fractionation of the cytoplasm (cyt.) and the chromatin (chr.). Histone H3 and TUBULINE (TUB.) were used as controls of the purity of the fractionation. Similar results were obtained in two independent preparations. (E) Schematic representation of the protocol to deplete ESRRB/NR5A2 in mitosis and during the M-G1 transition. (F) Western-blot analysis of ESRRB/NR5A2 in interphase and in mitosis in the three replicates used for RNA-seq analysis following the experimental scheme shown in (D). Ponceau stainings are shown as a loading control.

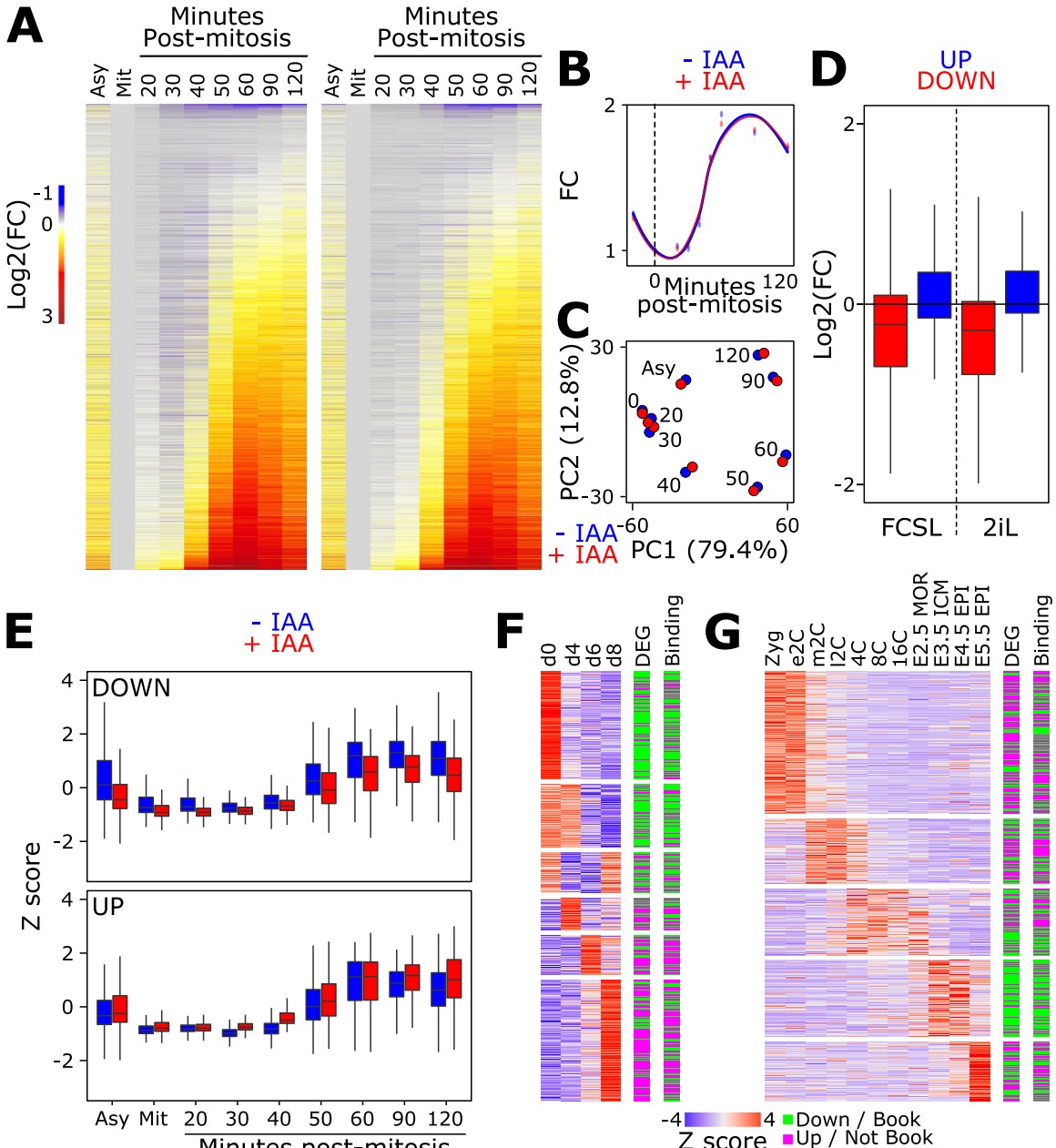

**Extended Data Fig. 8 | Regulation of selected genes by ESRRB/NR5A2 during the M-G1 transition.** (A) Reactivation heatmap presented as in Fig. 5A but in control (left) or ESRRB/NR5A2-depleted cells (right). (B) Global gene reactivation profile of all genes shown in (A) in presence (-IAA) or absence (+IAA) of ESRRB/NR5A2. (C) PCA analysis of post-mitotic transcription dynamics in presence (-IAA) or absence (+IAA) of ESRRB/NR5A2. (D) Fold-change of expression levels upon ESRRB/NR5A2 knock-out in two different media[14] (FCSL or 2iL) for the genes extracted from PC5/PC6 loadings as shown in Fig. 6A, B. (E) Transcription dynamics of an extended list of genes downregulated (top panel) or upregulated

(bottom panel) upon ESRRB/NR5A2 depletion during the M-G1 transition; 3 independent experiments for each time-point and condition were averaged. (F) Expression profile of the two groups of genes shown in (E) during embryoid bodies differentiation, organized by timing of maximal expression, with the identification of the groups they belong and their mitotic bookmarking status shown on the right. (G) Identical analysis to (F) but during early mouse embryogenesis. All boxplots in this figure represent the median as the horizontal bar, 25-75% percentiles as the box and 1.5-folds the inter-quartile range as whiskers.

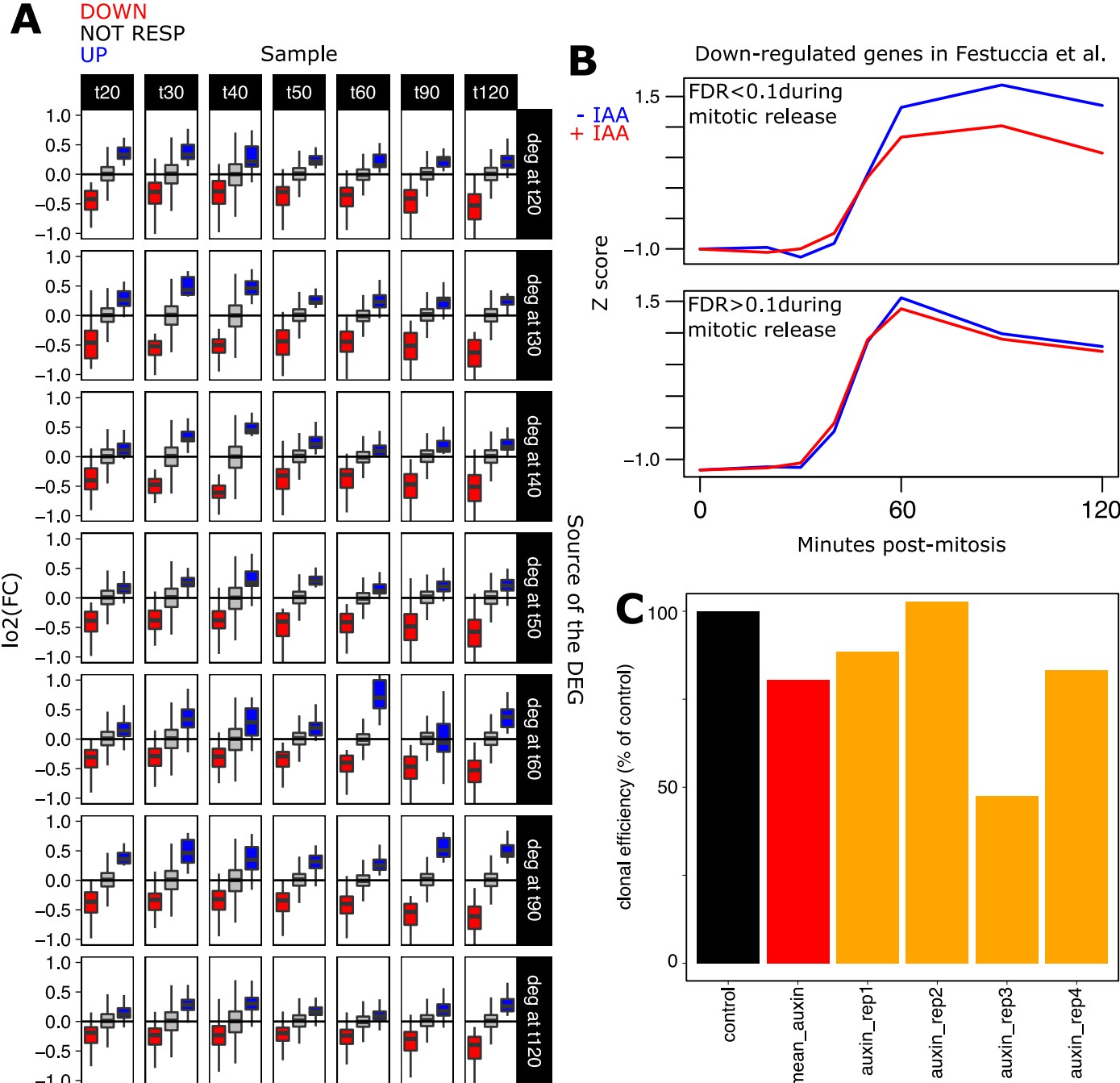

**Extended Data Fig. 9 | Additional analyses of ESRRB/NR5A2-depleted cells.** (A) Boxplots depicting log2FC expression upon IAA treatment for genes identified as statistically significant (FDR < 0.1) at each timepoint (organized in rows of boxplots) and plotted across all timepoints (organized in columns). All boxplots represent the median as the horizontal bar, 25-75% percentiles as the box and 1.5-folds the inter-quartile range as whiskers; 3 independent experiments for each time-point and condition were averaged. (B) Previous genes known to change expression in ESRRB and NR5A2 KO ES cells[14] were separated by their statistical response to IAA treatment after mitosis. (C) Cells were arrested in

mitosis in the presence or absence of IAA as shown in Fig.S7D. Subsequently they were shaken off, counted and plated at clonal density in the presence/ absence of IAA for 2 h, covering the length of G1. Next, IAA was withdrawn and cells were cultured for 7 days, when Alkaline Phosphatase staining was used to count the number of undifferentiated ES cell colonies. The results are shown as a % of clonogenicity in +IAA versus -IAA. The assay is subject to high variability due to intrinsic cell manipulations and counting but, overall, shows a reduction in clonogenicity upon the depletion of ESRRB/NR5A2 during a single mitosis/ M-G1 transition.

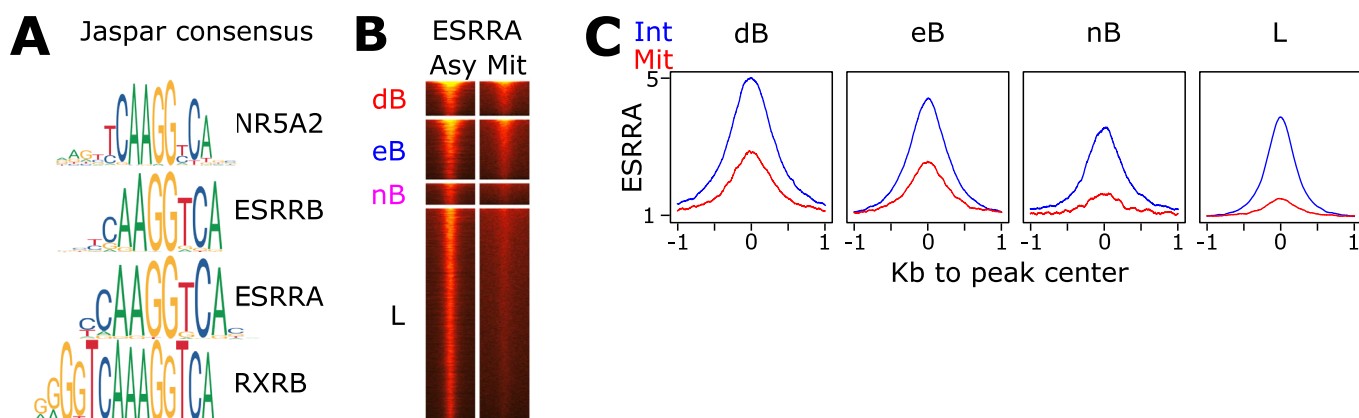

**Extended Data Fig. 10 | ESRRA is a primary candidate conferring additional redundancy to ESRRB/NR5A2 bookmarking.** (A) Correspondence of the motifs identified in our datasets with known motifs present in the Jaspar database. The short motif carrying a T at the 7th position, found in dB and eB regions, corresponds both to ESRRB and ESRRA. (B-C) Binding of ESRRA in asynchronous and mitotic ES cells across the 4 ESRRB/NR5A2 clusters presented as a heatmap (B) or as an average profile (C). The results are perfectly aligned with the presence of ESRRA consensus motifs in dB and eB and much less in nB and L, and strongly suggest that ESRRA is an additional mitotic bookmarking nuclear receptor acting conjunctly with ESRRB and NR5A2.

| | |
|---|---|

# Reporting Summary

Please do not complete any field with "not applicable" or n/a.  Refer to the help text for what text to use if an item is not relevant to your study.
For final submission: please carefully check your responses for accuracy; you will not be able to make changes later.

## Statistics

For all statistical analyses, confirm that the following items are present in the figure legend, table legend, main text, or Methods section.

| n/a | Confirmed | |
|---|---|---|
| ☐ | ☑ | The exact sample size (*n*) for each experimental group/condition, given as a discrete number and unit of measurement |
| ☐ | ☑ | A statement on whether measurements were taken from distinct samples or whether the same sample was measured repeatedly |
| ☐ | ☑ | The statistical test(s) used AND whether they are one- or two-sided<br>*Only common tests should be described solely by name; describe more complex techniques in the Methods section.* |
| ☑ | ☐ | A description of all covariates tested |
| ☐ | ☑ | A description of any assumptions or corrections, such as tests of normality and adjustment for multiple comparisons |
| ☐ | ☑ | A full description of the statistical parameters including central tendency (e.g. means) or other basic estimates (e.g. regression coefficient) AND variation (e.g. standard deviation) or associated estimates of uncertainty (e.g. confidence intervals) |
| ☐ | ☑ | For null hypothesis testing, the test statistic (e.g. *F*, *t*, *r*) with confidence intervals, effect sizes, degrees of freedom and *P* value noted<br>*Give P values as exact values whenever suitable.* |
| ☑ | ☐ | For Bayesian analysis, information on the choice of priors and Markov chain Monte Carlo settings |
| ☑ | ☐ | For hierarchical and complex designs, identification of the appropriate level for tests and full reporting of outcomes |
| ☑ | ☐ | Estimates of effect sizes (e.g. Cohen's *d*, Pearson's *r*), indicating how they were calculated |

*Our web collection on statistics for biologists contains articles on many of the points above.*

## Software and code

Policy information about availability of computer code

| Data collection | all data was generated internally or gathered from previous publications and public repositories, as indicated in the manuscript |
|---|---|
| Data analysis | all analyses were performed with standard packages (R version 3.6.3, bamsignals package, ggplot, DEseq2, ComplexHeatmaps, Bowtie2, MACs2, RSEM, STAR) |

For manuscripts utilizing custom algorithms or software that are central to the research but not yet described in published literature, software must be made available to editors and reviewers. We strongly encourage code deposition in a community repository (e.g. GitHub). See the Nature Portfolio guidelines for submitting code & software for further information.

## Data

Policy information about availability of data

All manuscripts must include a data availability statement. This statement should provide the following information, where applicable:
- Accession codes, unique identifiers, or web links for publicly available datasets
- A description of any restrictions on data availability
- For clinical datasets or third party data, please ensure that the statement adheres to our policy

The mass spectrometry proteomics data have been deposited to the ProteomeXchange Consortium via the PRIDE partner repository (PXD038950). NR5A2 ChIP-seq and RNA-seq of ESRRB/NR5A2-AID cells exiting mitosis are available at Gene Expression Omnibus (GSE220253).

## Research involving human participants, their data, or biological material

Policy information about studies with human participants or human data. See also policy information about sex, gender (identity/presentation), and sexual orientation and race, ethnicity and racism.

| | |
|---|---|
| Reporting on sex and gender | N/A |
| Reporting on race, ethnicity, or other socially relevant groupings | N/A |
| Population characteristics | N/A |
| Recruitment | N/A |
| Ethics oversight | N/A |

Note that full information on the approval of the study protocol must also be provided in the manuscript.

# Field-specific reporting

Please select the one below that is the best fit for your research. If you are not sure, read the appropriate sections before making your selection.

☑ Life sciences  ☐ Behavioural & social sciences  ☐ Ecological, evolutionary & environmental sciences

For a reference copy of the document with all sections, see [nature.com/documents/nr-reporting-summary-flat.pdf](http://nature.com/documents/nr-reporting-summary-flat.pdf)

# Life sciences study design

All studies must disclose on these points even when the disclosure is negative.

| | |
|---|---|
| Sample size | The number of replicates used in this study are standard in the field |
| Data exclusions | no dataset was arbitrarily excluded |
| Replication | All replicates are described in the methods or the legends of the figures |
| Randomization | In this study, randomization was not deemed necessary due to the implementation of orthogonal methods (eg FRAP, ChIP, MNase), strategically chosen to consolidate key results and control for potential confounding variables |
| Blinding | All sequencing data was analysed and all proteomic data was generated without prior knowledge of the biological nature of the groups. |

# Behavioural & social sciences study design

All studies must disclose on these points even when the disclosure is negative.

| | |
|---|---|
| Study description | |
| Research sample | |
| Sampling strategy | |
| Data collection | |
| Timing | |
| Data exclusions | |
| Non-participation | |
| Randomization | |

# Ecological, evolutionary & environmental sciences study design

All studies must disclose on these points even when the disclosure is negative.

| | |
|---|---|
| Study description | |
| Research sample | |
| Sampling strategy | |
| Data collection | |
| Timing and spatial scale | |
| Data exclusions | |
| Reproducibility | |
| Randomization | |
| Blinding | |

Did the study involve field work? ☐ Yes ☐ No

## Field work, collection and transport

| | |
|---|---|
| Field conditions | |
| Location | |
| Access & import/export | |
| Disturbance | |

# Reporting for specific materials, systems and methods

We require information from authors about some types of materials, experimental systems and methods used in many studies. Here, indicate whether each material, system or method listed is relevant to your study. If you are not sure if a list item applies to your research, read the appropriate section before selecting a response.

## Materials & experimental systems

| n/a | Involved in the study |
|---|---|
| ☐ | ☑ Antibodies |
| ☐ | ☑ Eukaryotic cell lines |
| ☑ | ☐ Palaeontology and archaeology |
| ☑ | ☐ Animals and other organisms |
| ☑ | ☐ Clinical data |
| ☑ | ☐ Dual use research of concern |
| ☑ | ☐ Plants |

## Methods

| n/a | Involved in the study |
|---|---|
| ☐ | ☑ ChIP-seq |
| ☑ | ☐ Flow cytometry |
| ☑ | ☐ MRI-based neuroimaging |

## Antibodies

| | |
|---|---|
| Antibodies used | 2 µg/ml mouse monoclonal anti-flag for ESRRB (M2 Sigma, F3165) 1µg/ml polyclonal rabbit anti-HA for NR5A2 (Abcam, ab9110) |
| Validation | highly validated antibodies against classical tags (flag and ha) further validated with the data provided in the manuscript using Auxin-inducible protein degradation of Esrrb and Nr5a2 |

# Eukaryotic cell lines

Policy information about cell lines and Sex and Gender in Research

| | |
|---|---|
| Cell line source(s) | All cell lines were derived from E14Tg2A, a standard cell line in ES research, originally obtained from Ian Chambers and Phil Avner |
| Authentication | All cell lines were tested by genotyping |
| Mycoplasma contamination | All cell lines were tested negative by PCR |
| Commonly misidentified lines (See ICLAC register) | none |

# Palaeontology and Archaeology

| | |
|---|---|
| Specimen provenance | |
| Specimen deposition | |
| Dating methods | |

☐ Tick this box to confirm that the raw and calibrated dates are available in the paper or in Supplementary Information.

| | |
|---|---|
| Ethics oversight | |

Note that full information on the approval of the study protocol must also be provided in the manuscript.

# Animals and other research organisms

Policy information about studies involving animals; ARRIVE guidelines recommended for reporting animal research, and Sex and Gender in Research

| | |
|---|---|
| Laboratory animals | |
| Wild animals | |
| Reporting on sex | |
| Field-collected samples | |
| Ethics oversight | |

Note that full information on the approval of the study protocol must also be provided in the manuscript.

# Clinical data

Policy information about clinical studies

All manuscripts should comply with the ICMJE guidelines for publication of clinical research and a completed CONSORT checklist must be included with all submissions.

| | |
|---|---|
| Clinical trial registration | |
| Study protocol | |
| Data collection | |
| Outcomes | |

# Dual use research of concern

Policy information about dual use research of concern

## Hazards

Could the accidental, deliberate or reckless misuse of agents or technologies generated in the work, or the application of information presented in the manuscript, pose a threat to:

| No | Yes | |
|----|----|----|
| ☑ | ☐ | Public health |
| ☑ | ☐ | National security |
| ☑ | ☐ | Crops and/or livestock |
| ☑ | ☐ | Ecosystems |
| ☑ | ☐ | Any other significant area |

## Experiments of concern

Does the work involve any of these experiments of concern:

| No | Yes | |
|----|----|----|
| ☑ | ☐ | Demonstrate how to render a vaccine ineffective |
| ☑ | ☐ | Confer resistance to therapeutically useful antibiotics or antiviral agents |
| ☑ | ☐ | Enhance the virulence of a pathogen or render a nonpathogen virulent |
| ☑ | ☐ | Increase transmissibility of a pathogen |
| ☑ | ☐ | Alter the host range of a pathogen |
| ☑ | ☐ | Enable evasion of diagnostic/detection modalities |
| ☑ | ☐ | Enable the weaponization of a biological agent or toxin |
| ☑ | ☐ | Any other potentially harmful combination of experiments and agents |

# Plants

| | |
|----|----|
| Seed stocks | |
| Novel plant genotypes | |
| Authentication | |

# ChIP-seq

## Data deposition

☑ Confirm that both raw and final processed data have been deposited in a public database such as GEO.

☑ Confirm that you have deposited or provided access to graph files (e.g. BED files) for the called peaks.

| Data access links | https://www.ncbi.nlm.nih.gov/geo/query/acc.cgi?acc=GSE220253 |
|----|----|
| *May remain private before publication.* | |
| Files in database submission | all raw sequencing reads and a table with regions and quantifications |
| Genome browser session (e.g. UCSC) | none |

## Methodology

| Replicates | 3 independent replicates per condition |
|----|----|
| Sequencing depth | all libraries have between 65,600,130 and 75,582,902 aligned fragments |
| Antibodies | M2 Sigma, F3165 |
| Peak calling parameters | standard MACS2 followed by replicate reproduction and statistical tests (generalized linear model), as described in the methods |
| Data quality | excellent, assessed by fastqc and reproducibility of published data |
| Software | none developed for this study (Bowtie2, MACS2, bamsignals, R3.6.3) |

# Flow Cytometry

## Plots

Confirm that:

☐ The axis labels state the marker and fluorochrome used (e.g. CD4-FITC).

☐ The axis scales are clearly visible. Include numbers along axes only for bottom left plot of group (a 'group' is an analysis of identical markers).

☐ All plots are contour plots with outliers or pseudocolor plots.

☐ A numerical value for number of cells or percentage (with statistics) is provided.

## Methodology

| | |
|---|---|
| Sample preparation | |
| Instrument | |
| Software | |
| Cell population abundance | |
| Gating strategy | |

☐ Tick this box to confirm that a figure exemplifying the gating strategy is provided in the Supplementary Information.

# Magnetic resonance imaging

## Experimental design

| | |
|---|---|
| Design type | |
| Design specifications | |
| Behavioral performance measures | |

| | |
|---|---|
| Imaging type(s) | |
| Field strength | |
| Sequence & imaging parameters | |
| Area of acquisition | |

Diffusion MRI    ☐ Used    ☐ Not used

## Preprocessing

| | |
|---|---|
| Preprocessing software | |
| Normalization | |
| Normalization template | |
| Noise and artifact removal | |
| Volume censoring | |

## Statistical modeling & inference

| | |
|---|---|
| Model type and settings | |
| Effect(s) tested | |

Specify type of analysis:    ☐ Whole brain    ☐ ROI-based    ☐ Both

Statistic type for inference

(See Eklund et al. 2016)

Correction

## Models & analysis

| n/a | Involved in the study |
|-----|----------------------|
| ☐ ☐ | Functional and/or effective connectivity |
| ☐ ☐ | Graph analysis |
| ☐ ☐ | Multivariate modeling or predictive analysis |

Functional and/or effective connectivity

Graph analysis

Multivariate modeling and predictive analysis

April 2023

