## [Peer Review File · Nature Structural & Molecular Biology]

Peer Review Information

Manuscript Title: Mitotic bookmarking redundancy by nuclear receptors in pluripotent cells

Corresponding author name(s): Nicola Festuccia, Pablo Navarro

Reviewer Comments & Decisions:

Decision Letter, initial version:

Message: 9th Feb 2023

Dear Dr. Navarro,

Thank you again for submitting your manuscript "Mitotic bookmarking redundancy by nuclear receptors mediates robust post-mitotic reactivation of the pluripotency network". I apologise again for the delay in responding, which resulted from the difficulty in obtaining suitable referee reports. Nevertheless, we now have comments (below) from the 3 reviewers who evaluated your paper. In light of those reports, we remain interested in your study and would like to see your response to the comments of the referees, in the form of a revised manuscript.

You will see that although Reviewers #2 (R#2) and #3 (R#3) appreciate the robustness and elegance of the cellular systems used, as well as the potential implications of these findings, Reviewer #1 (R#1) raises substantial concerns with respect to the advance offered in this manuscript vis-à-vis existing literature. More specifically, both R#1 and R#2 are uncertain if the observed phenotypes upon ESSRB and/or NR5A2 depletion arise due to actual mitotic bookmarking or other G1-related effects. The editorial team agrees that experimentally addressing whether the observed phenotypes arise due to specific mitotic effects, rather than G1-related effects, will be important for continuing to consider this work for publication in Nature Structural & Molecular Biology. To this end, investigating the effects of ESSRB and/or NR5A2 depletion with auxin wash-out experiments upon mitotic entry, as R#1 suggests (point 1) and R#2 alludes to, or other alternative experimental avenues that you might prefer to pursue, to address this concern, will be very useful. In addition to providing such data, we request that you address all technical concerns of the reviewers (for example points 3-6, 11, 12, 14 of R#1 and R#2), disambiguate any unclearly presented or overstated data, in accordance to the extensive guidance of R#2 but also R#1 (points 7-8, 15-18).

We are committed to providing a fair and constructive peer-review process. Do not hesitate to contact us if there are specific requests from the reviewers that you believe are

technically impossible or unlikely to yield a meaningful outcome.

We expect to see your revised manuscript within 3 months. If you cannot send it within this time, please contact us to discuss an extension; we would still consider your revision, provided that no similar work has been accepted for publication at NSMB or published elsewhere.

Reporting Summary:

When submitting the revised version of your manuscript, please pay close attention to our [href="https://www.nature.com/nature-portfolio/editorial-policies/image-integrity">Digital Image Integrity Guidelines. and to the following points below:](https://www.nature.com/nature-portfolio/editorial-policies/image-integrity)

Please note that all key data shown in the main figures as cropped gels or blots should be presented in uncropped form, with molecular weight markers. These data can be aggregated into a single supplementary figure item. While these data can be displayed in a relatively informal style, they must refer back to the relevant figures. These data should be submitted with the final revision, as source data, prior to acceptance, but you may want to start putting it together at this point.

SOURCE DATA: we urge authors to provide, in tabular form, the data underlying the graphical representations used in figures. This is to further increase transparency in data reporting, as detailed in this editorial (<http://www.nature.com/nsmb/journal/v22/n10/full/nsmb.3110.html>). Spreadsheets can be submitted in excel format. Only one (1) file per figure is permitted; thus, for multi-paneled figures, the source data for each panel should be clearly labeled in the Excel file; alternately the data can be provided as multiple, clearly labeled sheets in an Excel file. When submitting files, the title field should indicate which figure the source data pertains

to. We encourage our authors to provide source data at the revision stage, so that they are part of the peer-review process.

Data availability: this journal strongly supports public availability of data. All data used in accepted papers should be available via a public data repository, or alternatively, as Supplementary Information. If data can only be shared on request, please explain why in your Data Availability Statement, and also in the correspondence with your editor. Please note that for some data types, deposition in a public repository is mandatory - more information on our data deposition policies and available repositories can be found below: <https://www.nature.com/nature-research/editorial-policies/reporting-standards#availability-of-data>

[Redacted]

Sincerely,

Dimitris Typas

Associate Editor
Nature Structural & Molecular Biology
ORCID: 0000-0002-8737-1319

Referee expertise:

Referee #1: ESCs, pluripotency and chromatin organisation, NR5A2, HiC and computational genomics

Referee #2: Mitotic bookmarking, transcriptional regulation and TFs, ChIP-Seq and computational genomics, chromatin organisation

Referee #3: ChIP-MS, transcriptional regulation, ESCs and differentiation

Reviewers' Comments:

Reviewer #1:

Remarks to the Author:

This paper investigated mitotic bookmarking by the nuclear receptors ESRRB and NR5A2. Both transcription factors (TFs) bind to promoters and enhancers of active genes during mitosis in mouse embryonic stem (mES) cells. Auxin-mediated degradation of both TFs results in very small changes in transcriptional reactivation in a subset of genes in cells exiting mitosis. NR5A therefore follows in the footsteps of ESRRB (Festuccia et al., Nat Cell Biol 2016) and CTCF (Chervova et al., Embo rep 2023) as a mitotic bookmarking factor in ES cells.

The strengths of this manuscript are the use of genetically modified mES cells in which the TFs are tagged with fluorescent proteins for imaging or AID for auxin-mediated degradation. The work builds on a body of literature showing that several nuclear receptors bind to mitotic chromosomes (e.g. Saradhi et al., Biochimica et biophysica acta 2005; Kumar et al., Biochimica et biophysica acta 2008) and ESRRB in particular was shown to be a mitotic bookmarking TF by the Navarro lab several years ago (Festuccia et al., Nat Cell Biol 2016).

This manuscript has several weaknesses. First, the work contains hardly any novelty or conceptual advance towards a better understanding of whether mitotic bookmarking has an important role in regulating post-mitotic transcription. The contribution of ESRRB/NR5A2 to transcriptional reactivation after mitosis is so small that conventional principal component analysis cannot distinguish between the transcriptomes with and without their auxin-induced degradation; the analysis has to resort to PC5 and PC6, which has its own problems, and even then, the effects on pluripotency-associated genes are unconvincing. Secondly, and related to this, the work falls short of providing any evidence that mitotic bookmarking has a biological relevance beyond some minor attenuation of gene reactivation, i.e. is mitotic bookmarking by nuclear receptors required to maintain pluripotency? Third, the finding that NR5A2, like ESRRB, mitotically bookmarks chromatin is interesting for researchers in this field. Some differences in biophysical behavior are described for the two TFs but no further mechanistic insights are provided. In broader terms, the functioning of ESRRB and NR5A2 together is not novel since they also function together in maintaining mES pluripotency, as shown by the Navarro lab (Festuccia et al.,

Development 2021). ESRRB and NR5A2 are also not the only mitotic bookmarking factors with minor effects on gene reactivation in ES cells, as a few hundred genes are also attenuated by CTCF depletion, as shown recently by the Navarro lab (Chervova et al., EMBO reports 2023). Lastly, the manuscript suffers from a lack of clarity in the figures, legends and explanations, and the majority of the discussion is speculative.

Main comments:

1. The manuscript falls short of testing the functional relevance of the genes that are attenuated by loss of mitotic bookmarking. Is G1 phase transcription sufficient to maintain pluripotency? It would be important to perform an auxin washout experiment during release from mitosis and to test whether transcription by newly synthesized ESRRB and NR5A2 is sufficient to maintain pluripotency and mES transcription. If yes, then this would argue against a major functional role of mitotic bookmarking, at least for transcriptional regulation of pluripotency-associated genes.
2. The transcriptional changes with and without TF are so small that they instill little confidence in being biologically relevant (Fig 6). The authors have to resort to an analysis of PC5 and PC6 and should include a critical discussion of what are the caveats of looking at these as opposed to PC1 and PC2. Are any of the differences with and without auxin statistically significant for PC5/PC6 (Fig 6B)? Are any of the extremely small differences (note that the y-axis can be as little as 2.5 TPM) shown in Fig 6D statistically significant? The results of statistical analyses should be included in the legend.
3. The minor changes in transcription could be either because few genes are regulated by bookmarking or protein depletion is incomplete. To assess this, the authors need to show a dilution series to give an idea of how much ESRRB and NR5A2 are reduced by Western blotting. It appears that residual ESRRB is still present in both interphase and mitotic samples treated with auxin (Fig S6A). This Western blot also needs a loading control. For NR5A2, several additional bands are shown in the Western blot (Fig S6A). Are these cross-reacting bands? If so, how do the authors explain that the intensities of these additional bands differ between lanes?
4. The transcriptional effects of degradation of ESRRB/NR5A2 during mitosis are weak. One possibility is that the degradation of chromatin-bound TFs is less effective than unbound ones and that therefore the effects are so mild. To exclude this, calibrated ChIP of G1 phase cells after mitotic auxin treatment is needed to test whether the peaks are still detectable.
5. ChIP data are used to generate four groups of binding patterns that form the basis for further analysis. One concern is that ChIP can also result in peaks that are independent of the target proteins. To exclude these non-specific peaks, the authors should take advantage of their single KO ES cells as controls for interphase and mitotic ChIP experiments. The resulting data will be more robust and convincing.
6. It is not straightforward to get a clear sense of the upregulated and downregulated targets versus non-changing targets only in z-score plots. The RNA-seq results with significance thresholds indicated should be included. Using an unbiased analysis, how many genes and by what fold change are significantly downregulated following IAA treatment, without including previously identified targets?

7. As the authors explain in the introduction, ESRRB and NR5A2 function redundantly in the maintenance of mES pluripotency (Festuccia et al., Development 2021). Moreover, given the high level of redundancy of TFs of nuclear receptors and their structurally related DNA binding domains and similar DNA motifs, it is not unexpected that they would function redundantly in other scenarios as well. It is unclear why the abstract states that the authors “discover an unexpected redundancy among members of the protein superfamily of nuclear receptors” (abstract). This contradicts their own arguments in the introduction and goes against the substantial knowledge in the literature on these TFs. The abstract should therefore be reworded to reflect that this is an expected redundancy.

8. Some figures and legends are difficult to follow due to lack of details. For example, it is unclear in Fig 2D which TF is shown and how many cells were quantified. It is also unclear from the figure and legend whether C and D were performed with transiently expressed fusion constructs or genetically modified ES cells. Although some of this information can be deduced from the main text and the methods, legends should provide sufficient information (including biological replicates) to autonomously understand the data in the figure.

9. What is the rationale for fusing the DBD to the LBD, rather than using the whole protein, in assaying residence time? Which amino acids were left out in this construct and why? Does the full-length construct overexpressed by the same means recapitulate the low recovery rate in interphase?

10. How reproducible are the FRET results? Fig 1D and Fig S1B seem to show somewhat different results for NR5A2, with little recovery in interphase for S1B and some recovery in interphase for 1D at 20 s. Are there other variables between these experiments?

11. For all cells shown in Fig 2, negative controls for background fluorescence using wild-type ES cells should be included.

12. For the FRAP data, what is the positive control of a freely moving protein to show full recovery?

13. How comparable is the ectopic expression of DBD-, DBD-LBD-GFP fusions and endogenous proteins by Western blotting?

14. Please provide evidence of purity of mitotic ES cells and state how this was determined.

15. Please clarify whether binding profiles were obtained with wild-type ES cells or using tagged fusion TFs.

16. “We conclude that the presence of different versions of the ESRRB/NR5A2 motif... are direct determinants of the behavior of ESRRB and NR5A2 in mitosis” (203). The motifs were not mutated. Therefore, the binding and the motifs correlate but it cannot be concluded that the motifs are “direct determinants”, which overstates the data.

17. The discussion states that “we have identified a family of gene regulators, nuclear receptors, as potentially common mitotic bookmarking factors” (335). In fact, the paper shows that two nuclear receptors, of which ESRRB had been shown previously, bookmark chromatin in mitosis. The above claim would be more substantial if more than two factors,

which cooperate in pluripotency anyhow, are shown to have such bookmarking functions. It would be more honest to state that they have identified one more bookmarking factor, NR5A2.

18. The discussion section is long and largely speculative. It can be intellectually interesting to consider the broader implications but the balance between conclusions and speculations needs to be maintained. It is therefore advisable to compact the last two paragraphs into a few sentences, especially since the speculations regarding ligand-dependent nuclear receptors are not directly relevant to the orphan nuclear receptors investigated in the manuscript. The speculations on evolution of ligand-dependent nuclear receptors may be better placed into a perspectives or opinion piece.

Minor comments:

1. The ESRRB/NR5A2 responsive genes are stated to be "enriched for pluripotency regulators". How is enrichment quantified in the gene group? It might be more accurate to simply say "include pluripotency regulators".
2. A schematic of the DBD-LBD fusions would be helpful. It would appear that these lack other domains normally present in the wild-type protein. What was the rationale for excluding these?
3. All IF data should include a scale bar.
4. "Jumpstarting" is not a scientific term.
5. Biological "significance" (330) is a term that should be reserved for statistical analysis.

Reviewer #2:

Remarks to the Author:

This is a well written report on the effects of potential mitotic bookmarking of nuclear receptors on transcriptional reactivation in G1-phase. The study includes a very nice biochemical approach to identify potential mitotic chromatin associated proteins. The authors then follow up on two orphan nuclear receptors ESRRB and NR5A2 and study their retention on mitotic chromatin in relation to binding sites, chromatin context and requirement for post-mitotic gene activation.

The subject of post-mitotic genome reactivation should be of interest to readers in gene expression and epigenetics. The quality of the data is generally very good. My comments can be mostly addressed textually.

I believe the discussion could be organized in a manner that integrates some results better with regards to the main hypothesis. For example, are the effect of ESRRB and NR5A2 on nucleosome positioning or chromatin residence times causally linked to "bookmarking".

More generally, I think it is really difficult not just in this report but in the field in general to distinguish true bookmarking, i.e. the requirement specifically throughout mitosis of a transcription factor, from its requirement in G1. I do not want to hold the present report to higher standards than the field in general, which might entail experiments such as

shorter depletion of ESRRB and/or NR5A2 specifically in M-phase and not into G1. However, I think that a more honest title would be appropriate. For example, nuclear receptors cooperate during genome reactivation after mitosis, or something along those lines.

1. My major question relates to the proposed redundancy of ESRRB and NR5A2. Ideally, they would have included separate ESRRB and NR5A2 depletion experiments, especially if NR5A2 has not been studied as a potential bookmark before. Without additional experiments, they could at least compare their downregulated genes to known ESRRB targets and look at differences in binding patterns.
2. It may not be surprising that no pluripotency factors were detected in anti-ESRRB ChIP material. Was ChIP-MS carried out with an antibody against a pluripotency factor as point of reference? Also, simply from the literature, where MS had been carried out, have the ESRRB-associated factors been found?
3. "Co-localization" of NR5A2 and ESRRB cannot be appreciated in the images because "coating" is so broad.
4. The ESRRB-associated factors could be actually bound to ESRRB or simply reside in proximity on the DNA fragment that was part of the complex. That ESRRB and NR5A2 bind to chromatin independently of each other and have different residence times supports the latter. Have the authors considered DNase treatment in the IP to distinguish between these possibilities more globally?
5. Please clarify why ESRRB was not a focus, since it would be the prime candidate for overlapping functions with ESRRB.
6. For the FRAP experiments were the constructs expressed comparably?
7. There seem to be ChIP seq peaks that are unique to mitosis, or is this an issue of normalization and scaling. What are the characteristics of these "M-only" sites, motifs, chromatin features etc? How do M-only marked genes re-activate? M-only peaks might offer an opportunity to test M-phase binding specific effects.
8. In Fig.5 using fold-change over mitosis might be misleading given the low expression in mitosis. What would the results look like if gene activity is plotted as % of late G1? Also, it is not clear from panels B and C that reactivation of dB genes is faster. It might simply be a higher average level of expression of dB genes since they are bound by higher NR5A2 and ESRRB in interphase. I think that this is an important distinction and should be made clear in the text because it differentiates between actual bookmarking vs transcriptional activity.
9. Many genes spike in early G1, but this seems to be ignored in the discussion even though Fig.5A suggests spiking at a lot of genes. Please comment.
10. A point related to #7 is brought up in the discussion (line 390): "In accord with this view, the binding by TFs that do not act as mitotic bookmarking factors in our cells, such as OCT4, SOX2 and NANOG, is also associated with accelerated transcription patterns after mitosis in comparison with unbound genes" This is critical because it distinguishes between bookmarking vs transcription activation, and I think this idea should be introduced earlier to help the reader conceptualize what is being studied here. In this regard, the title of the paper seems a bit too strong. The abstract refers to the most "rapidly" activated genes as being bookmarked, but again, this is not fully supported here.
11. Unless I am mistaken, the same point could be made when considering gene reactivation upon IAA treatment in Fig.6B,D. The max is different (also in asynch cells) but the kinetics don't look different.
12. Fig.6: has it been verified by ChIP that NR5A2 and ESRRB are depleted from chromatin upon IAA treatment?
13. As to the IAA-upregulated genes, are they direct targets? Is there "repressive

bookmarking”?

14. The embryoid body experiment seems to be added as an afterthought. It is unclear to this reader how this experiment with heterogenous cells relates to mitotic bookmarking.

Minor:

1. The term “coating” mitotic chromosomes is not ideal since the factors in question occupy distinct regions.
2. A bit more info about NR5A2 in the intro would be helpful.
3. “the high occurrence of nuclear receptors in mitotic cells is compelling, representing half of the most confident mitotic hits” Perhaps rephrase, because the protein identification is biased with ESRRB as anchor.
4. This sentence needs to be made clearer: “Indeed, if at nB regions ESRRB is bound in interphase but mostly lost in mitosis, at eB regions NR5A2 is not found in mitosis simply because this TF is not efficiently recruited at these loci even in interphase”
5. I think Fig.3 D-H could be made supplemental.
6. Fig.5B: the annotation is confusing. Please clarify how the % genes are defined. In panel C, could actual mean expression levels be plotted.
7. Please add a cartoon of the timeline of G2 arrest release, nocodazole and IAA exposure. Where the cells sorted etc so the reader does not have to search the references.
8. Line 314; “with nearly 80% of downregulated genes being bound at known regulatory elements” It looks more like 70%
9. Line 423: “super-family of regulators may have a constitutive role”. What does constitutive mean here?

Reviewer #3:

Remarks to the Author:

Chervova et al investigate the redundancy of nuclear receptors Esrrb and Nr5a2 in the mitotic bookmarking of genes. Previously it was shown that the effect of individual factors was modest. Now the authors test the effect of rapidly depleting by degron both Esrrb and Nr5a2 at the same time. This dual effect turns out to be big; 1000 genes are not properly re-activated after mitosis, including many pluripotency genes. This shows that Esrrb and Nr5a2 together are highly important for proper reactivation of genes after mitosis in ES cells, which is a major finding, suitable for NSMB.

The authors start with an analysis by ChIP-MS which factors bind (near) Esrrb in asynchronous and mitotic cells. In asynchronous cells they find previously identified proteins but mitotic cells show an enrichment of nuclear receptors or their co-factors. Esrrb and Nr5a2 bind to mitotic chromosomes independently from each other (show by KO lines for one or other factor). FRAP experiments show that Nr5A2 has a much longer residence time than Esrrb, setting it apart as a stable mark during mitosis. The DNA binding domain of Nr5a2 is sufficient for this.

The authors then show that Esrrb and Nr5A2 bind often together on the genome in interphase and during mitosis and provide a rich annotation and characterization of Esrrb, Nr5A2 binding sites.

The authors then annotated genes as being bound in mitosis by Esrrb, Nr5A2, both or none and show that genes rapidly activated after mitosis enrich for binding sites of both Esrrb and Nr5A2.

The authors then depleted Esrrb and Nr5a2 simultaneously by degron and tested gene

reactivation in depleted cells. Although the decrease in reactivation is modest it could be shown that about a 1000 genes was affected, a much higher number than previously shown, and these genes were enriched for Esrrb, nr5a2 mitotic bindingsites and contained many pluripotency genes. Genes activated by Esrrb, nr5a2 mitotic marking were transiently upregulated during the naïve state of early embryogenesis.

The experiments are well performed, precise, information-rich, appropriate and the conclusions correspond to the results.

One minor comment. The color coding of Fig. 1B is unclear. High confidence hits are red but some categories of proteins have their own (variable) color and the light (beige?) color belonging to low confidence mitotic hits. So if proteins have their "own group's" color they are not mitotic hits and then there are low confidence they are beige and if high confidence they are red? If so, that would be good to indicate in the legend.

Author Rebuttal to Initial comments

Reviewer #1:

Remarks to the Author:

This paper investigated mitotic bookmarking by the nuclear receptors ESRRB and NR5A2. Both transcription factors (TFs) bind to promoters and enhancers of active genes during mitosis in mouse embryonic stem (mES) cells. Auxin-mediated degradation of both TFs results in very small changes in transcriptional reactivation in a subset of genes in cells exiting mitosis. NR5A therefore follows in the footsteps of ESRRB (Festuccia et al., Nat Cell Biol 2016) and CTCF (Chervova et al., Embo rep 2023) as a mitotic bookmarking factor in ES cells.

The strengths of this manuscript are the use of genetically modified mES cells in which the TFs are tagged with fluorescent proteins for imaging or AID for auxin-mediated degradation. The work builds on a body of literature showing that several nuclear receptors bind to mitotic chromosomes (e.g. Saradhi et al., Biochimica et biophysica acta 2005; Kumar et al., Biochimica et biophysica acta 2008) and ESRRB in particular was shown to be a mitotic bookmarking TF by the Navarro lab several years ago (Festuccia et al., Nat Cell Biol 2016).

This manuscript has several weaknesses. First, the work contains hardly any novelty or conceptual advance towards a better understanding of whether mitotic bookmarking has an important role in regulating post-mitotic transcription. The contribution of ESRRB/NR5A2 to transcriptional reactivation after mitosis is so small that conventional principal component analysis cannot distinguish between the transcriptomes with and without their auxin-induced degradation; the analysis has to resort to PC5 and PC6, which has its own problems, and even then, the effects on pluripotency-associated genes are unconvincing. Secondly, and related to this, the work falls short of providing any evidence that mitotic bookmarking has a biological relevance beyond some minor attenuation of gene reactivation, i.e. is mitotic bookmarking by nuclear receptors required to maintain pluripotency? Third, the finding that NR5A2, like ESRRB, mitotically bookmarks chromatin is interesting for researchers in this field. Some differences in biophysical behavior are described for the two TFs but no further mechanistic insights are provided. In broader terms, the functioning of ESRRB and NR5A2 together is not novel since they also function together in maintaining mES pluripotency, as shown by the Navarro lab (Festuccio et al., Development 2021). ESRRB and NR5A2 are also not the only mitotic bookmarking factors with minor effects on gene reactivation in ES cells, as a few hundred genes are also attenuated by CTCF depletion, as shown recently by the Navarro lab (Chervova et al., EMBO reports 2023). Lastly, the manuscript suffers

from a lack of clarity in the figures, legends and explanations, and the majority of the discussion is speculative.

Main comments:

1. The manuscript falls short of testing the functional relevance of the genes that are attenuated by loss of mitotic bookmarking. Is G1 phase transcription sufficient to maintain pluripotency? It would be important to perform an auxin washout experiment during release from mitosis and to test whether transcription by newly synthesized ESRRB and NR5A2 is sufficient to maintain pluripotency and mES transcription. If yes, then this would argue against a major functional role of mitotic bookmarking, at least for transcriptional regulation of pluripotency-associated genes.

We agree with Reviewer 1 that establishing a system that would allow to perform loss-of-function assays of *Esrrb/Nr5a2* exclusively in mitosis and not at all during early G1 is lacking in our paper. This is however a hardly addressable and in fact general caveat in the field, as noted by Reviewer 2. We have attempted the assay proposed by the reviewer of washing out auxin as cells are released from the mitotic block. Unfortunately, the expression level of both *Esrrb* and *Nr5a2* only modestly increases and remains very low during the first hour upon release: it is not easily visible by immunostaining (Fig.R1A) and the recovery is minor after 2h as evaluated by western-blot (Fig.R1B).

Fig.R1. (A) illustrative immunostaining of ESRRB and NR5A2 one hour after releasing control mitotic cells (top) or IAA-treated mitotic cells. In both cases the cells were released in the absence of IAA. Note the NR5A2 signal after the release is not nuclear and is not specific. (B) Western-blot of NR5A2 and ESRRB in control and IAA-treated mitotic cells (left) and after 30, 60 and 120' of releasing them in the absence of IAA. Loading corresponds to GAPDH.

This was to be expected because cells need to undergo a full cycle of new protein expression (from transcription to nuclear import). This will intrinsically take time and, moreover, *Nr5a2* transcription is itself attenuated upon Auxin treatment (as shown in the original ms), making the experiment even more challenging. We have also tried sophisticated approaches to address the reviewer request: to control the mitotic behavior of TFs with potentially increased time resolution, we aimed at establishing an anchor-

away system in our previously reported inducible *Esrrb*-Nr5a2 double KO ES cells (Festuccia, Development 2021). First, we found NLS mutations of *ESRRB* that only partially alter its nuclear localization (Fig.R2A). This was important to ensure *ESRRB* traffics to the cytoplasm and can interact with an “anchor”, consisting of tandem FKBP-F36V (Clackson et al, PNAS 1998) dimerization domains linked to a membrane bound peptide (LYN). Then, we fused *Esrrb* carrying the NLS mutation with an identical FKBP dimerization domain. When expressing the anchor and *Esrrb*-FKBP in the same cells, it is possible to enforce *Esrrb* cytoplasmic/membrane sequestration by treatment with a rapamycin analogue, AP20187, that induces dimerization of the two tandem FKBP domains, without interfering with endogenous proteins (Fig.R2B). The idea behind this strategy was to be able to sequester *ESRRB* away from the chromosomes during mitosis and rapidly control its capacity to enter the nucleus upon entry into G1. Unfortunately, we observed that both in the absence or presence of the Rapamycin analogue, *ESRRB* is unable to globally enrich on, or “coat”, mitotic chromosomes as it normally does (Fig.R2C), casting doubts on the functionality of these modified proteins in mitosis. Moreover, we noticed that restoring nuclear localization of *Esrrb*-FKBP in wash-out experiments took several hours, invalidating the immediate utility of this approach.

Fig.R2 (A) Live imaging of *ESRRB*-GFP carrying point mutations in the NLS to enable partial cytoplasmic localization. (B) The same construct as in (A) was further fused to the FKBP dimerization domain and the cells treated with Rapamycin to force its full delocalization from the nucleus. (C) Unfortunately, both in the absence/presence of Rapamycin, this *ESRRB* variant is unable to focally enrich on mitotic chromosomes.

While we believe this system has strong potential, particularly to study the M-G1 transition, setting it up appropriately will require us a large amount of time with no guarantee of success. In sum, we have made

all possible efforts to address the reviewer request but, unfortunately, the experiment is extremely challenging and we were not able to engineer a functional protein sequestration system in a time compatible with the revision process.

2. The transcriptional changes with and without TF are so small that they instill little confidence in being biologically relevant (Fig 6). The authors have to resort to an analysis of PC5 and PC6 and should include a critical discussion of what are the caveats of looking at these as opposed to PC1 and PC2. Are any of the differences with and without auxin statistically significant for PC5/PC6 (Fig 6B)? Are any of the extremely small differences (note that the y-axis can be as little as 2.5 TPM) shown in Fig 6D statistically significant? The results of statistical analyses should be included in the legend.

The Reviewer expresses doubts that the transcriptional changes observed have any biological relevance. To address this, we have opted to perform a functional assay whereby cells were depleted of *Esrrb/Nr5a2* during M/G1 with auxin, as done for our RNA-seq analyses, and subsequently reseeded in clonal conditions without auxin. One week after plating, we observe that the depletion of *Esrrb/Nr5a2* during a single M/G1 transition leads to reduced numbers of undifferentiated colonies. However, the results are very variable among our 4 independent replicates (Fig.R3), which is expected given the complexity of the manipulation involved. We have added this result as Fig.S9C and commented it page 12 lines 337-341.

*Fig.R3 Double Auxin *Esrrb/Nr5a2* cells were arrested in mitosis in the presence of Auxin and then released at clonal density in the presence of Auxin for 2h. They were then cultured for 7 days in the absence of Auxin and the number of Alkaline-Phosphatase colonies counted. In black controls not treated with Auxin; in red, the average clonal efficiency after Auxin; in orange the 4 individual replicates*

The referee questions the validity of using PC5/PC6 in our PCA analysis instead of PC1/PC2. There is a clear rationale behind our choice: the first principal components reflect the major and genome-wide changes occurring after mitosis, including a prominent burst of transcription activity, as we reported earlier (Chervova et al. Embo Reports 2022). On the face of this major burst that occurs for virtually every active gene and that captures most of the variance of the dataset, we were actually very pleased to be able to identify any principal component capturing the effects of *Esrrb/Nr5a2* depletion. We have now clarified this in the text page 11 lines 307-308. The referee also asks whether PC5/6 differences are statistically significant: when collectively addressed, all time points for genes that are Down – which represent the

focus of our message – have $p < 0.05$. This has now been introduced to the figure and its legend as well as mentioned in the text page 11, lines 312-314. Similarly, the referee also requests p values of the individual gene traces shown in Fig.6D. While our purpose was to show the general tendency of several pluripotency regulators regardless of their individual performance, we have now included this in the figure and legend as requested.

3. The minor changes in transcription could be either because few genes are regulated by bookmarking or protein depletion is incomplete. To assess this, the authors need to show a dilution series to give an idea of how much ESRRB and NR5A2 are reduced by Western blotting. It appears that residual ESRRB is still present in both interphase and mitotic samples treated with auxin (Fig S6A). This Western blot also needs a loading control. For NR5A2, several additional bands are shown in the Western blot (Fig S6A). Are these cross-reacting bands? If so, how do the authors explain that the intensities of these additional bands differ between lanes?

The referee suggests that our depletion may not be total, explaining the weak effects we observe. Although incomplete, Esrrb/Nr5a2 depletion is acute – quantifying this with dilution series does not seem appropriate as the levels are actually very low. Moreover, we acknowledge that our WB lacks a loading control. However, we feel it is important to show the WB of the samples that were actually used in RNA-seq, as we did in the original paper. Unfortunately, these samples are not any longer available. To meet the reviewer criticism, we now provide a Ponceau staining of the samples prepared for RNA-seq in our new Fig.S7E, showing similar protein loading in every lane. We would like to highlight that we show WB data of the 3 independent auxin treatments, all revealing similar levels of Esrrb/Nr5a2 depletion and, we believe, excluding potential issues due to sample loading that would invalidate our observations. Moreover, we also show in Fig.S7A-C additional immuno-staining (see point 4) and WB with a loading control (Fig.R1B), all supporting the efficiency of the double depletion system.

More importantly, in our opinion the weak effects we observe are largely due to the redundancy by other nuclear receptors, a key message of our work that is supported by our ChIP-MS approach. In particular, Esrra, as suggested by Reviewer 2, could be playing a key role too. Additional arguments in support of increased redundancy are now given by the analysis of Esrra ChIP-seq (see point 17). Invalidating all of these factors simultaneously is, we believe, excessively demanding. We have clarified this in the results: *“it remains possible that additional mitotic bookmarking by other nuclear receptors (Fig.1) partially*

compensates for the loss of ESRRB/NR5A2” (page 12 lines 341-343) and throughout the new discussion: “However, the consequences of the loss of ESRRB/NR5A2 during the M-G1 transition are, for most genes, relatively modest (...) These observations may be ascribed to three major phenomena. First, additional bookmarking factors may cooperate with ESRRB/NR5A2, as shown for ESRRB, or target independent regions (...) Although challenging, it will be important in the future to establish new tools to invalidate the activity of all nuclear receptors identified as mitotic ESRRB binding partners, such that the importance and redundancy of this process can be fully tested.”.

4. The transcriptional effects of degradation of ESRRB/NR5A2 during mitosis are weak. One possibility is that the degradation of chromatin-bound TFs is less effective than unbound ones and that therefore the effects are so mild. To exclude this, calibrated ChIP of G1 phase cells after mitotic auxin treatment is needed to test whether the peaks are still detectable.

In keeping with point 3, Reviewer 1 suggests that the small amount of remnant proteins could be chromatin bound and asks for a calibrated ChIP-seq to address the efficiency of Esrrb/Nr5a2 depletion from the chromatin. Indeed, only with a calibrated experiment where non-mouse chromatin is spiked-in can this assay be rigorously done. Performing this experiment is however not trivial, mainly due to antibody issues and species cross-reactivity – we note in particular that to ChIP NR5A2 we had to tag the endogenous locus with an epitope.

Fig.R4. Immunostaining of ESRRB/NR5A2 in the absence (top) of after Auxin treatment (bottom).

To address the Reviewer concern we are now providing IF analyses of the depletion showing that mitotic cells treated with auxin lack significant levels of Esrrb/Nr5a2 still bound to the chromosomes (Fig.R4). This has been added to Fig.S7A,B and is commented page 10 lines 290-291.

5. ChIP data are used to generate four groups of binding patterns that form the basis for further analysis. One concern is that ChIP can also result in peaks that are independent of the target proteins. To exclude

these non-specific peaks, the authors should take advantage of their single KO ES cells as controls for interphase and mitotic ChIP experiments. The resulting data will be more robust and convincing.

We have already analysed Esrrb ChIP-seq in single Esrrb KO cells and show the specificity of our assay (Festuccia et al, NCB 2016 and Development 2021). Moreover, we are using highly validated antibodies against Flag and HA, epitopes that were introduced in our lines, at the endogenous loci, particularly because good Nr5a2 antibodies are difficult to obtain. Also, we note that we find the Esrrb/Nr5a2 motif in a high percentage of the peaks we identify (more than 75% at bookmarked peaks), an observation that does not support the possibility that we are detecting a large number of non-specific peaks.

6. It is not straightforward to get a clear sense of the upregulated and downregulated targets versus non-changing targets only in z-score plots. The RNA-seq results with significance thresholds indicated should be included. Using an unbiased analysis, how many genes and by what fold change are significantly downregulated following IAA treatment, without including previously identified targets?

To meet the Reviewer's comment, we have now included a new supplemental figure (Fig.S9A, reproduced here as Fig.R5) where we present log₂ FC boxplots for each set of down or upregulated genes identified at every time point (FDR<0.1) and plot them across all timepoints. In this figure, as requested by the referee, no external target has been considered. This analysis shows that, while upregulated genes display a more variable behavior, regardless of the timepoint at which genes were called as downregulated, they remain so across all timepoints, with the fold change progressively decreasing from t40 onwards. The total number of genes with FDR<0.1 is 990 for downregulated and 237 for upregulated targets

Fig.R5. Fold Changes of DEGs (FDR<0.1) identified at each timepoint and calculated across the whole kinetic.

(out of the 1013 and 941, respectively, presented in the original paper after combining previous datasets describing Esrrb/Nr5a2 responsive genes). Thus, the vast majority of the genes downregulated after mitosis correspond to bona-fide responsive genes in our data. Yet, upregulated genes that are largely derived from previous work, do nevertheless display the expected tendency and we think it is important to show this, as we did initially. We thank the referee for asking this question that further supports the notion that the primary effect of mitotic Esrrb/Nr5a2 bookmarking is to activate genes, as we concluded in our original manuscript.

7. As the authors explain in the introduction, ESRRB and NR5A2 function redundantly in the maintenance of mES pluripotency (Festuccia et al., Development 2021). Moreover, given the high level of redundancy of TFs of nuclear receptors and their structurally related DNA binding domains and similar DNA motifs, it is not unexpected that they would function redundantly in other scenarios as well. It is unclear why the abstract states that the authors “discover an unexpected redundancy among members of the protein superfamily of nuclear receptors” (abstract). This contradicts their own arguments in the introduction and goes against the substantial knowledge in the literature on these TFs. The abstract should therefore be reworded to reflect that this is an expected redundancy.

While we disagree with the Reviewer and believe that finding a redundancy in mitotic bookmarking activity is surprising, given that the norm for a TF is to be evicted from its DNA binding sites during mitosis, we have removed the term “*unexpected*” from the abstract.

8. Some figures and legends are difficult to follow due to lack of details. For example, it is unclear in Fig 2D which TF is shown and how many cells were quantified. It is also unclear from the figure and legend whether C and D were performed with transiently expressed fusion constructs or genetically modified ES cells. Although some of this information can be deduced from the main text and the methods, legends should provide sufficient information (including biological replicates) to autonomously understand the data in the figure.

We apologize that some key information was indeed lacking from Fig.2. We have now provided, we hope, sufficient information.

9. What is the rationale for fusing the DBD to the LBD, rather than using the whole protein, in assaying

residence time? Which amino acids were left out in this construct and why? Does the full-length construct overexpressed by the same means recapitulate the low recovery rate in interphase?

We apologize if our description of the FRAP was not sufficiently clear. All the data shown in principal figure 2 and supplementary figure 2B were obtained using full-length Esrrb and Nr5a2 fusion proteins expressed from the endogenous loci. In contrast, for supplementary figure 2C we resorted to the constitutive transgenic expression of either the DBD fused to GFP or the DBD and all the remaining C-terminal region that constitutes the LBD (as already stated in the legend to Fig.S2: *“Quantification of comparative FRAP assays of GFP fusions with the DNA binding domain of NR5A2 alone (DBD NR5A2) or with its C-terminal moiety containing the ligand binding domain (DBC-Ct)*”). Specifically, the DBD is considered the region spanning the two zinc fingers and a c-terminal extension characteristic of orphan nuclear receptors. The sequence is aa: “MDEDLEELCP VCGDKVSGYH YGLLTCECK GFFKRTVQNN KRYTCIENQN CQIDKTQRKR CPYCRFKKCI DVGMMKLEAVR ADRMRGGRNK FGPMYKRDRRA LKQQKKALIR”. The main reason for this was to test whether the LBD, a source of known protein-protein interactions, was influencing the intrinsic recovery time of the DBD: we show that it does in interphase but not in mitosis. This is an important result that agrees with the strong reduction of potential interactors we observe in our mitotic proteome.

10. How reproducible are the FRET results? Fig 1D and Fig S1B seem to show somewhat different results for NR5A2, with little recovery in interphase for S1B and some recovery in interphase for 1D at 20 s. Are there other variables between these experiments?

Fig.R6 Un-normalized FRAP data of Nr5a2 for Fig.1D (red) and Fig.S1B (blue)

We agree that, while qualitatively speaking Nr5a2 shows little recovery, particularly compared to Esrrb, the two plots to which Reviewer 1 refers to are apparently showing some variation in the recovery of Nr5a2 in interphase. They are two explanations for this. First, the two plots do not cover the same temporal range; second, these two experiments were performed for different purposes. For the one shown in Fig.S2B, the aim was to compare Esrrb and Nr5a2 and we performed FRAP for both with the same settings as we normally FRAP Esrrb (Festuccia Nature Cell Biology 2015, Genome Research 2019). For the one shown in Fig.2D we aimed at strongly bleaching Nr5a2 to follow its recovery during a much longer time. In both cases, we min-max normalized the data as explained in the methods of the original version. In Fig.R6 we present to the referee the data without min-max

normalization for the assay shown in Fig.S2B (blue) and Fig.2D (red). While we purposely FRAPed at different intensities in both assays, we believe that the results are quite similar.

11. For all cells shown in Fig 2, negative controls for background fluorescence using wild-type ES cells should be included.

We are not sure to understand the rational of this request as we do not think this is a canonical control in FRAP experiments. Moreover, our analyses correct each measure of the photobleached spot with a measurement at a non-photobleached area, which corrects for general biases of fluorescence measurements (autofluorescence and the progressive loss of intensity during imaging).

12. For the FRAP data, what is the positive control of a freely moving protein to show full recovery?

GFP alone has been used in FRAP assays many times and its fluorescence proved to be extremely rapidly recovered after photobleaching. However, in our case this control seems inappropriate because GFP will fully diffuse in the huge cytoplasm of mitotic cells. Thus, comparing its behavior to chromosome-bound proteins would be highly misleading.

13. How comparable is the ectopic expression of DBD-, DBD-LBD-GFP fusions and endogenous proteins by Western blotting?

We agree that this information is lacking in the manuscript. The DBD- and DBD-LBD-GFP fusions are ectopically expressed at high levels compared to the endogenous protein, as now shown in the new Fig.S2E (left). However, we do not make a direct comparison between these two and the endogenous, what we compare is their differential behavior in interphase and in mitosis, focusing on cells with approximately equivalent expression levels (Fig.S2E, right).

14. Please provide evidence of purity of mitotic ES cells and state how this was determined.

Our lab has a long-lasting experience in producing high quality mitotic cells (Festuccia et al. Nature Cell Biology, 2016; Festuccia et al. Genome Research 2019; Owens et al. Elife 2019; Dubois et al. Development 2021; Chervova et al. Embo Reports 2022). As described in the methods we assessed the purity by DAPI

staining and microscopy, which is the more direct way to determine the number of cells with mitotic figures. All preparations with more than 5% of remnant interphase cells are systematically discarded. The basis of this contamination threshold was experimentally established in a previous publication (Festuccia et al. Genome Research 2019).

15. Please clarify whether binding profiles were obtained with wild-type ES cells or using tagged fusion TFs.

As mentioned in point 5, all our ChIP-seq assays were done with tagged proteins expressed from their endogenous loci. This was clearly stated in the methods of the previous version but we have further clarified this.

16. “We conclude that the presence of different versions of the ESRRB/NR5A2 motif... are direct determinants of the behavior of ESRRB and NR5A2 in mitosis” (203). The motifs were not mutated. Therefore, the binding and the motifs correlate but it cannot be concluded that the motifs are “direct determinants”, which overstates the data.

We agree with this correction. The sentence now reads: “*are directly related to*”

17. The discussion states that “we have identified a family of gene regulators, nuclear receptors, as potentially common mitotic bookmarking factors” (335). In fact, the paper shows that two nuclear receptors, of which ESRRB had been shown previously, bookmark chromatin in mitosis. The above claim would be more substantial if more than two factors, which cooperate in pluripotency anyhow, are shown to have such bookmarking functions. It would be more honest to state that they have identified one more bookmarking factor, NR5A2.

We partially disagree with Reviewer 1 because our ChIP-MS strategy reveals precisely the presence of several nuclear receptors in the close vicinity of chromatin-bound Esrrb in mitosis. Therefore, we consider all the identified nuclear receptors as mitotic bookmarking factors that bind with Esrrb at a significant proportion of chromatin sites. However, it is true that profiling binding of additional nuclear receptors by ChIP-seq would substantiate this interpretation. We now provide this evidence by showing that a subset

of *Esrrb* and *Nr5a2* mitotic targets are also bookmarked by *Esrra*. These results are shown in FigS10 and discussed in page 12 lines 343-352.

18. The discussion section is long and largely speculative. It can be intellectually interesting to consider the broader implications but the balance between conclusions and speculations needs to be maintained. It is therefore advisable to compact the last two paragraphs into a few sentences, especially since the speculations regarding ligand-dependent nuclear receptors are not directly relevant to the orphan nuclear receptors investigated in the manuscript. The speculations on evolution of ligand-dependent nuclear receptors may be better placed into a perspectives or opinion piece.

We agree, we have fully rewritten and significantly shorten the discussion.

Minor comments:

1. The *ESRRB/NR5A2* responsive genes are stated to be “enriched for pluripotency regulators”. How is enrichment quantified in the gene group? It might be more accurate to simply say “include pluripotency regulators”.

Standard gene set enrichment analyses were done using Enrichr, as described in the methods, including an enrichment analysis against a refined collection of pluripotency genes (PluriNetWork WP1763), as specified already in Table S5.

2. A schematic of the DBD-LBD fusions would be helpful. It would appear that these lack other domains normally present in the wild-type protein. What was the rationale for excluding these?

We direct Reviewer 1 to our response to point 9. The constructs include either the DBD or the DBD and all the rest of the C-terminal containing the LBD. Whether the N-terminal part of the protein plays additional functions is of interest but beyond the scope of the manuscript, since the LBD already confers longer residence to the DBD. Specifically, the DBD is considered the region spanning the two zinc fingers and a c-terminal extension characteristic of orphan nuclear receptors.

3. All IF data should include a scale bar.

This has been included

4. “Jumpstarting” is not a scientific term.

We have changed it to “*reactivating*”

5. Biological “significance” (330) is a term that should be reserved for statistical analysis. We disagree, we prefer to specify “*statistical significance*” when referring to this notion.

Reviewer #2:

Remarks to the Author:

This is a well written report on the effects of potential mitotic bookmarking of nuclear receptors on transcriptional reactivation in G1-phase. The study includes a very nice biochemical approach to identify potential mitotic chromatin associated proteins. The authors then follow up on two orphan nuclear receptors ESRRB and NR5A2 and study their retention on mitotic chromatin in relation to binding sites, chromatin context and requirement for post-mitotic gene activation. The subject of post-mitotic genome reactivation should be of interest to readers in gene expression and epigenetics. The quality of the data is generally very good. My comments can be mostly addressed textually.

I believe the discussion could be organized in a manner that integrates some results better with regards to the main hypothesis. For example, are the effect of ESRRB and NR5A2 on nucleosome positioning or chromatin residence times causally linked to “bookmarking”.

More generally, I think it is really difficult not just in this report but in the field in general to distinguish true bookmarking, i.e. the requirement specifically throughout mitosis of a transcription factor, from its requirement in G1. I do not want to hold the present report to higher standards than the field in general, which might entail experiments such as shorter depletion of ESRRB and/or NR5A2 specifically in M-phase and not into G1. However, I think that a more honest title would be appropriate. For example, nuclear receptors cooperate during genome reactivation after mitosis, or something along those lines.

1. My major question relates to the proposed redundancy of ESRRB and NR5A2. Ideally, they would have included separate ESRRB and NR5A2 depletion experiments, especially if NR5A2 has not been studied as a potential bookmark before. Without additional experiments, they could at least compare their downregulated genes to known ESRRB targets and look at differences in binding patterns.

We agree that a more systematic analysis of single depletions would have been more elegant. However, we had already reported that the loss of *Esrrb* alone leads to minor transcriptional effects after mitosis

(Festuccia et al. NCB 2016). Moreover, we have also shown that Nr5a2 KO ES cells display extremely minor transcriptional deregulations, which strongly suggests that its single depletion during M/G1 would be largely inconsequential. Hence, we directly established a double depletion system.

Fig.R7. Comparison of the effects of *Esrrb*/*Nr5a2* depletion for genes known to be activated by these TFs (Festuccia et al. Development 2021), split by FDR during the kinetic, as indicated

We would like to acknowledge Reviewer 2 for suggesting an analysis we had not considered: while we used previously identified targets to analyze our RNA-seq, we did not think of specifically looking at previous targets that do not respond to auxin in our new experiments. This is interesting because these genes are likely secondary targets of *Esrrb*/*Nr5a2* and comparing their association to our CHIP-seq clusters is highly relevant. Hence, we separated previously identified genes activated by *Esrrb*/*Nr5a2* in two categories, those that display an FDR<0.1 at any given time during our kinetic (which we would consider direct targets), and those that do not that would be considered indirect targets (Fig.R7 top vs bottom). While direct targets of *Esrrb*/*Nr5a2* show

attenuated reactivation after mitosis, indirect targets do not. Moreover, direct targets are more enriched in mitotic binding sites of *Esrrb* and *Nr5a2* (dB; 47%) compared to indirect targets (26%; Fisher exact test $p= 8.736e-08$, odds ratio = 2.54). This data supports a functional role of *Esrrb*/*Nr5a2* during the M/G1 transition. Since this analysis takes advantage of the genes statistically called significant in our dataset, it is related to point 6 from Reviewer 1. Therefore, we have combined both analyses in a new Fig.S9. The results are commented page 12 lines 330-334.

2. It may not be surprising that no pluripotency factors were detected in anti-ESRRB CHIP material. Was CHIP-MS carried out with an antibody against a pluripotency factor as point of reference? Also, simply from the literature, where MS had been carried out, have the ESRRB-associated factors been found?

In light of a potential misunderstanding, we point out that we do actually identify many pluripotency TFs in our CHIP-MS, but specifically in interphase and not in mitosis. We also show that the main *Esrrb* interactors previously found by MS (Pluripotency TFs, Mediator, NuRD, SWI/SNF) are present in our

interphase ChIP-MS and rarely in mitosis. All these results are already presented in Fig.1. To make this point clearer, we have slightly modified two sentences: “*In agreement with previous reports^{26,27}, **we found known interactors of ESRRB**, such as other pluripotency TFs and members of the Mediator, NuRD and Swi/Snf complexes, associated with ESRRB in asynchronous cells (Fig.1A, Fig.S1B and Table S1).*” And also “*Notably, **none of the pluripotency TFs associated with ESRRB in interphase** were identified in mitosis*”.

3. “Co-localization” of NR5A2 and ESRRB cannot be appreciated in the images because “coating” is so broad.

We understand the term co-localization can be misleading in this context. This has been changed in the single sentence where the term was used; it now reads: “*NR5A2-GFP fusion proteins were detected with ESRRB on mitotic chromosomes*”.

4. The ESRRB-associated factors could be actually bound to ESRRB or simply reside in proximity on the DNA fragment that was part of the complex. That ESRRB and NR5A2 bind to chromatin independently of each other and have different residence times supports the latter. Have the authors considered DNase treatment in the IP to distinguish between these possibilities more globally?

We thank Reviewer 2 for this suggestion, which we actually already envisioned. However, ChIP-MS works with crosslinked material and we feel the assay will fall short in providing a clear answer. Moreover, if the factors were never binding together and we were simply capturing factors that are “in proximity”, then we would have expected to find other TFs known to coat the mitotic chromosomes, such as CTCF (Owens et al. Elife 2019) for instance, and many others that have this capacity (Raccaud et al. Nat Com. 2019).

5. Please clarify why ESRRB was not a focus, since it would be the prime candidate for overlapping functions with ESRRB.

Esrra is indeed an obvious candidate. We decided to focus on Nr5a2 for three main reasons. First, because we had already two ongoing projects on the consequences of Esrrb/Nr5a2 double depletion, one in ES cells and another in embryos (Festuccia et al. Development 2021 and Biorxiv 2023). Second, because for these projects we had already generated double auxin lines for Esrrb/Nr5a2 and generated cells with Flag-

tagged Nr5a2 expressed from the endogenous loci (Festuccia et al. Development 2021). Third, highly efficient antibodies were not found for Esrra.

To respond to the reviewer's suggestion, after substantial efforts, we have now identified an antibody with sufficient efficiency in ChIP-seq assays. While the antibody does not perform extremely well and gives variable IP results depending on the batch, the observations that can be obtained appear to confirm that Esrra bookmarks the mitotic chromatin alongside Esrrb and Nr5a2 (Fig.R8). In specific, we show evidences of preferential mitotic binding of Esrra at regions bookmarked by Esrrb, as expected from the motif analyses that we had already included in the previous version of the manuscript. This is presented in page 12 lines 341-352) and in new Fig.S10.

Fig.R8. Average binding profiles of ESRRRA at regions bookmarked by Esrrb and Nr5a2 (dB), Esrrb (eB), Nr5a2 (nB) or none (L), in interphase (blue) and in mitosis (red).

6. For the FRAP experiments were the constructs expressed comparably?

We direct Reviewer 2 to Reviewer 1 point 13: this data has now been added to the manuscript.

7. There seem to be ChIP seq peaks that are unique to mitosis, or is this an issue of normalization and scaling. What are the characteristics of these “M-only” sites, motifs, chromatin features etc? How do M-only marked genes re-activate? M-only peaks might offer an opportunity to test M-phase binding specific effects.

We are unsure on which figures Reviewer 2 bases this comment. If the figure in question is Fig3A then the use of different scales has indeed given a misleading impression— mitotic binding events are often of lower magnitude than in asynchronous cells. However, the question is valid, and our previous work has already addressed the possibility that Esrrb makes M-only binding events. We had experimentally concluded that binding specific to mitosis does not occur: in all cases, when a binding event is seemingly specific to mitosis by ChIP-seq, clear signs of binding can be revealed in interphase using ChIP-qPCR (Festuccia et al. NCB 2016).

8. In Fig.5 using fold-change over mitosis might be misleading given the low expression in mitosis. What would the results look like if gene activity is plotted as % of late G1? Also, it is not clear from panels B and C that reactivation of dB genes is faster. It might simply be a higher average level of expression of dB genes since they are bound by higher NR5A2 and ESRRB in interphase. I think that this is an important distinction and should be made clear in the text because it differentiates between actual bookmarking vs transcriptional activity.

We understand the comment regarding our use of fold-change over mitosis; however, we feel we lack additional information from the reviewer to understand exactly the suggested representation of the data. Indeed, while we can certainly express the results shown in Fig.5 as a % of late G1, it remains unclear how the genes should be ranked. Based on other comments from Reviewer 2, we chose to order genes by the “speed” at which they reach (or surpass) the expression measured in our last time-point (120 min), and check whether the correlations to bookmarking by Esrrb/Nr5a2 hold. To analyze this, we built and ordered the heatmap of percentages of expression vs our last timepoint with the mean of t30 to t90. Notably, ordering this set of genes with such criteria reveals an almost identical correlation to the enrichment of

Fig.R9. Reordering of our dataset based on the % of t120 levels with the data shown either as a % of t120 (left) or log2FC to mitosis (middle). On the right, proportion of dB, B and L genes in groups of 1000 genes descending through the heatmap.

dB genes as that shown in the paper (Fig.R9). Moreover, it is clear that the most rapidly reactivated genes tend to be the most strongly reactivated genes when the log2FC to mitosis is considered. We thank the referee for this suggestion, which reinforces our views that dB genes tend to be more rapidly and more strongly reactivated. This has been added as Fig.S6 and is commented page 10, lines 281-285.

The referee also mentions that the average profile of dB genes does not show a more rapid reactivation; however, as written in the text, the reactivation is indeed faster than unbound genes. It is true that compared to L genes we do observe a slight acceleration of dB genes but we agree the major differences are seen when maximal levels are attained. As illustrated above, the speed of reactivation largely correlates with the level of transcription attained.

Additionally, we have to consider that a main limitation of such correlations is how genes and ChIP-seq peaks are associated. We opted in the manuscript to use promoter binding and combine this with promoter-enhancer contacts available from the literature, as explained in the text. Already with this analysis we observe that focusing on promoter-bound Esrrb/Nr5a2 does allow observing a difference in timing between L and B/dB genes (Fig.S5). Since including enhancers has its limits, given the difficulties of enhancer-gene associations, we are now showing an additional analysis using another popular association scheme, by proximity, and ignoring promoter and enhancer binding (Fig.R10). This analysis clearly shows a difference in timing between groups, correlating with a difference in reactivation strength. We have now included this analysis in Fig.S5B and discussed it in the text page 10 lines 272-275.

Fig.R10. Mean reactivation dynamics of dB (red), B (magenta), L (blue) genes together with unbound genes (black) associating genes to each category by the presence of a peak within 50kb of the promoter.

9. Many genes spike in early G1, but this seems to be ignored in the discussion even though Fig.5A suggests spiking at a lot of genes. Please comment.

We have published a paper which focusses specifically on the spiking behavior of the vast majority of genes in ES cells exiting mitosis (Chervova et al. 2022). Therefore, we did not think appropriate to describe this again in the present manuscript. However, we did in fact mention and argue about this point in the first section of the discussion. Perhaps our term of “*global burst*”, with which we refer to the spiking nature of gene reactivation after mitosis, was not clear enough. We have now clarified this in the discussion page 14 lines 403-405.

10. A point related to #7 is brought up in the discussion (line 390): “In accord with this view, the binding by TFs that do not act as mitotic bookmarking factors in our cells, such as OCT4, SOX2 and NANOG, is also associated with accelerated transcription patterns after mitosis in comparison with unbound genes” This is critical because it distinguishes between bookmarking vs transcription activation, and I think this idea should be introduced earlier to help the reader conceptualize what is being studied here. In this regard, the title of the paper seems a bit too strong. The abstract refers to the most “rapidly” activated genes as being bookmarked, but again, this is not fully supported here.

We partially agree with this comment. On the one hand, the fact that genes regulated by non-bookmarking pluripotent TFs (or in fact genes associated with Lost Esrrb/Nr5a2 sites) reactivate more efficiently than those that are not is important: we had tried to make this clear in the manuscript. On the other, we feel that introducing this notion earlier would be difficult, given that a frame of results is needed to understand the statement. In addition, we feel further elaborating on this aspect is relatively risky as it is impossible to exclude that genes for which we do not have evidence of mitotic bookmarking by Esrrb/Nr5a2 may not be bookmarked by other potential bookmarking factors, be it nuclear receptors or not. Therefore, we would prefer to leave the text untouched regarding this aspect. Moreover, we hope that the analyses shown in our response to point 8 more clearly show the association between “rapidly” reactivated genes and mitotic bookmarking by Esrrb/Nr5a2. Nevertheless, we agree that the abstract and the title can be changed to avoid overstating a direct link between bookmarking and gene reactivation speed, which is not formally demonstrated. For the abstract, we have changed “*the most rapidly and strongly reactivated ones*” to “*the most efficiently reactivated ones*”; the new title is: “***Mitotic bookmarking redundancy by nuclear receptors in pluripotent cells***”.

11. Unless I am mistaken, the same point could be made when considering gene reactivation upon IAA treatment in Fig.6B,D. The max is different (also in asynch cells) but the kinetics don't look different.

We agree that for most genes affected by auxin depletion of Esrrb/Nr5a2 the max expression seems to change more than the reactivation speed. However, as we mention in the paper, the effects of Esrrb/Nr5a2 are visible from the very first time points after mitosis (see also our response to point 6 from Reviewer 1). We believe this is a solid sign that the reactivation speed is altered for most genes. Moreover, we also discuss in the text that the timing of reactivation might be due to other mechanisms involved in the global post-mitotic burst we described earlier (Chervova et al. 2022). Therefore, while we agree that mitotic bookmarking is generally thought to control the timing of reactivation (even though this is not strongly supported by any data in the literature), the possibility that its main effect is that of tuning the level of transcription attained early after mitosis should be considered.

12. Fig.6: has it been verified by ChIP that NR5A2 and ESRRB are depleted from chromatin upon IAA treatment?

To perform this assay rigorously we would need to use a calibrated ChIP-seq approach but, as commented in our response to Reviewer 1 point 4 this is not trivial for Esrrb and Nr5a2. The immuno-stainings we now provide, while not directly addressing site-specific interactions, partially reply to this question (point 4 of Reviewer 1, Fig.R4; included in the manuscript in Fig.S7A,B).

13. As to the IAA-upregulated genes, are they direct targets? Is there “repressive bookmarking”?

This is an important question that we had already addressed in the paper, at least indirectly, since we see no statistical evidence for an enrichment of Esrrb/Nr5a2 bookmarking peaks at upregulated genes (Fig.6E,F).

14. The embryoid body experiment seems to be added as an afterthought. It is unclear to this reader how this experiment with heterogenous cells relates to mitotic bookmarking.

The purpose of this experiment was to test if the genes controlled by Esrrb/Nr5a2 during the M/G1 transition, despite being statistically enriched in pluripotency regulators as established by gene set enrichment, display expression changes during the gain/loss of pluripotency that would support such function. The patterns we observe do indeed suggest these genes are associated with important developmental transitions.

Minor:

1. The term “coating” mitotic chromosomes is not ideal since the factors in question occupy distinct regions.

It is today unclear if what observed by microscopy, which we refer to as coating, is directly related to site-specific interactions. We believe coating is an appropriate term as it reflects the diffuse staining observed by microscopy, without suggesting the biochemical mode by which this chromosomal enrichment is established.

2. A bit more info about NR5A2 in the intro would be helpful.

We have now added more info about Nr5a2 in the introduction page 3 lines 59-64, as requested.

3. “the high occurrence of nuclear receptors in mitotic cells is compelling, representing half of the most

confident mitotic hits” Perhaps rephrase, because the protein identification is biased with ESRRB as anchor.

We have changed this sentence to: *“the high occurrence of nuclear receptors associated with ESRRB in mitotic cells is compelling, representing half of the most confident mitotic hits”*

4. This sentence needs to be made clearer: “Indeed, if at nB regions ESRRB is bound in interphase but mostly lost in mitosis, at eB regions NR5A2 is not found in mitosis simply because this TF is not efficiently recruited at these loci even in interphase”.

We have rephrased to: *“Indeed, at nB regions ESRRB is bound in interphase but mostly lost in mitosis whereas at eB regions Nr5a2 is not efficiently recruited neither in interphase nor in mitosis.”*

5. I think Fig.3 D-H could be made supplemental.

We rather believe that, although partially redundant as we look at motifs through different angles, these panels are important as they provide a clear and simple explanation to the different behaviors reported by ChIP-seq.

6. Fig.5B: the annotation is confusing. Please clarify how the % genes are defined. In panel C, could actual mean expression levels be plotted.

We believe the legend to Fig5B clearly states how the % of genes is defined: *“Percentage of genes bound by ESRRB/NR5A2 (left) and assigned to dB, B and L regions (right), for groups of 1000 genes sliding down the heatmap, with a step of 10 genes.”*

We do not think that averaging actual expression levels would be helpful, given the high dynamic range we observe. For instance, the referee can see in Fig.6D that individual pre-mRNAs can range between 1 and 25 or more TPM. To focus on dynamics and relative changes the global level of expression needs to be corrected in a gene-specific manner, something that mitotic normalization and z scoring allows to do.

7. Please add a cartoon of the timeline of G2 arrest release, nocodazole and IAA exposure. Where the cells sorted etc so the reader does not have to search the references.

We have added such cartoon in supplementary material Fig.S7D

8. Line 314; “with nearly 80% of downregulated genes being bound at known regulatory elements” It looks more like 70%

We think the reviewer is omitting the Lost category, which is considered in our statement. We have dB+B+L=85.2% of genes (the text has been corrected to “around 85%”). We note the sentence continues with: “and around 50% being bookmarked by both ESRRB and NR5A2”.

9. Line 423: “super-family of regulators may have a constitutive role”. What does constitutive mean here?
This sentence has been fully removed.

Reviewer #3:

Remarks to the Author:

Chervova et al investigate the redundancy of nuclear receptors Esrrb and Nr5a2 in the mitotic bookmarking of genes. Previously it was shown that the effect of individual factors was modest. Now the authors test the effect of rapidly depleting by degron both Esrrb and Nr5a2 at the same time. This dual effect effect turns out to be big; 1000 genes are not properly re-activated after mitosis, including many pluripotency genes. This shows that Esrrb and Nr5a2 together are highly important for proper reactivation of genes after mitosis in ES cells, which is a major finding, suitable for NSMB.

The authors start with an analysis by ChIP-MS which factors bind (near) Esrrb in asynchronous and mitotic cells. In asynchronous cells they find previously identified proteins but mitotic cells show an enrichment of nuclear receptors or their co-factors.

Esrrb and Nr5a2 bind to mitotic chromosomes independently from each other (show by KO lines for one or other factor). FRAP experiments show that Nr5A2 has a much longer residence time than Esrrb, setting it apart as a stable mark during mitosis. The DNA binding domain of Nr5A2 is sufficient for this. The authors then show that Esrrb and Nr5A2 bind often together on the genome in interphase and during mitosis and provide a rich annotation and characterization of Esrrb, Nr5A2 binding sites. The authors then annotated genes as being bound in mitosis by Esrrb, Nr5A2, both or none and show that genes rapidly activated after mitosis enrich for binding sites of both Esrrb and Nr5A2. The authors then depleted Esrrb and Nr5a2 simultaneously by degron and tested gene reactivation in depleted cells. Although the decrease in reactivation is modest it could be shown that about a 1000 genes was affected, a much higher number than previously shown, and these genes were enriched for Esrrb,nr5a2 mitotic bindings ites and contained many pluripotency genes. Genes activated by Esrrb, nr5a2 mitotic marking were transiently upregulated during the naïve state of early embryogenesis.

The experiments are well performed, precise, information-rich, appropriate and the conclusions correspond to the results.

One minor comment. The color coding of Fig. 1B is unclear. High confidence hits are red but some categories of proteins have their own (variable) color and the light (beige?) color belonging to low confidence mitotic hits. So if proteins have their “own group’s” color they are not mitotic hits and then there are low confidence they are beige and if high confidence they are red? If so, that would be good to indicate in the legend.

We thank the referee for the minor comment: the legend has been changed accordingly.

Decision Letter, first revision:**Message:** 30th Aug 2023

Dear Dr. Navarro,

Thank you again for submitting your manuscript "Mitotic bookmarking redundancy by nuclear receptors in pluripotent cells". I apologise for the delay in responding, which resulted from the difficulty in obtaining suitable referee reports. Nevertheless, we now have comments (below) from the 3 reviewers who evaluated your paper. In light of these reports, we remain interested in your study and would like to see your response to the comments of the referees, in the form of a revised manuscript.

You will see that even though the experts are positive about the extensive revisions and how they further strengthen the work, certain concerns have lingered and require addressing in a revised manuscript. More specifically, the following points should be addressed, in most cases preferably experimentally: points 2 and 3 of reviewer #2, and the following raised by reviewer #1 (auxin washout experiment, different analysis of the already performed G1 washout experiment ("these auxin washout cells should be analyzed for pluripotency and gene expression"), as well as either calibrated, conventional ChIP or WB in case none of the previous is feasible ("NR5A2 and ESRRB are depleted from chromatin upon IAA treatment.") Furthermore, we deem that textually addressing the remaining discussion/clarification points raised by the experts would further boost the value of this manuscript and encourage you to heed them.

Please be sure to address/respond to all concerns of the referees in full in a point-by-point response and highlight all changes in the revised manuscript text file. If you have comments that are intended for editors only, please include those in a separate cover letter.

We expect to see your revised manuscript within 2-3 months. If you cannot send it within this time, please contact us to discuss an extension; we would still consider your revision, provided that no similar work has been accepted for publication at NSMB or published elsewhere.

Reporting Summary:

Data availability: this journal strongly supports public availability of data. All data used in accepted papers should be available via a public data repository, or alternatively, as Supplementary Information. If data can only be shared on request, please explain why in your Data Availability Statement, and also in the correspondence with your editor. Please note that for some data types, deposition in a public repository is mandatory - more information on our data deposition policies and available repositories can be found below: <https://www.nature.com/nature-research/editorial-policies/reporting-standards#availability-of-data>

While we encourage the use of color in preparing figures, please note that this will incur a

charge to partially defray the cost of printing. Information about color charges can be found at <http://www.nature.com/nsmb/authors/submit/index.html#costs>

[Redacted]

Sincerely,

Dimitris Typas
Associate Editor
Nature Structural & Molecular Biology
ORCID: 0000-0002-8737-1319

Reviewers' Comments:

Reviewer #1:

Remarks to the Author:

The authors have to a limited degree addressed some of the comments and suggestions by the reviewer. Most importantly, the first main comment was not addressed in a satisfactory manner. The big unresolved question is whether the mitotic bookmarking by ESRRB and NR5A2 is functionally relevant for gene expression in mES cells. This reviewer, and principally the same comment was made by reviewer 2 in the third paragraph, suggested an auxin washout experiment during the release of mitosis to address this critical missing point. Even if such an experiment is not routine in the field, science does progress and the aim of this work submitted to NSMB is presumably not just to add one more TF to the list of TFs that have been shown to bind to chromatin in mitosis.

The authors have made good headway in proceeding with the suggested experiment and provided evidence that the two TFs are newly synthesized within 2 h after release from

mitosis and auxin washout by Western blotting. They point out that the abundance of the proteins is less at 2 h post-release than in unperturbed cells and decided to terminate the experiment at this point. However, the question was not whether protein abundance is fully restored but whether G1 transcribed TFs are sufficient for proper gene expression and the maintenance of pluripotency. Therefore, these auxin washout cells should be analyzed for pluripotency and gene expression and the required tools plus the approach are in the authors' hands. If the mES cells are showing statistically significant downregulation of genes including pluripotency genes, then it would suggest that the mitotic inheritance (though not necessarily only the chromatin-bound fraction) of the TFs is needed because insufficient amounts of TFs are synthesized de novo in G1 phase. Such a conclusion could still be toned down by the authors in case of concerns that auxin treatment has a detrimental effect on NR5A2 expression. Alternatively, if the mES cells maintain pluripotency and proper gene expression despite starting out with lower TF abundance, then this would strongly argue against a functional relevance of the mitotic bookmarking. Either of these results would be interesting and important to know.

It is also worth pointing out that presumably these TFs can regulate gene expression even when present at less than 100% abundance, given that heterozygous knockout mice – unlike homozygous knockout mice – are viable and fertile. Indeed, Gregorio Gil's lab showed that the Nr5a2 (referred to as FTF) heterozygous mice have merely one third of the amount of Nr5a2 protein compared to wild-type (Castillo-Olivares et al., J Cell Biol 2004). Therefore, it is not inconceivable that even though the abundance of newly synthesized TFs in G1 phase mES cells is lower than in unperturbed cells, they might still be sufficient to regulate gene expression. I therefore do not see an argument as to why the auxin washout cells as presented in Fig R1 should not be analyzed for pluripotency and gene expression and consider this a necessary requirement for publication of this manuscript.

The reseeded experiment (Fig R3) does not address the issue of small transcriptional changes. It also does not address the biologically functional relevance of these since the colonies were grown without auxin and thus the TFs are constitutively being degraded and not only during M/G1. This would not seem to be helpful to be included in the manuscript.

This reviewer's comment 4 and reviewer 2's comment 12 are both asking for evidence that NR5A2 and ESRRB are depleted from chromatin upon IAA treatment. The best experiment to address this is calibrated ChIP. Second best is conventional ChIP. A third option, though less definitive, is a Western blot of fractionated chromatin-bound vs soluble proteins with appropriate controls to demonstrate fractionation. The IF images (Fig R4) are not sufficient to settle this point. The authors should in any case discuss the possibility that residual chromatin-bound TF could potentially lead to an underestimation of the transcriptional effects in the results.

Reviewer #2:

Remarks to the Author:

The authors addressed most of my comments and the manuscript is improved. Overall, this is a very nice study. I have only a few remaining suggestions.

1. I believe that one of my main initial points is still valid: The title of the paper seems a bit too strong. Since mitosis-specific bookmarking is challenging to prove for any factor,

and since the current report forgivably falls short of proving this, the title should not promise more than what is being delivered. A title such as "Redundancy among nuclear receptors during genome reactivation after mitosis", or something along those lines would not alter the impact of the study but would be more honest.

2. To address the challenge in differentiating between true mitotic bookmarking by a given factor versus its requirement during early G1, have the authors ever tried to deplete ESRRB and/or NR5A2 only in during G1 entry and compared the results to depletion in mitosis plus G1 entry? For example, treating cells with auxin only during the last one of the four hrs of nocodazole exposure, or only during release into G1 phase. This might give a clue as to a mitosis-specific function.

3. My previous point #2 was of course in reference to the lack of pluripotency factor in anti-ESRRB ChIP material in mitotic cells. Therefore, I had suggested including a control experiment with an anti-pluripotency factor antibody.

4. Also, please add to the text in the Results section actual numbers indicating how many singly and doubly bound sites there are in interphase and in mitosis.

5. It would help to include in the Discussion a deeper consideration of the booking marking "redundancy" of NR5A2 and ESRRB. B category genes (bound in mitosis by only one of the two factors) do not reactivate more quickly than L genes. This suggests that a single factor is insufficient for bookmarking. At first pass this would run counter to the idea of redundancy at least at a subset of genes. In other words, if those genes were bound by both factors in mitosis, they might be functionally bookmarked.

6. Of the 1000 genes downregulated upon auxin treatment, how many are unresponsive to single factor depletion, in this case ESRRB (using data from their previous paper)? That would help address the issue of redundancy. I apologize if this was done and I missed it.

Minor points:

1. I would not annotate regions as nB, eB, dB. The capital letter should refer to the relevant factor(s). Nb, Eb, Db would be preferable.

2. In Fig.S7E there is huge NR5A2 variability between replicates in the (-) auxin condition. Why?

3. Line 343: "the combined bookmarking activity of ESRRB and NR5A2 at promoters and enhancers primarily fosters the activation of around 1000 genes associated with pluripotency" Reword this. Not all 1000 genes are pluripotency genes.

4. Legend to Fig S9C: replace "shacked-off" with "shaken off"

Reviewer #3:

Remarks to the Author:

The authors have corrected my minor comments adequately and seem to have done their best to accommodate the comments of the other 2 reviewers. I support publication.

Author Rebuttal, first revision:

Point-by-point response to the reviewers' Comments

Reviewer #1:

The authors have to a limited degree addressed some of the comments and suggestions by the reviewer. Most importantly, the first main comment was not addressed in a satisfactory manner. The big unresolved question is whether the mitotic bookmarking by ESRRB and NR5A2 is functionally relevant for gene expression in mES cells. This reviewer, and principally the same comment was made by reviewer 2 in the third paragraph, suggested an auxin washout experiment during the release of mitosis to address this critical missing point. Even if such an experiment is not routine in the field, science does progress and the aim of this work submitted to NSMB is presumably not just to add one more TF to the list of TFs that have been shown to bind to chromatin in mitosis.

The authors have made good headway in proceeding with the suggested experiment and provided evidence that the two TFs are newly synthesized within 2 h after release from mitosis and auxin washout by Western blotting. They point out that the abundance of the proteins is less at 2 h post-release than in unperturbed cells and decided to terminate the experiment at this point. However, the question was not whether protein abundance is fully restored but whether G1 transcribed TFs are sufficient for proper gene expression and the maintenance of pluripotency. Therefore, these auxin washout cells should be analyzed for pluripotency and gene expression and the required tools plus the approach are in the authors' hands. If the mES cells are showing statistically significant downregulation of genes including pluripotency genes, then it would suggest that the mitotic inheritance (though not necessarily only the chromatin-bound fraction) of the TFs is needed because insufficient amounts of TFs are synthesized de novo in G1 phase. Such a conclusion could still be toned down by the authors in case of concerns that auxin treatment has a detrimental effect on NR5A2 expression. Alternatively, if the mES cells maintain pluripotency and proper gene expression despite starting out with lower TF abundance, then this would strongly argue against a functional relevance of the mitotic bookmarking. Either of these results would be interesting and important to know. It is also worth pointing out that presumably these TFs can regulate gene expression even when present at less than 100% abundance, given that heterozygous knockout mice – unlike homozygous knockout mice - are viable and fertile. Indeed, Gregorio Gil's lab showed that the Nr5a2 (referred to as FTF) heterozygous mice have merely one third of the amount of Nr5a2 protein compared to wild-type (Castillo-Olivares et al., J Cell Biol 2004). Therefore, it is not inconceivable that even though the abundance of newly synthesized TFs in G1 phase mES cells is lower than in unperturbed cells, they might still be sufficient to regulate gene expression. I therefore do not see an argument as to why the auxin washout cells as presented in Fig R1 should not be analyzed for pluripotency and gene expression and consider this a necessary requirement for publication of this manuscript.

Reviewer 1's emphasis on examining gene expression changes following auxin washout is duly acknowledged. However, our viewpoint differs substantially. First, because while *"merely one third of the amount of Nr5a2 protein"* may be functional in other contexts, as reminded by Reviewer 1, in our case the expression after 2 hours of auxin washout is very low, as we showed in our previous response. Second, because the argument that even a minor re-expression of NR5A2/ESRRB could impact results does not align with the reactivation dynamics observed within the experimental timeline. Indeed, considering the reactivation of most genes is nearly complete 30-40 minutes post-mitosis, the marginal expression of NR5A2/ESRRB after a 2-hour washout implies that reactivation will occur in the presence of negligible levels of these 2 TFs, making the assay superfluous. Executing an intricate RNA-seq release experiment with replicates and several time-points – which is the most rigorous way to address this objection – will not only yield results of limited clarity and utility but also be remarkably resource-intensive. Therefore, this uncertain, laborious and expensive experiment has not been attempted; we hope the referees will understand our arguments.

The reseeded experiment (Fig R3) does not address the issue of small transcriptional changes. It also

does not address the biologically functional relevance of these since the colonies were grown without auxin and thus the TFs are constitutively being degraded and not only during M/G1. This would not seem to be helpful to be included in the manuscript.

In contrast to the referee's claim, the TFs are NOT constitutively being degraded since auxin was only added during the release from mitosis and the colonies were then allowed to grow for 7 days without auxin. Therefore, we believe that this experiment is important as it tests the physiological long-term relevance of losing ESRRB/NR5A2 during a single M/G1 transition.

This reviewer's comment 4 and reviewer 2's comment 12 are both asking for evidence that NR5A2 and ESRRB are depleted from chromatin upon IAA treatment. The best experiment to address this is calibrated ChIP. Second best is conventional ChIP. A third option, though less definitive, is a Western blot of fractionated chromatin-bound vs soluble proteins with appropriate controls to demonstrate fractionation. The IF images (Fig R4) are not sufficient to settle this point. The authors should in any case discuss the possibility that residual chromatin-bound TF could potentially lead to an underestimation of the transcriptional effects in the results.

The notion of incomplete depletion of chromosomal ESRRB/NR5A2 raised by Reviewer 1 was a legitimate concern initially shared by Reviewer 2. While we understand the need for stringent validation, it's worth noting that Reviewer 2 was satisfied by our previous response. As previously explained, technical constraints preclude a calibrated ChIP-seq approach due to antibody issues; a regular ChIP-seq will not address the question. Even though the existing stainings have been meticulously conducted and provide substantial insights into the depletion of chromosomal proteins, we have now included two additional analyses to address the reviewer criticism: 1/ we have quantified the level of chromosomal signal in the previous stainings (new Fig.S7C); 2/ we have performed a Western blot of fractionated chromatin versus cytoplasm in mitotic cells (new Fig.S7D). Both analyses clearly show the depletion of ESRRB/NR5A2 from mitotic chromatin is acute.

Reviewer #2:

The authors addressed most of my comments and the manuscript is improved. Overall, this is a very nice study. I have only a few remaining suggestions.

1. I believe that one of my main initial points is still valid: The title of the paper seems a bit too strong. Since mitosis-specific bookmarking is challenging to prove for any factor, and since the current report forgivably falls short of proving this, the title should not promise more than what is being delivered. A title such as "Redundancy among nuclear receptors during genome reactivation after mitosis", or something along those lines would not alter the impact of the study but would be more honest.

While we initially agreed to tune down our first title, we do not think the current one requires further changes. The main reason behind this decision is that we believe our paper comes to one important and fully supported conclusion: the fact that several nuclear receptors act as mitotic bookmarking TFs. In our view the term mitotic bookmarking is properly used in the title to denote TFs engaging in site-specific binding in mitosis.

2. To address the challenge in differentiating between true mitotic bookmarking by a given factor versus its requirement during early G1, have the authors ever tried to deplete ESRRB and/or NR5A2 only in during G1 entry and compared the results to depletion in mitosis plus G1 entry? For example, treating cells with auxin only during the last one of the four hrs of nocodazole exposure, or only during release into G1 phase. This might give a clue as to a mitosis-specific function.

Reviewer 2's suggestion of altering auxin treatment schemes to differentiate between mitosis and G1 effects is insightful. Nonetheless, rapid and nearly complete gene reactivation occurs within 30-40 minutes post-mitosis and such short times would not be sufficient to achieve substantial reduction of ESRRB and NR5A2 by starting auxin treatment in G1. Indeed, ESRRB/NR5A2 depletion remains incomplete even after 1h, which justifies our choice to treat ESCs for 4h during nocodazole arrest. Consequently, the experimental design will fail to capture the intended differences.

3. My previous point #2 was of course in reference to the lack of pluripotency factor in anti-ESRRB ChIP material in mitotic cells. Therefore, I had suggested including a control experiment with an anti-pluripotency factor antibody.

We thank Reviewer 2 for this clarification. The suggestion to conduct a control IP targeting a pluripotency TF during mitosis is however not fully justified according to our previous publications. Indeed, we have already demonstrated that the regions bound by ESRRB in mitosis are strongly enriched in regions not binding other pluripotency TFs (NANOG, OCT4, SOX2), even in interphase (Festuccia et al. Genome Research 2019). Consequently, the absence of pluripotency TFs in our mitotic mass-spec is an anticipated outcome and aligns with the observed behavior of ESRRB, NANOG, OCT4 and SOX2 during cell division. Pursuing this control will not yield substantial insights beyond what we have already documented. Moreover, we believe this point is beyond the scope of our current paper.

4. Also, please add to the text in the Results section actual numbers indicating how many singly and doubly bound sites there are in interphase and in mitosis.

This information was already present in the results, page 9: *"This led to gene groups of similar size: dB, 3529; B, 2594; L, 2671; unbound 5163."*

5. It would help to include in the Discussion a deeper consideration of the bookmarking "redundancy" of NR5A2 and ESRRB. B category genes (bound in mitosis by only one of the two factors) do not reactivate more quickly than L genes. This suggests that a single factor is insufficient for bookmarking. At first pass this would run counter to the idea of redundancy at least at a subset of genes. In other words, if those genes were bound by both factors in mitosis, they might be functionally bookmarked.

While dB genes display the highest difference with L genes in terms of post-mitotic reactivation kinetics, we do not think it is entirely true that this is not the case for B genes. Indeed, as we show in Fig.S5 and comment in the text page 10, B genes do indeed reactivate more rapidly than L genes when considering either promoter-restricted bound events (Fig.S5B top left) or binding events within 50kb of the TSS (Fig.S5B bottom); it is only when considering enhancer elements that this correlation is lost (Fig.S5B top right). Since linking enhancers to direct responsive genes is not trivial, even when using hiC-based information as we did here, we cannot exclude that this issue is more related to incorrect extrapolations of enhancer-promoter pairs than to a lack of functionality of single bookmarked events at enhancers. Thus, we believe it would be more rigorous to avoid excessive elaboration on these aspects. We have clarified our observations page 10: *"Moreover, ignoring promoters and enhancers, and associating genes to dB, B and L groups by proximity (<50kb), further confirmed that both the timing and the strength of gene reactivation were favored by ESRRB/NR5A2 binding in mitosis **and to a lower extent by binding of a single factor (Fig.S5B).**"*

6. Of the 1000 genes downregulated upon auxin treatment, how many are unresponsive to single factor depletion, in this case ESRRB (using data from their previous paper)? That would help address the issue of redundancy. I apologize if this was done and I missed it.

We did not comment on this point because only around 130 genes were found responsive to ESRRB

depletion in early G1 in our previous work, where we used a rather slow Dox-inducible system to deplete ESRRB. To address the reviewer comment, we have compared the two lists and found that only 42 genes of the previously ~130 are present in our new list of ~1000 genes. This indicates that more than half of the previously identified responsive genes in early G1 are indirect ESRRB targets that are downregulated only after a prolonged ESRRB depletion. We have now added the following sentence page 12: ***“Of note, only 42 downregulated and 38 upregulated genes were found to be differentially transcribed during early G1 upon the single loss of ESRRB⁵, supporting the notion of ESRRB/NR5A2 redundancy”.***

Minor points:

1. I would not annotate regions as nB, eB, dB. The capital letter should refer to the relevant factor(s). Nb, Eb, Db would be preferable.

We prefer to keep our current notation.

2. In Fig.S7E there is huge NR5A2 variability between replicates in the (-) auxin condition. Why?

We agree that the level of expression of NR5A2 is variable in our cultures. While we have not studied this in detail, NR5A2 expression is known to be relatively low expressed and highly heterogeneous in ES cells cultured in FCS+LIF; it is particularly sensitive to small changes in experimental conditions such as confluence, volume of medium, frequency of media change and passaging, etc. We cannot exclude the variability we observe is related to one or several of these or other similar parameters.

3. Line 343: “the combined bookmarking activity of ESRRB and NR5A2 at promoters and enhancers primarily fosters the activation of around 1000 genes associated with pluripotency” Reword this. Not all 1000 genes are pluripotency genes.

The sentence now reads: “... the activation of around 1000 genes ***enriched in regulators or markers of pluripotency***”.

4. Legend to Fig S9C: replace “shacked-off” with “shaken off”

Done as requested.

Decision Letter, second revision:

Message: Our ref: NSMB-A47175B

31st Oct 2023

Dear Dr. Navarro,

Thank you for submitting your revised manuscript "Mitotic bookmarking redundancy by nuclear receptors in pluripotent cells" (NSMB-A47175B). It has now been seen by the original referee that retained reservations and their comments are below. Though this reviewer deems that further experimentation is necessary for the paper to be accepted, given the explicit support of the other two experts from the previous round and what was deemed as experiments that would be quite useful, but would not significantly change or boost the value of the manuscript, we have editorially discussed and decided that the paper can be accepted in principle in Nature Structural & Molecular Biology, pending revisions to comply with our editorial/formatting guidelines and to discuss experimental limitations and caveats, in accordance with the guidelines of reviewer #1, as well as toning down certain conclusions and statements, in accordance to the guidance of reviewer #2 from the previous round.

We are now performing detailed checks on your paper and will send you a checklist detailing our editorial and formatting requirements in about two weeks. Please do not upload the final materials and make any revisions until you receive this additional information from us.

To facilitate our work at this stage, it is important that we have a copy of the main text as a word file. If you could please send along a word version of this file as soon as possible, we would greatly appreciate it; please make sure to copy the NSMB account (cc'ed above).

Sincerely,

Dimitris Typas
Associate Editor
Nature Structural & Molecular Biology
ORCID: 0000-0002-8737-1319

Reviewer #1 (Remarks to the Author):

The authors have not addressed the first main comment from the first review, which I explicitly stated as "a necessary requirement for publication of this manuscript" in the second review round. To reiterate, the big unresolved question is whether the mitotic bookmarking (or mitotic inheritance) by ESRRB and NR5A2 is functionally relevant for gene expression in mES cells. I do not accept that the experiment is argued away because

of a different viewpoint. The data needs to speak for itself.

There are two possible outcomes for de novo TF synthesis in G1 phase without mitotic inheritance. 1) Gene expression and especially pluripotency-related genes are largely unchanged, in which case this is strong evidence for lack of a functional requirement of mitotic inheritance or bookmarking in regulating pluripotency gene expression. This is important to know and lies at the heart of this work. 2) Gene expression is affected. This would suggest that mitotic inheritance is functionally important and that potentially the abundance of TFs in G1 phase matters for proper gene expression.

The authors seem to argue along the lines that there is too little re-expression of the TFs after auxin washout. However, it is not possible to know what is "too little" in mESCs, given that a 66% reduction of Nr5a2 protein is compatible with life. If the TF abundance is below a critical threshold, then the most likely result is that gene expression is affected. This would at face value support the authors' model. The experimental limitations and interpretations can be discussed in the manuscript. But if TF mitotic inheritance is not essential because de novo TF synthesis in G1 phase can rescue gene expression, then this would be crucial to know.

I stand by my comments from rounds 1 and 2 that the auxin washout experiment is a requirement for publication of this manuscript.

Final Decision Letter:

Message 30th Nov 2023

:

Dear Dr. Navarro,

We are now happy to accept your revised paper "Mitotic bookmarking redundancy by nuclear receptors in pluripotent cells" for publication as an Article in Nature Structural & Molecular Biology.

As soon as your article is published, you can generate your shareable link by entering the DOI of your article here: http://authors.springernature.com/share. Corresponding authors will also receive an automated email with the shareable link

Your paper will be published online soon after we receive proof corrections and will appear in print in the next available issue. You can find out your date of online publication by contacting the production team shortly after sending your proof corrections. Content is published online weekly on Mondays and Thursdays, and the embargo is set at 16:00 London time (GMT)/11:00 am US Eastern time (EST) on the day of publication. Now is the time to inform your Public Relations or Press Office about your paper, as they might be interested in promoting its publication. This will allow them time to prepare an accurate and satisfactory press release. Include your manuscript tracking number (NSMB-A47175C) and our journal name, which they will need when they contact our press office.

About one week before your paper is published online, we shall be distributing a press release to news organizations worldwide, which may very well include details of your work. We are happy for your institution or funding agency to prepare its own press release, but it must mention the embargo date and Nature Structural & Molecular Biology. If you or your Press Office have any enquiries in the meantime, please contact press@nature.com.

Please note that *Nature Structural & Molecular Biology* is a Transformative Journal (TJ). Authors may publish their research with us through the traditional subscription access route or make their paper immediately open access through payment of an article-processing charge (APC). Authors will not be required to make a final decision about access to their article until it has been accepted. <https://www.springernature.com/gp/open-research/transformative-journals> Find out more about Transformative Journals

Authors may need to take specific actions to achieve <https://www.springernature.com/gp/open-research/funding/policy-compliance-faqs> compliance with funder and institutional open access mandates. If your research is supported by a funder that requires immediate open access (e.g. according to <https://www.springernature.com/gp/open-research/plan-s-compliance> Plan S principles) then you should select the gold OA route, and we will direct you to the compliant route where possible. For authors selecting the subscription publication route, the journal's standard licensing terms will need to be accepted, including <https://www.springernature.com/gp/open-research/policies/journal-policies> self-archiving policies. Those licensing terms will supersede any other terms that the author or any third party may assert apply to any version of the manuscript.

Sincerely,

Dimitris Typas
Associate Editor
Nature Structural & Molecular Biology
ORCID: 0000-0002-8737-1319